# A new look at the environmental conditions favorable to secondary ice production.

A. Korolev[1], I. Heckman[1], M. Wolde[2], A.S. Ackerman[3], A.M. Fridlind[3], L. Ladino[1,4], P. Lawson[5], J. Milbrandt[1], E. Williams[6]

[1]Environment and Climate Change Canada, Toronto, ON, Canada
[2]National Research Council, Ottawa, ON, Canada
[3]NASA Goddard Institute for Space Studies, New York, NY, USA,
[4]Centro de Ciencias de la Atmósfera, Universidad Nacional Autonoma de Mexico, Mexico City, Mexico,
[5]Stratton Park Engineering Company, Boulder, CO, USA
[6]Massachusetts Institute of Technology, Boston, MA, USA

*Correspondence to*: Alexei Korolev ([alexei.korolev@canada.ca](mailto:alexei.korolev@canada.ca))

**Abstract.** This study attempts a new identification of mechanisms of secondary ice production (SIP) based on the observation of small faceted ice crystals (hexagonal plates or columns) with typical sizes smaller than 100 μm. Due to their young age, such small ice crystals can be used as tracers for identifying the conditions for SIP. Observations reported here were conducted in oceanic tropical mesoscale convective systems (MCS) and mid-latitude frontal clouds in the temperature range from 0°C to -15°C and heavily seeded by aged ice particles. It was found that in both MCSs and frontal clouds, SIP was observed right above the melting layer and extended to higher altitudes with colder temperatures. The roles of six possible mechanisms to generate the SIP particles are assessed using additional observations. In most observed SIP cases, small secondary ice particles spatially correlated with liquid phase, vertical updrafts and aged rimed ice particles. However, in many cases, neither graupel nor liquid drops were observed in the SIP regions, and therefore, the conditions for an active Hallett-Mossop process were not met. In many cases, large concentrations of small pristine ice particles were observed right above the melting layer starting at temperatures as warm as -0.5°C. It is proposed that the initiation of SIP above the melting layer is stimulated by the recirculation of large liquid drops through the melting layer with convective turbulent updrafts. After re-entering a supercooled environment above the melting layer they impact with aged ice, freeze and shatter. The size of the splinters generated during SIP was estimated as 10 μm or less. A principal conclusion of this work is that only the freezing drop shattering mechanism could be clearly supported by the airborne in-situ observations.

## 1. Introduction

Secondary ice production (SIP) has long been acknowledged as a fundamental cloud microphysical process (e.g., Cantrell and Heymsfield, 2005; Field et al., 2017). Along with the other leading processes in cold clouds, such as primary ice formation via activation of ice nucleating particles (INPs), particle vapor growth, aggregation, riming and sedimentation, SIP is likely to commonly play a critical role in the formation of size distributions and habits of ice particles (e.g., Ackerman et al. 2015; Ladino et al. 2017). Through the modulation of ice particle concentration, SIP can thereby impact precipitation formation, rate of glaciation of mixed-phase clouds, the longevity of ice clouds, cloud electrification, and radiative properties of clouds. On the global scale, SIP may significantly impact the hydrological cycle and climate in general. However, the commonality and precise mechanisms of SIP have remained

persistently poorly established. Understanding of mechanisms of SIP is of great importance for developing a parameterization of the ice initiation processes in weather prediction and climate models.

The significance of SIP was recognized only after the beginning of regular airborne studies of cloud

microstructure in different geographical regions (e.g. Koenig 1963, 1965; Hobbs, 1969; Mossop, 1970, 1985; Mossop et al. 1972; Ono, 1972; Hallett et al. 1978; Hobbs and Rangno 1985, 1990; Beard 1992; and many others). A systematically observed difference of up to five orders of magnitude between concentrations of INP and measured ice concentration urged provision of an explanation of the physical processes underlying this discrepancy. One of the explanations suggested an enhancement of the concentration of ice particles via a mechanism unrelated to the primary

ice formation. Several possible mechanisms were proposed to explain such so-called secondary production of ice crystals.

Historically, the first proposed mechanism to explain SIP focused on droplet fragmentation during freezing (e.g. Langham and Mason, 1958; Mason and Maybank, 1960; Kachurin and Bekryaev, 1960). During the freezing of a cloud droplet, isolated pockets of liquid water may become trapped inside an ice shell. The expansion of water during

subsequent freezing results in an increase of pressure inside the ice shell. If the pressure exceeds a critical point, then the ice shell may break into fragments to relieve the internal pressure. Newly formed ice fragments may serve as INPs and result in an enhancement of ice concentration.

Subsequent laboratory studies demonstrated that fragmentation of freezing drops depends on many factors such as droplet temperature before freezing, environmental temperature, droplet size, concentration of $CO_2$ and other gases

dissolved in water, the crystalline nature of the ice shell (i.e. monocrystalline or polycrystalline), drop rotation during freezing, and the type of INP employed for droplet freezing and the manner of droplet suspension in the laboratory (Muchnik and Rudko 1961; Evans and Hutchinson 1963; Stott and Hutchinson 1965; Dye and Hobbs 1966, 1968; Johnson and Hallett 1968; Brownscombe and Thorndike 1968; Hobbs and Alkezweeney 1968; Takahashi and Yamashita 1969, 1970; Pitter and Pruppacher 1973; Takahashi 1975,1976; Wildeman et al. 2017; Lauber et al. 2018).

A review of the laboratory studies of droplet freezing showed a large diversity of reported results, and conditions required for droplet shattering during freezing remain not well understood.

Splintering during ice particle riming is another mechanism that can potentially explain apparent SIP (Macklin, 1960; Latham and Mason, 1961). Hallett and Mossop (1974) and Mossop and Hallett (1974) observed splinter formation during riming in a cloud chamber with liquid water content of ~1 g/m$^3$ and droplet concentration 500 cm$^{-3}$.

They found that splinter production is active in the air temperature range from -3°C to -8°C, and its rate has a pronounced maximum at an air temperature of -5°C and drop impact velocity of 2.5 m/s. At these conditions, one splinter was produced per 250 droplets of diameter $D>24$ μm. The phenomenon of splinter production during riming is usually referred to as the Hallett-Mossop (HM) mechanism. Several studies have aimed at understanding the physical mechanism responsible for the splinter production (e.g. Choularton et al., 1978, 1980; Emersic and

Connolly, 2017). However, despite these efforts, the physical mechanism underlying this phenomenon is still under debate.

The collision of ice particles may result in their mechanical fragmentation and the production of secondary ice. This hypothesis was stimulated by observations of ice particle fragments collected during airborne studies (e.g. Hobbs and Farber, 1972; Takahashi, 1993) and ground-based ones (Juisto and Weikmann, 1973). Collisional fragmentation

of ice particles was explored in the laboratory by Vardiman (1978) and Takahashi et al. (1995). However, the
       obtained results do not allow an unambiguous conclusion about ice-ice collisional fragmentation and its contribution
       in SIP.

       When an ice crystal collides with a supercooled drop, it will experience thermal shock due to the release of latent
       heat of the freezing drop. This will cause a differential expansion of the ice crystal and may result in its
fragmentation. This phenomenon was observed during laboratory studies by Dye and Hobbs (1968) and Hobbs and
       Farber (1972). Due to the current lack of laboratory studies, the efficiency of ice particle fragmentation due to thermal
       shock and its effect on SIP remains inconclusive.

       Ice particle fragmentation and formation of secondary ice may occur during sublimation in subsaturated areas near
       cloud edges or underneath the cloud base. The phenomenon of fragmentation during sublimation was studied by
Oraltay and Hallett (1989), Dong et al. (1994), Bacon et al. (1998). However, it remains unclear whether small
       fragments formed in the sub-saturated environment can re-enter supersaturated cloud and act as SIP particles. This
       appears to be a significant limitation on the efficacy of sublimation breakup as a SIP mechanism.

       Gagin (1972) proposed a mechanism for SIP due to the activation of INP in high transient supersaturation areas
       around freezing drops. After nucleation, the freezing drop temperature rises to 0°C. If the surrounding air is colder
than 0°C, the surface of the freezing drop acts as a source of water vapor to a colder environment. The resulting water
       vapor diffuses radially outward. Depending on the air humidity, it may create at some distance from the droplet a
       region with supersaturated air. Rosinski et al. (1975) and Gagin and Nozyce (1984) studied nucleation of INPs around
       suspended freezing drops with 1–2 mm diameter. However, simply due to limited laboratory studies, the effect of INP
       activation around freezing drops on SIP remains insufficiently quantified.

The hypothesis that ice concentration measurements are widely subject to artifacts induced by airborne
       instruments has been discussed over a long period of time. Larger ice particles may bounce off a forward probe's tips
       or inlet, and shatter into smaller fragments. After rebounding, the shattered fragments may travel into the sample area
       and cause multiple artificial counts of small ice (e.g. Gardiner and Hallett, 1985; Gayet et al., 1996; Heymsfield,
       2007; McFarquhar et al. 2007; Jensen et al. 2009; Field et al., 2003). The following introduction of antishattering K-
tips (Korolev et al. 2013b) along with the interarrival time algorithm (Field et al. 2006) allowed for a significant
       mitigation of the effect of shattering and an improvement in the ice particle measurements. As was shown by Korolev
       et al. (2011, 2013a) and Lawson (2011) a measured concentration of ice particles smaller than 200μm can be
       enhanced due to the shattering effect by up to two orders of magnitude.

       The latter finding brings up a question that some early airborne studies that pointed out the discrepancy between
concentrations of ice particles and INPs might be contaminated by shattering artifacts, which resulted in an
       enhancement of the measured concentration of small ice. However, numerous recent in-situ measurements, which
       applied the antishattering techniques, are in general consistent with the early SIP observations and they also showed
       that in many clouds, ice particle concentrations are still much higher than the INP concentration (e.g. Crosier et al.
       2011, 2014; Crawford et al. 2012; Stith et al. 2014; Lawson et al, 2015; 2017; Lloyd et al. 2015; Lasher-Trapp et al.
2016; Keppas et al. 2017; Ladino et al. 2017; Sullivan et al, 2017, 2018; and others)

       Another source of artifacts in measurements of high concentration of ice by optical array probes (OAP), is related
       to fragmentation of particle images when particles pass through the sample volume close to the edge of the depth-of-

field (DoF) (Korolev 2007). A few one-two pixel images resulted from fragmentation of large out-of-focus images have an enhanced artificial contribution into particle concentration due to their very small sample volumes. This problem is recognized by many research groups. One solution to this is the exclusion of the first two or three size bins compromised by the ambiguity of the DoF definition and contamination by image fragments. Due to the extent that particles from the first two or three size bins (<30 - 80μm depending on the OAP type) may significantly contribute to the total ice concentration, a limitation is imposed on the measurements of total concertation of ice particles in SIP cloud regions.

Most observations of an enhanced concentration of ice particles have been attributed to the HM-process. The list of these studies extends over 30 publications, so we name only a few of them here (e.g. Ono, 1971, 1972; Harris-Hobbs and Cooper, 1987; Bower et al. 1996; and others). In these studies, the conclusions about the HM-process were obtained based on the observed association with graupel and columnar ice crystals. Fewer studies attributed observations of high ice concentration to drop shattering (e.g. Koenig 1963, 1965; Braham, 1964; Rangno 2008; Lawson et al. 2017). Ice-ice collisional fragmentation was identified as a source of SIP in natural clouds by Hobbs and Farber (1972), Takahashi (1993), Schwarzenboeck et al. (2009). As can be seen, the identification of SIP gravitates towards the HM-process, whereas mechanisms such as activation of INP in transient supersaturation around freezing drops, ice fragmentation due to thermal shock or sublimation were not even considered. In this regard the question that arises is, could these observations reflect an actual occurrence of different types of SIP?

The present study is focused on revisiting the role of different SIP mechanisms and identifying conditions favorable for SIP. Cloud regions with ongoing ice multiplication were identified with the help of a new technique based on the identification of small faceted ice crystals smaller than 60-100μm measured by a Cloud Particle Imager (CPI). The newly developed technique was applied to the data set collected in mature tropical mesoscale convective systems (MCS) and in midlatitude frontal clouds. The roles of six possible mechanisms to generate the SIP particles are assessed using additional observations: fragmentation of freezing drops, splintering during the HM-process, ice-ice collisional breakup, ice fragmentation during thermal shock, fragmentation during ice sublimation, and INP nucleation in transient supersaturation. The variety of environmental conditions associated with SIP will be considered based on six specific cases sampled tropical MSC (4 cases) and midlatitude frontal clouds (2 cases).

## 2. Data sets

Measurements were conducted from the National Research Council (NRC) Convar580 research aircraft during two field campaigns: High Ice Water Content (HIWC) and the Buffalo Area Icing and Radar Study 2/Weather Radar Validation Experiment (BAIRS2/WERVEX).

The HIWC flight operations were conducted out of Cayenne (French Guiana) in May 2015. A total of fourteen Convair580 research flights were conducted in the frame of the HIWC campaign with the average flight endurance of approximately 4 hours. Most of the flights were performed in oceanic MCS in altitudes ranging from 6500m to 7200m and temperatures from 0°C to -15°C. The observations of MCSs were performed during their mature stages, when the area of clouds with longwave brightness temperatures colder than -50°C from GOES13 approached or surpassed its maximum. At that stage, most of the volume of the MCS above the freezing level was nearly glaciated,

with embedded mixed-phase regions mainly associated with vertical updrafts (Korolev et al., 2018). However, the studied MCS during the observations remained dynamically active with updrafts peaking at 15-20m/s.

The BAIRS2/WERVEX flight operations were conducted over Southern Ontario and Upstate New York from January to March 2017. A total of five research flights were conducted in precipitating frontal cloud systems. In the framework of this study, the analysis will be focused on two flights performed on February 7 and March 24, 2017 in
the range of altitudes from 1500m to 3000m and temperature ranges from +5°C to -10°C.

The NRC Convair-580 was equipped with state-of-the-art cloud microphysical and thermodynamic instrumentation. Size distributions of aerosol particles were measured by a DMT Ultrahigh Sensitivity Aerosol Spectrometer (UHSAS) (Cai et al. 2008). Measurements of ice particle number concentration and ice water content (IWC) were extracted from composite particle size distributions measured by optical array 2D imaging probes (OAP)
a PMS 2DC (Knollenberg, 1981), a SPEC 2-Dimensional Stereo 2DS (Lawson et al. 2006) and a DMT Precipitation Imaging Probe PIP (Baumgardner et al. 2001). Cloud droplet size distributions were measured by a PMS Forward Scattering Spectrometer Probe FSSP (Knollenberg, 1981) and a DMT Cloud Droplet Probe CDP (Lance et al. 2010). Cloud particle images were measured with the SPEC CPI (Lawson et al. 2001). Bulk liquid water content (LWC) and total water content (TWC) were measured with a SkyPhysTech Nevzorov probe (Korolev et al. 1998) and a SEA
IsoKinetic probe (IKP) (Davison et al. 2011). A Rosemount Icing Detector was used for detection of liquid water at $T$<-5C (Mazin et al. 2001). The extinction coefficient was measured with the ECCC Cloud Extinction Probe (Korolev et al. 2014). Vertical velocity was measured by Rosemount 858 (Williams and Marcotte, 2000) and Aventech AIMMS-20 (Beswick et al. 2008). The Convair-580 was also equipped with NRC Airborne W-band and X-band radars (NAWX) with Doppler capability (Wolde and Pazmany, 2005). The UHSAS and IKP were employed only
during the HIWC project and were not used during BAIRS 2/WERVEX.

In order to mitigate the effect of shattering artifacts on ice particle measurements (Korolev et al. 2011), all cloud particle probes were equipped with anti-shattering K-tips (Korolev et al. 2013a). The remaining shattering artifacts were filtered out during data post-processing with the help of the modified interarrival time algorithm (Korolev and Field, 2015).
The collected cloud microphysical data were processed with the help of the ECCC D2G software. This software allowed composite visualization and analysis of cloud microphysical, thermodynamic, radar, and aircraft data probes.

### 3. Methodology

#### 3.1 Basic assumptions

If initiation of secondary ice occurs in a supersaturated environment, then the newly formed ice particles start growing through water vapor diffusion and some fraction of secondary ice particles may turn into faceted ice crystals. If the growth time is shorter than certain typical time $\tau_{corr}$, then these faceted ice crystals may still be associated with the environment of their origin. At time scale $t > \tau_{corr}$ the size and shape of ice crystals may undergo significant metamorphosis, and secondary ice particles may lose their spatial correlation with the environment of their origin due to
horizontal and/or vertical advection and turbulent diffusion. This process is schematically shown in **Fig.1**.

This concept was used to develop a method for the identification of SIP regions. This method is based on the following approximations:

(1)  Small faceted ice crystals (hexagonal plates or columns) originate from secondary ice production.

(2)  During some time $\tau_{corr}$, the newly formed ice crystals remain associated with the environment where they originated.

If these approximations are valid, then small pristine ice crystals can be used as tracers of the environmental conditions favorable to SIP. The following subsections aim to assess $\tau_{corr}$ and the typical size of small faceted ice crystals.

### 3.2  Ice crystal habits

In order for an ice crystal to grow as a hexagonal prism, its growth begins as a monocrystalline ice particle.

As discussed in the introduction, most potential SIP mechanisms are related to the fragmentation of existing ice particles. Since water drops frozen at $T_a > -15°C$ tend to be monocrystalline (e.g. Pitter and Pruppacher, 1973; Hallett 1964), their fragments will also be monocrystalline. In addition, if a large ice particle is polycrystalline, the probability of its small fragment to be monocrystalline remains high. Therefore, the condition of monocrystallinity is expected to be satisfied for most small ice fragments with $L_{max}$ <40-50μm. Formation of ice fragments with typical sizes down to 20μm is supported by video material of the breakup of freezing drops from Wildeman et al. (2017) and Lauber et al. (2018)

### 3.3  Assessment of spatial correlation time

Condition (2) in section 3.1, requires assessment of a typical time ($\tau_{corr}$) such that for time $t < \tau_{corr}$, the changes of cloud environment parameters (e.g. air temperature ($T_a$), humidity ($RH$), ice particle concentration ($N_i$), droplet concentration ($N_d$), liquid water content ($LWC$), ice water content ($IWC$), etc.) are insignificant, and the SIP-generated ice particles remain within this environment.

In order to assess $\tau_{corr}$, the main typical time scales of cloud dynamics and kinetics, such as the time of phase relaxation $\tau_p$, glaciation time $\tau_{gl}$, turbulent diffusion time $\tau_t$ vertical advection time $\tau_v$, and particle residence time $\tau_r$, have to be estimated.

The time scale $\tau_p$ characterizes the response of the cloud environment to changes of in-cloud humidity (e.g., due to entrainment, vertical motion, interaction between liquid and ice phases). So, in order for $RH$ to relax to its steady-state value, it is required that

$$\tau_p < \tau_{corr} \tag{1}$$

For mixed-phase clouds, after neglecting the effect of the vertical velocity, $\tau_p$ can be written as (Korolev and Mazin, 2003)

$$\frac{1}{\tau_p} = \frac{1}{\tau_{p\ ice}} + \frac{1}{\tau_{p\ liq}} \tag{2}$$

where $\tau_{p\ ice} = \frac{a_i(T,P)}{N_i \bar{r}_i}$ is the time of phase relaxation in the ice clouds, $\tau_{p\ liq} = \frac{a_l(T,P)}{N_l \bar{r}_l}$ is the time of phase relaxation in liquid clouds, $N_i$, $N_l$, $\bar{r}_i$, $\bar{r}_l$ are the concentrations and average radii of ice particles and liquid droplets, and $a_i$, $a_l$ are coefficients dependent of pressure $P$ and temperature $T$.

The glaciation time scale characterizes the transit time of the mixed-phase cloud into an all-ice cloud due the Wegener-Bergeron-Findeisen (WBF) process (Wegener, A., 1911; Bergeron, T., 1935). This process results in

complete evaporation of liquid droplets ($N_d(t > \tau_{gl}) = 0$) and changes of steady-state relative humidity ($RH(t > \tau_{gl}) \to RH_{s\,ice}$).

Therefore, it is required that

$$\tau_{corr} < \tau_{gl} \tag{3}$$

The glaciation time scale can be estimated as (Korolev and Mazin, 2003)

$$\tau_{gl} = \frac{b(T,P)}{S_i}\left(\left(\frac{W_{l0} + W_{i0}}{N_i}\right)^{\frac{2}{3}} - \left(\frac{W_{i0}}{N_i}\right)^{\frac{2}{3}}\right) \tag{4}$$

where $S_i$ is the supersaturation over ice at saturation over water; $W_{l0}$, $W_{i0}$ are the initial liquid and ice water content, respectively; $N_i$ is the concentration of ice particles; $b(T,P)$ is the coefficient dependent of pressure $P$ and temperature $T$.

Turbulent mixing results in a spatial transport of the SIP particles and a decrease in their concentration. Turbulent mixing may result in biases in the assessment of the spatial scales of the SIP regions and the concentration of the SIP

particles. Therefore, $\tau_{corr}$ should relate to the turbulent mixing time as

$$\tau_{corr} < \tau_t \tag{5}$$

The typical time of turbulent mixing of a cloud parcel with a spatial scale $L$ can be estimated as (e.g. Landau and Lifshitz, 1987)

$$\tau_t = \varepsilon^{-\frac{1}{3}}L^{\frac{2}{3}} \tag{6}$$

where $\varepsilon$ is the turbulent energy dissipation rate.

Vertical transport of a cloud parcel affects $T_a$ and $RH$. Assuming an adiabatic temperature change $\Delta T$, the typical

time of vertical transport can be written as

$$\tau_v = \frac{\Delta T}{u_z \gamma_w} \tag{7}$$

where $u_z$ is the vertical velocity, and $\gamma_w$ is the moist adiabatic lapse rate. So, in order to limit the amplitude of $T_a$ and $RH$, $\tau_{corr}$ and $\tau_v$ should relate as

$$\tau_{corr} < \tau_v \tag{8}$$

Residence time of an ice particle is determined by the fall velocity $u_{ice}$ and cloud parcel size $L$ and is equal to

$$\tau_{res} = \frac{L}{u_{ice}} \tag{9}$$

In order that the ice particle remains in the cloud volume, it is required that

$$\tau_{corr} < \tau_{res} \tag{10}$$

Summarizing Eqs.(1),(3),(5),(8),(10) yields the condition for $\tau_{corr}$

$$\tau_p < \tau_{corr} < \min(\tau_{gl}, \tau_t, \tau_v, \tau_{res}) \tag{11}$$

Typical values of $\tau_p, \tau_{gl}, \tau_t, \tau_v, \tau_{res}$ will be assessed for the following conditions: $T_a = -5°C$, $P = 700$ mb, $N_i = 200$ L$^{-1}$, $N_d = 100$cm$^{-3}$, $\bar{r}_d = 8$μm, $\bar{r}_i = 100$μm, $L = 200$-300m, $\varepsilon = 10^2$ m$^2$/s$^3$, $u_z = 1$-4 m/s, temperature change limit $|\Delta T| < 2°C$, vertical fall velocity of a solid column with $L_{max} = 100$μm and $u_{ice} = 0.1$m/s.

Substituting $T, P, L, \varepsilon, N_d, N_i, \bar{r}_d, \bar{r}_i, \Delta T, u_{ice}$ in Eqs.(2),(4),(6),(7),(9) yields to $\tau_p \approx 5$s, $\tau_{gl} \approx 320$s, $\tau_t \approx 160$s,

$\tau_v \approx 80$s, $\tau_{res} \approx 2000$s. It should be noted that $\tau_p, \tau_{gl}, , \tau_t, , \tau_v$ are sensitive to the above parameters and may be

different from the obtained estimates. However, the above assessment provides the magnitude of the typical times for SIP cloud regions. Based on the above estimates, it would be reasonable to assume that $\tau_{corr}$ should not exceed 60-120s.

### 3.4 Assessment of ice particle sizes

The estimate of $\tau_{corr}$ allows for the assessment of ice particle sizes that they may grow up to during this time. Since SIP is expected to occur in liquid or mixed-phase clouds, then the water vapor humidity will be close to saturation over water (Korolev and Isaac, 2006).

**Figure 2** shows the calculated length of columns, which were grown by water vapor deposition at saturation over liquid water at different temperatures. The results of the calculations are in good agreement with the laboratory studies of ice growth in Fukuta and Takahashi (1999). As shown in **Fig.2**, during $\tau_{corr}$ the length of hexagonal columns $L_{max}$ may reach 50µm to 150µm depending on the temperature and the aspect ratio ($R = h/2a$). Based on this assessment, for the following identification of SIP, the size of small faceted crystals will be limited by $L_{max} < 100$µm.

### 3.5 Identification of SIP particles

Acquisition of small ice particles images was conducted with the help of the SPEC CPI (Lawson et al. 2001). The CPI was designed for recording 256 grey-level images of ice particles with 2.3µm resolution at a rate of up to approximately 500 images per second. Even though the acquisition rate of particle images is lower than that for 2D-imaging optical array probes, the CPI provides crisp, high-resolution, photographic quality images of small ice particles. This feature is critical for the goals of this study. Binary OAP images (e.g. SPEC 2DS, DMT CIP, PMS 2DC) have lower pixel resolution (from 10µm to 25µm), and their appearance may be significantly modified by diffraction effects (e.g. Korolev 2007; Guélis et al. 2019).

Identification of small pristine ice particles from the CPI imagery was performed with the help of a pre-trained convolutional neural network (Krizhevsky et al., 2017) fine-tuned for the identification of small hexagonal faceted ice crystals. The habit of faceted ice particles was limited to hexagonal prism type crystals: columns, short columns and plates. Examples of CPI images that were used in the final tuning are presented in **Fig.3a**.

Validation, based on hand-labeled images held out from training (950 from each of the three categories), showed that only 4% were misclassified. Although the occurrence of small faceted ice crystals was rare, since they also tended to appear in clusters, a clear signal of their occurrence could be seen above noise from false positives.

Examples of images of small ice particles falsely identified as pristine faceted ice are shown in **Fig.3b**. As it is seen from **Fig.3b**, the centers of growth of the ice crystals are absent in the images. From a crystallographic viewpoint, such crystals cannot be formed during vapor deposition growth, and they are most likely the result of breakups after impact with the CPI inlet (Appendix A). Such particles were excluded from the analysis as described in Appendix A.

It is worth noting that some or similar images with irregular shapes as in **Fig.3b** could be a result of SIP, and therefore have a natural origin. Thus, fragments of droplets shattered during freezing may appear as irregularly shaped ice before they develop facets. So, the assessment of the concentration of the SIP particles based on the estimates of the concentration of small faceted ice particles can be considered as a lower limit.

In this study, the sizes of particle images are estimated from the maximum size of the image measured in all possible directions ($L_{max}$). Note, that for randomly oriented hexagonal thin plates $L_{max}$ provides an estimate of the diameter of the prism base ($a$) with accuracy better than 15%. For hexagonal columns, $L_{max}$ is not representative of the prism height $h$, and depending on the column orientation, it can be either $L_{max} > h$ or $L_{max} < h$.

Due to the uncertainty of the CPI sample area definition affected by the settings of acceptance of out-of-focus images during sampling and post-processing, we will be using counting rate (s$^{-1}$) of small faceted ice particles to characterize their concentration. The assessment of the concentration of faceted ice provided in the foregoing discussion was done based on the comparisons of the CPI counting rate of droplets with $D > 40\mu m$ and that measured by 2DS. After identification of the scaling coefficient for the conversion of the CPI droplet rate into concentration, this coefficient was applied to the counting rate of small hexagonal crystals. This procedure is based on the approximation that the droplets and ice crystals $< L_{max}$ are in the same size range and their CPI sample volumes are approximately the same. The accuracy of such estimation of the concentration of small ice particles is estimated as ±50%.

## 4. Results

### 4.1 SIP observations in tropical MCSs

In this section, we present the observations of SIP during the Convair-580 flight in a tropical MCS on May 15, 2015. The MCS was located off the shore of French Guiana with its center approximately 350km north-east of Cayenne. **Figure 4** shows two GOES13 infrared images of the MCS with an overlay of Convair-580 flight tracks. During the flight leg in **Fig.4a** (UTC 09:23-10:22) the altitude varied between 5600m and 5700m with the air temperature ranging from -4°C to -6°C. As it is seen in **Fig.4a,** the Convair-580 crossed three convective cells with the cloud top brightness temperatures ranging between approximately-55°C and -65°C (marked by dashed circles). The flight leg in **Fig.4b** (UTC 11:23 - 12:07) was performed at altitudes ranging from 7000m to 7300m and temperatures from -11°C to -15°C. Despite its decaying stage, the MCS remained dynamically active at the Convair-580 flight level. As will be discussed below, it was found that SIP was observed in convective cloud regions indicated by circles in **Fig.4a,b.**

**Figure 5** presents a time series of cloud microphysical parameters corresponding to the flight leg in **Fig.4a**. The top panel (**Fig.5a**) shows the CPI counting rate of small faceted ice crystals with $L_{max} < 60\mu m$ and $100\mu m$. Grey vertical strips indicate cloud sections identified as SIP regions. In this cloud segment, the concentration of small pristine ice with $L_{max} < 100\mu m$ attains values up to $N_{pr100} \approx 500$L$^{-1}$. Based on the discussion in section 3 the origin of these small pristine ice crystals is attributed to the vicinity of the level of their observation.

After including aged pristine ice crystals with $L_{max} < 200\mu m$, the concentration of faceted ice crystals reached $N_{pr200} \approx 900$L$^{-1}$. As was shown in Ladino et al. (2017) the estimated INP concentration remained nearly constant during the flight operations in French Guiana, and for the temperature range -6°C$< T <$-4°C it was approximately $N_{INP} \sim 10^{-2}$L$^{-1}$. So, the estimated $N_{INP}$ is nearly four to five orders of magnitude lower than the concentration of small pristine ice particles $N_{pr100}$ and $N_{pr200}$. Therefore, the observed small ice particles cannot be explained by heterogeneous ice nucleation, and the most likely pathway of their formation is SIP.

To address the question regarding conditions favorable for SIP, we explore the correlations of different microphysical parameters. As seen from Table 1, the ice particle concentration has the highest correlation coefficient with droplets $D >60\text{-}80\mu$m. In many apparent SIP regions, droplets over 300µm in diameter were registered by the CPI. However, in some cloud regions with $D >60\mu$m, small faceted ice was not observed. Such cloud regions in **Fig.5** are indicated by pink strips.

**Table 1**. Correlation coefficient between droplet concentration in different size ranges and concentration of small faceted ice crystals with $D_{\max} <100\mu$m for the cloud segment in **Fig.5** for 30 s and 60 s averaging.

| Dropl.Conc. | $D>20\mu$m | $D>40\mu$m | $D>60\mu$m | $D>80\mu$m | $D>100\mu$m |
|---|---|---|---|---|---|
| Corr.Coeff. (30s) | 0.48 | 0.66 | 0.85 | 0.77 | 0.69 |
| Corr.Coeff. (60s) | 0.56 | 0.71 | 0.9 | 0.85 | 0.8 |

The analysis of the entire HIWC data set showed that, as a rule, SIP was not observed or was very unproductive in supercooled liquid clouds with droplets $D_{\max} <40\mu$m. One of such cases in **Fig.5** is indicated by a yellow strip. In this specific cloud region, the maximum size of droplets measured by FSSP and CDP did not exceed $D_{\max} =30\mu$m.

Comparing **Fig.5a,f** also indicates that intense SIP was observed in cloud regions with enhanced turbulence or vertical updrafts. Yet in the regions on the left side of **Fig.5a** (UTC 09:33-09:38), SIP was observed in the absence of any significant turbulence or updraft ($u_z < 0.2$m/s).

**4.1.1 Case 1**

**Figure 6** shows CPI images of cloud particles from a 5-second cloud segment (UTC 09:40:33 – 09:40:38) in **Fig.5**. This cloud segment is characterized by an enhanced concentration of small faceted ice particles ($L_{\max} < 100\mu$m) estimated as approximately $N_{pr100} \approx 450$ L$^{-1}$. The majority of the CPI images of droplets are larger than 40µm diameter with drizzle size drops up to 200µm (**Fig.6a**). The droplet concentration measured by FSSP and CDP is quite low and varies from 2cm$^{-3}$ to 6cm$^{-3}$, whereas the concentration of droplets with $D > 40\mu$m assessed from the CPI and 2DS data varies between 1cm$^{-3}$ and 3cm$^{-3}$.

Some of the droplets, identified as frozen and indicated in **Fig.6a** by blue frames, have distorted shapes and bulges. As documented by Lauber et al. (2018) the formation of bulges may be accompanied by bubble bursting or jetting, which may be a primary source of SIP particles. A few other droplets in the red frames appear as fragments of shattered droplets. Altogether, the presence of droplet fragments and frozen droplets with bulges are supportive of SIP from shattering of freezing drops.

The concentration of frozen drops in **Fig.6a** is estimated as $N_{frd} \sim 6$L$^{-1}$. This concentration is still much higher than the concentration of INP $N_{INP} \sim 10^{-2}$L$^{-1}$ at $T$=-5°C (Ladino et al. 2017), and therefore, droplet freezing cannot be explained by heterogeneous nucleation on INPs alone. This gap serves as a basis for explaining droplet freezing due to impact with splinters produced by shattered freezing drops.

It is worth noting that the actual concentration of frozen droplets in **Fig.6a** may be higher than the estimate $N_{frd}$, since some drops may freeze without deformation, and after complete freezing, they may become transparent again

and appear as liquid drops (e.g. Mason and Maybank, 1960). The phase state of such drops cannot be unambiguously identified and, in the frame of this study, are considered to be liquid.

**Figure 6b** shows images of aged ice particles sampled in the same cloud volume as the newly generated SIP ice particles in **Fig.6a**. The aged ice particles come in two distinct types: faceted columns with $L_{max} < 400\mu m$ and graupel with $L_{max} < 1000\mu m$. Presence of graupel is a necessary condition for the HM-process (Hallett and Mossop 1974). However, visual analysis of graupel images (**Fig.6b**) shows that their surfaces appear smooth without small-scale features. This appearance suggests that liquid droplets spread over the graupel's surface and freeze as a film.

The way in which the droplets spread is determined primarily by the droplet's size and air temperature (Macklin and Payne, 1969; Dong and Hallett, 1989).

        The surface of graupel in **Fig.6b** appears different than the surfaces of rimed ice cylinders in lab experiments on secondary ice production (Macklin, 1960; Choularton and Latham, 1978; Choularton et al. 1980; Emersic and Connolly 2017). The surfaces of the rimed ice cylinders were highly inhomogeneous with distinct images of frozen

droplets and small features down to $10\mu m$, which presumably serve as a source of splintering. Comparing these observations with laboratory studies poses a question regarding whether graupel without small scale features, as in **Fig.6b**, could produce splinters.

        Another condition for the HM-process is the presence of droplets smaller than $12\mu m$ (Mossop, 1978, 1985). For the case in **Fig.6b**, the concentration of droplets with $D <15\mu m$ is estimated from the CDP and FSSP data to be

$0.5cm^{-3}$ to $1cm^{-3}$. The probability of graupel collision with droplets at such a small concentration is likely too low to have any significant effect on the HM-process.

        **4.1.2 Case 2**

        **Figure 7a** shows another 5-second segment with successive cloud particle images measured by the CPI in another SIP region (UTC 09:46:39-09:46:44). Enlarged cloud droplets and SIP particles from **Fig.7a** are shown in **Fig.7b**.

The concentration of SIP particles is estimated as $70L^{-1}$, which is lower than the previous case. The concentration of droplets with $D > 40\mu m$ is also lower, and it is estimated from the 2DS and CPI measurements as $0.2-0.3cm^{-3}$. The droplet concentration with $D<40\mu m$ measured by FSSP and CDP is approximately $1cm^{-3}$. However, due to the large concentration of ice in this cloud region, half of the FSSP and CDP measured concentration ($\sim 0.5cm^{-3}$) may be caused by shattering artifacts (Korolev et al. 2013b). No droplets larger than $70\mu m$ were observed in this cloud segment.

As seen from **Fig.7a**, the background aged ice is represented by columnar shaped particles with well-developed facets with minor riming. Some ice particles highlighted by purple frames have features of recirculation. These particles started their growth as columns at $-8°C< T_a <-4°C$, then they were ascended to a plate growth condition (e.g. $-18°C< T_a <-12°C$) and turned into capped columns. Then they were brought down by a downdraft or sedimented back to the columnar growth environment ($-8°C< T_a <-4°C$) and developed columns growing out of the

plate edges.

        What is important about the case in **Fig.7** is that no graupel, heavily rimed ice or significant amount of liquid droplets were observed here. Therefore, the SIP in this specific cloud region formally does not meet the HM-process criteria.

        **Figure 8** shows a time series of microphysical and state parameters in the same cloud area as in **Fig.5** but at a

higher altitude (7000m< $H$ <7300m) and lower temperature ($-14°C< T_a <-12°C$). This locale offers the opportunity

to consider the evolution of ice crystals initiated at lower levels, and to explore the initiation of new ice in colder environments. **Figure 8a** shows that small faceted particles are spread horizontally over the entire cloud environment. The clustering of the small ice parties and their association with updrafts and liquid droplets is less pronounced than at the temperature level of -4°C to -6°C (**Fig.5**). As follows from **Figs.8b-f**, the liquid phase appears in horizontally narrow segments associated with vertical updraft regions. As discussed in Korolev (2007) updrafts may extend the maintenance of the liquid phase in mixed-phase clouds or completely suppress the WBF process. The majority of the cloud segment in **Fig.8** is associated with high IWC peaking up to $3g/m^3$ within an ice number concentration up to $1cm^{-3}$. Liquid phase with no updraft in this kind of environment can exist only for a short time period. For example, a mixed-phase cloud with $LWC \sim 0.1g/m^3$ and $u_z = 0$ will be glaciated within 50s at $T = -10C$.

**4.1.3 Case 3**

**Figure 9a** presents a sequence of cloud particle images measured during a 10 second time interval (UTC 12:05:31-12:05:41) at $T_a = -14°C$ and $H = 7250m$. The measurements were conducted within a moderate updraft $2m/s < u_z < 6m/s$. As it is seen, aged ice particles are represented by graupel, a few lightly rimed particles, and numerous columns. The origin of columns is related to nucleation at lower levels (~5300-5700m) at temperatures corresponding to columnar growth ($-10°C < T_a < -4°C$).

**Figure 9b** shows a subset of zoomed-in images of droplets and small faceted ice particles extracted from **Fig.9a**. The majority of the small faceted ice particles are hexagonal plates. According to Magono and Lee (1966), these types of plates are expected to form in the near-saturated-over-water air within the temperature range $-12°C < T_a < -18°C$. Hence, the origin and growth habit of the observed plates is consistent with the temperature range where they were sampled.

The concentration of droplets with $D < 40\mu m$ is estimated from FSSP and CDP as less than $1cm^{-3}$, and the concentration of droplets with $D > 40\mu m$ is estimated from 2DS as $\sim 2cm^{-3}$. Therefore, even though the ensemble of particles in **Fig.9** contains graupel, the rest of the parameters, such as temperature and concentration of small and large droplets, are well outside the envelope of conditions required for the HM-process, as documented in the literature.

**4.1.4 Case 4**

**Figure 10a** shows another example of ice particles sampled approximately one kilometer away from those shown in **Fig.9**. This cloud region is characterized by the absence of liquid phase. However, the concentration of small ice particles in **Fig.10** appears to be even higher than that of the small ice in **Fig.9**, where liquid droplets were present. It is worth noting that, in most observational studies, the presence of liquid was considered as one of the necessary conditions for SIP. However, in this particular case, it can be argued that the absence of liquid droplets may be explained by their evaporation as a result of the WBF process just before the cloudy air arrived at the level of observation. The small ice plates in **Fig.10b** could be formed at lower levels with temperatures $-14°C < T_a < -12°C$ when liquid droplets were still present in the parcel. After that, the plates ascended in the glaciated updraft to a higher level.

The variety of habits of small ice particles in **Figs.9** and **10** show that SIP apparently occurred continuously during ascent through different levels, with temperatures ranging from -2°C to -14°C (at the level of observation).

**Figure 11** shows a summary of the concentrations of small faceted ice crystals and droplets averaged over the entire Convair-580 HIWC data set. This data was collected in ten tropical MCSs with a total sampling length of 9580km within the temperature range -15°C< $T_a$ <0°C. It was found that small faceted ice crystals, along with cloud drops, occurred in spatial clusters with a typical horizontal extension from a few hundred meters to a few kilometers. In many cases, regions with liquid droplets and regions with enhanced concentrations of the small ice may be separated by a few hundred meters or kilometers. In these SIP cloud regions, the concentration of drops and SIP particles is significantly higher than their average values shown in **Fig.11**.

**Figure 11** shows that, on average, the concentration of SIP particles increases, and the concentration of liquid droplets decreases with increasing height within the entire bulk of MCSs at -15°C< $T_a$. These trends may be related to the cumulative effect of vertical transport of SIP particles by the convective updrafts.

### 4.2  SIP observations in mid-latitude frontal clouds

The next observation of SIP was conducted in clouds associated with mid-latitude winter frontal systems during the BAIRS2/WERVEX project on March 24th, 2017. **Figure 12** shows GOES 16 IR image (a) and Buffalo NEXRAD reflectivity (b) overlaid with the Convair-580 flight track. The cloud regions identified as SIP are indicated by dashed circles.

**Figure 13** shows a one-hour segment of in-situ cloud microphysical measurements sampled from the Convair-580. During these measurements, the Convair-580  performed a series of porpoise and spiral ascents and descents in the vicinity of the melting layer with altitude and temperature changing in the ranges of 2400m<$H$<4200m, and -6°C< $T_a$ <+2°C, respectively.

It turned out that in mid-latitude frontal clouds the correlation between the concentration of small faceted ice crystals and liquid droplets is very similar to that observed in tropical MCSs at $T_a$ >-6°C. The correlation coefficients between the concentrations of droplets with different diameters and small faceted ice particles are shown in Table 2. As follows from Table 2 the best correlation is reached for droplets with $D$ >40μm. Whereas for the tropical MCS best correlation is reached for droplets with $D > 60$ μm (Table 1).

Similar to tropical MCSs, in frontal clouds SIP was not observed in liquid and mixed-phase clouds with $D$ <30μm. Such cloud segments are indicated by yellow strips in **Fig.13**. Most cases of SIP in **Fig.13** were associated with cloud regions with enhanced turbulence ($u_z \sim \pm$ 3m/s).

**Table 2**. Correlation coefficient in different size ranges between droplet concentration and concentration of small faceted ice crystals with $L_{max} < 100$μm for the cloud segment in **Fig.13** with 30s and 60s averaging.

| Dropl.Conc. | $D$>20μm | $D$>40μm | $D$>60μm | $D$>80μm | $D$>100μm |
|---|---|---|---|---|---|
| Corr.Coeff. (30s) | 0.44 | 0.51 | 0.48 | 0.26 | 0.11 |
| Corr.Coeff. (60s) | 0.65 | 0.71 | 0.59 | 0.29 | 0.18 |

### 4.2.1 Case 5

**Figure 14a** shows a sequence of CPI images of cloud particles from a 40s cloud segment with enhanced concentrations of small faceted ice crystals. In this cloud region, the concentration of small ice crystals with $L_{max} <$

100µm peaked up to approximately $N_{pr100} \approx 1000L^{-1}$. Like the case in **Fig.6** a number of frozen drops with deformed shapes (blue frames) were observed in this SIP region. The concentration of visually identified frozen drops is estimated at approximately $N_{frd} \approx 30L^{-1}$. During the BAIRS2/WERVEX project, the UHSAS probe was not installed on the Convair-580, and therefore, the concentration of INP could not be assessed using the approach from Ladino et al. (2017). However, the estimated concentrations $N_{pr100}$ and $N_{frd}$ still appear to be much higher than expected INP concentrations of $10^{-6}$ to $10^{-3}$ L$^{-1}$ at a -2°C to -5°C temperature range (e.g. Kanji et al., 2017, DeMott et al. 2016; Price et al. 2018; Welti et al. 2018; Creamean et al. 2018; Wex et al. 2019).

The aged ice particles in **Fig.14b** are represented by rimed columns and graupel-like particles. Therefore, this case is consistent with the conditions required for the HM-process.

In **Fig.14b**, there are a few ice particles with small faceted crystals stuck to their surfaces, which are indicated using brown frames. The origin of small faceted ice on the surface of large particles may be explained by: (1) vapor deposition regrowth of rime into faceted crystals, or (2) aggregation of newly formed small and pre-existing large ice particles. Option (1) may not be relevant to the particles in **Fig.14b**, since a closer look at the small particles reveals that the centers of their growth are separated from the surface of the large ice particle.

Another argument supporting aggregation is that droplets $D$<100µm, at $T_a$ >-10°C tend to freeze as monocrystals (e.g. Hallett, 1964; Pitter and Pruppacher, 1973). Small droplets freezing on the surface of a monocrystalline particle usually have the same orientation of principal crystallographic axis (e.g. Pitter and Pruppacher 1973; Iwabuchi and Magono, 1975; Uyeda and Kikuchi, 1978). If the rimed droplets continue to grow through vapor deposition, they will regrow into faceted crystals with the orientation of principal axes the same as that of the 'host' crystal. Examples of such ice crystals can be found in **Figs.7** and **9** (brown frames). The alternative to this arrangement is when small faceted ice crystals on the surface of a frozen drop, (brown-red frame) **Fig.14b**, have clearly multi-directional crystallographic orientations. Therefore, these small ice crystals most likely formed independently of the frozen drop before they were aggregated.

It is worth noting that the ice particles in the brown-red frame includes five visible small faceted ice crystals attached to the surface of the frozen drop. Aggregation of the small crystals may be enhanced by electrostatic charges, which fragmented particles may have after shattering. Charge separation during droplet shattering was observed in studies by many research groups (e.g. Mason and Maybank 1960; Kachurin and Bekryaev, 1960; Latham and Mason, 1961; Evans and Hutchinson, 1963; Scott and Hutchinson, 1965; Kolomeychuk et al. 1975). Therefore, the observation of small faceted ice aggregated to the surface of large particles with different orientations of principal axis is supportive of their formation due to SIP.

### 4.2.2 Case 6

**Figure 15** shows another example of a spatial sequence of particle images from a cloud region with enhanced concentrations of faceted ice particles apparently resulting from SIP. What is interesting about this is that the background aged ice particles were not observed here. Ice particles are either faceted ice crystals or frozen drops. The absence of small droplets and graupel suggests that the HM-process is not relevant to this case and that SIP most likely occurred here due to shattering of large drops. This hypothesis is supported by the presence of a large number of images of fragmented (red frames) and deformed frozen drops (blue frames). The presence of such droplets supports the SIP mechanism of shattering of freezing drops. It should be noted that the sizes of most of the faceted ice

crystals in **Fig.15** exceed 100-200µm. Therefore, the age of such particles exceeds the threshold time $\tau_{corr}$ as discussed in section 3.3. However, the purpose of this case is to show another example of SIP, in which the criteria for the HM-process are not met.

Figure 16 shows the average concentration of faceted ice crystals and droplets for two flights from the
510 BAIRS2/WERVEX field campaign. As it is seen, the concentration of drops with $D$ >60µm decreases with the decrease of $T_a$. However, the concentration of faceted ice particles has a maximum at -3.5°C< $T_a$ <-1.5°C. This type of behavior is different from those in tropical MCSs as shown in **Fig.11**. This difference may be explained by the absence of well-defined convective regions present in MCSs, which transport liquid droplets to the upper levels and extend the temperature range of SIP. A narrower SIP temperature range in the studied frontal clouds may be also
explained by SIP regions being associated with the mixed-phase layer embedded into a deep ice cloud. The cloud top temperature of the mixed-phase layers is limited by $T_a$ =-6°C to-7°C, which is well reflected in **Fig.16**.

### 4.3 Effect of aircraft produced ice particles on the measurements

Aircraft-produced ice particles (APIP) (e.g. Rangno and Hobbs, 1983; Woodley et al., 1991) may be confused with SIP ice crystals, and therefore, result in biases in the interpretation of measurements. Contamination by APIP
may occur if the aircraft re-enters the cloud region where the APIP were transported by vertical or horizontal advection. Typically, this may happen if the aircraft traverses through the region of its previous operation.

The contamination by APIP is excluded for the cases 1 and 2 (**Figs.6** and **7**) (sections 4.1.1, 4.1.2) since the Convair580 flew along a nearly straight line and never re-entered regions of earlier operations (**Fig.4a**). The cases 3 and 4 (**Figs.9,10**) (sections 4.1.3, 4.1.4) might be contaminated by APIP since the clouds were sampled in an area
close to which the Convair580 flew 8 minutes earlier. However, since cases 3 and 4 were sampled in a convective region with an updraft velocity $u_z$=2-5m/s (**Fig.8 f**), the potential APIP were expected to be removed from the area of the measurements by vertical wind.

Case 5 (**Fig.14**) (section 4.2.5) was sampled during ascent through the cloud (Fig.13h) at approximately 12:30 (see also **Fig.12a**). This cloud region was not affected by the previous operation of the Convair580, and therefore,
contamination by APIP of this area is dismissed. Similarly, case 6 (**Fig.15**) (section 4.2.6) was sampled during descent through a mixed-phase layer, which was not affected by previous Convair580 flight operations.

### 5.  Initial size of secondary ice particles

Knowledge about the initial size and number concentration of secondary ice is of great importance for the
535 parameterization of SIP processes in atmospheric models, including weather prediction and climate models, particularly when using multi-moment microphysics schemes. The number and size of SIP particles determine the rate of water vapor depletion, release of latent heat, cloud dynamics, and glaciation time. Because of their slow fall velocity, small SIP particles will stay longer in the environment of their origin. Small fragments will also spread faster over clouds being transported by turbulent diffusion or vertical updrafts.  On the contrary, large SIP fragments
will precipitate down and have a shorter residence time in the cloud. Besides that, small ice fragments have higher probability to be monocrystalline, and therefore regrow into pristine faceted ice crystals. Whereas, large ice fragments most likely keep an irregular shape during the subsequent growth by water vapor deposition. The size of the fragments also plays an important role in charge separation and cloud electrification in general (e.g. Jayarante et al.

1983). Altogether, the size distribution of primary SIP particles has a great significance for precipitation production,
radiation properties and lifetime of clouds.

In this section, we will estimate typical initial sizes of the SIP particles. Identification of initial sizes of secondary ice from the CPI imagery may be problematic because of the limited pixel resolution, and ambiguity of distinguishing secondary ice fragments from natural cloud particles. In order to address this issue, we will use an indirect assessment of the initial sizes of secondary ice.

**Figure 17** shows images of ice particles sampled in frontal clouds at temperatures ranging from -1°C to -1.5°C. All small faceted ice crystals in this cloud region appear to be thin plates (red frames in **Fig.17a**). The thickness of the plates $h$ is estimated as varying in the range from 10μm to 20μm. Since the smallest size of drops in this region $D_{min} \approx 40 \mu m > h$, then the origin of these plates cannot be attributed to the deposition growth on frozen droplets.

The plates in **Fig.17a** have plane parallel basal surfaces without steps. None of these thin plates have a visually identifiable center of initial growth. Such a shape is suggestive that the secondary ice particles, on which these plates were formed, were monocrystalline and their initial sizes $L_{min0}$ were smaller than the thickness of the plates, i.e. $L_{max0} < h$. In this case, the secondary ice particles were completely embedded inside the plates and became part of the crystallographic lattice. So, there will be no additional refraction of transmitted light and the plates will appear uniform as in **Fig.17a**. Therefore, the smallest initial size of the secondary ice particles is estimated as $L_{min0} \leq$ 10 μm.

Secondary ice particles representing a large end of their initial sizes are shown in **Fig.17**, which presents images of fragments of shattered frozen drops. Most of these images were collected in SIP regions indicated by grey areas in **Fig.17**. The maximum size of droplet fragments **Fig.17** is limited by $L_{max0} \approx 400$ μm. In general, $L_{max0}$ is determined by the maximum size of ice particles that participate in SIP. Thus, for the case of freezing raindrops, $L_{max0}$ can be extended to a few millimeters.

The obtained estimates suggest that at the moment of initiation, secondary ice particles are represented by a cascade of sizes ranging from 10μm (or smaller) to a few hundred microns (or larger). This estimate of initial sizes of SIP particles is consistent with the videos by Wildeman et al. (2017) and Lauber et al. (2018), which showed a variety of fragments with different sizes formed during shattering of freezing drops.

### 6.    Shapes of small secondary ice particles

The shapes of secondary ice particles that develop during $\tau_{corr}$ may shed light on the environmental conditions associated with the SIP initiation.

A quick look at the ice particle images in **Figure 6, 7, 14, 15, 17** shows that the aspect ratio ($R = h/a$) of small ice crystals (hexagonal prisms) may noticeably vary within the same SIP cloud region.

**Figure 19** shows small faceted ice crystals sampled in different SIP cloud regions (**Fig.5**) with narrow temperature ranges from -5.5°C< $T_a$ <-5°C. As seen from **Fig.19**, despite the minor changes of $T_a$, the habits of small ice crystals varied from plates to long columns, and the aspect ratio changed in the range of 0.3< $R$ <6.

Based on laboratory studies, $R$ depends on the air temperature $T_a$ and supersaturation over ice $S_i$ of the environment where the ice crystals were grown (e.g. Mason, 1971, Kobayashi, 1961; Bailey and Hallett, 2009). Therefore, it is expected that ice crystals that were formed in the same cloud volume and were exposed to the same $T_a$

and $S_i$, should have the same $R$. Thus, the question arises, why do ice crystals with different habits form in the same cloud volume?

There are several possibilities as to how $R$ may vary. The environment with $T_a >$-4C and $S_w > 0$ corresponds to the plate growth condition. Therefore, the plates shown in the upper row in **Fig.19** could be formed a few hundred meters below at $T_a >$-4C and then be brought up to the level of observation with a convective updraft. The internal structure of some plates in the upper row (i.e. images #8,9,11,14, and 15) is indicative of the changing $T_a$ and $S_i$ that ice crystals may experience during ascent.

As seen in **Fig.19**, most of the ice crystals are solid columns and thick plates. Following laboratory studies (*ibid*) such ice habits form at $T_a \approx$-5°C in the environment supersaturated with respect to ice ( $S_i > 0$) but undersaturated with respect to water ($S_w < 0$). Therefore, the cloudy air in the SIP region, despite any presence of liquid drops was undersaturated with respect to water. Such conditions may occur during the repartitioning of water between ice and liquid phases, when the WBF process is active (Korolev and Mazin, 2003, Pinsky et al. 2018).

Ice crystals with $R\sim 1$ may be formed as a result frozen droplets developing facets and turning into isometric hexagonal prisms (e.g. Gonda and Yamazaki 1978, 1984; Magono et al. 1979; Takahashi and Mori, 2006). Long columns with $3 < R < 6$ , shown in the two bottom rows in **Fig.19**, correspond to the growth condition with $S_w \geq 0$ and $T_a \sim$-5°C (*ibid*).

Accordingly, the shape of secondary ice crystals during the early stage of their evolution may vary from plates to solid columns. At a later stage, ice particles metamorphosize in shape in accordance to their evolving $T_a(t)$ and $S_i(t)$. Thus, **Figs.9** and **10** show that columns tend to be the dominant shape of the aged secondary ice particles after ascending from 5600m (-5°C) to 7200m (-15°C). The aspect ratio and size of the aged columns vary in the ranges of $2 < R < 4$ and 150μm$< L_{max} <$450μm, respectively.

### 7. Interaction of secondary ice with the cloud environment

The purpose of this section is to identify how secondary ice particles may evolve after their formation. Understanding of possible scenarios of secondary ice evolution is important for the interpretation of the obtained results and developing cloud simulations. The interactions between secondary ice and environment are specifically important for small ice splinters ($L_{max} <$10μm) due to different types of instability related to this size range. Below we consider four possible scenarios of how secondary ice particles may evolve after their production:

#### 7.1 Vapor deposition growth

This scenario consists of vapor deposition growth of individual secondary ice particles, which requires supersaturation over ice. The necessary condition for this scenario is supersaturation over ice. This condition is satisfied in mixed-phase clouds and in updrafts in ice clouds (Korolev and Mazin, 2003). Examples of the secondary ice particles regrown into hexagonal plates and columns are shown in **Figs.6,7,10,14,15,17**. This scenario conserves the concentration of SIP particles ($N_{SIP}$).

#### 7.2 Scavenging by liquid droplets

Because of the high concentration of droplets in mixed-phase clouds (typically $10^1$-$10^2$cm$^{-3}$), scavenging of secondary ice particles by liquid drops may have a high frequency of occurrence. Examples of images of frozen drops measured in SIP cloud regions are shown in **Fig.20**. Most of these images do not have any large ice crystals attached

to them. Therefore, it would be reasonable to assume that they were nucleated by secondary ice particles, presumably smaller than 10-20μm. More examples of frozen drops in SIP regions can be seen in **Figs.6,14,15,17** (indicated by blue frames). Because of the high concentration of the frozen drops (section 4) their formation cannot be explained by nucleation via heterogeneous INP.

Scavenging of secondary ice particles by liquid droplets may result in shattered freezing drops and an increase in the concentration of secondary ice. This process induces a positive feedback loop and under certain conditions may result in an avalanche increase in the concentration of secondary ice particles. The possibility of ice multiplication due to a chain reaction was proposed in early studies (e.g. Kachurin and Bekryaev, 1960; Mason and Maybank, 1960; Koenig, 1963; Braham, 1964, Mossop et al. 1964; and others). The observation of frozen and fragmented drops inside the SIP regions can be used as evidence that chain reactions are part of the ice multiplication process.

Droplet freezing may also occur without shattering. In this case, frozen drops keep growing through vapor deposition. Examples of large frozen drops with developing facets are shown in **Fig.21**. Observations of frozen drops re-growing into hexagonal prisms, as in **Fig.21**, is indicative that these drops were nucleated by embryonic monocrystalline secondary ice particles. As seen from **Fig.21**, depending on the stage of their growth, some frozen drops developed not only basal and prism faces, but also pyramidal faces. Such evolution of frozen drops was observed in laboratory studies by Gonda and Yamazaki (1978, 1984), Magono et al. (1979), Takahashi and Mori (2006). Additional examples of frozen drops with developed facets can be found in **Figs.14,15,17** (green frames).

### 7.3 Scavenging by aged ice particles

After their initiation, secondary ice particles may be scavenged by aged ice particles. As follows from laboratory studies, shattering of freezing drops is usually accompanied by charge separation (e.g. Mason and Maybank, 1960; Kachurin and Bekryaev, 1960; Evans and Hutchinson, 1963; Stott and Hutchinson, 1965; Kolomeychuk et al. 1975). Static electric charges may significantly enhance the scavenging of secondary ice by liquid drops and/or pre-existing ice, and result in the rapid reduction of the concentration of secondary ice. An example of secondary ice scavenged by bigger ice particles is shown in **Fig.14b**.

### 7.4 Sublimation of secondary ice

Small secondary ice particles may undergo complete sublimation if SIP occurs in the environment undersaturated over ice. For example, at $T_a$ =-5°C and $RH_w$ =90% ($RH_{ice}$=95%) a 10μm ice particle will completely sublimate during $t_{ev} \approx$ 4s.

Subsaturation in ice or mixed-phase clouds may occur due to entrainment of dry air. Thus, Pinsky et al. (2018) showed that in mixed-phase cloud, complete sublimation of small ice crystals during entrainment and mixing of dry air may occur prior to the complete evaporation of liquid droplets.

Ice clouds may also become subsaturated in downdrafts (Korolev and Mazin 2003). Thus, in an ice cloud parcel with $N_{ice}$ =200L[-1], and $D_{ice}(0)$ =200μm, $RH_{ice}(0)$=100%, $T_a(0)$ =-8°C, descending with $u_z$ =-4m/s, relative humidity over ice in $t$ =20s will be $RH_{ice}(t)$=95%. If such a parcel contained ice splinters with $D_{ice} \approx$10μm, they would completely sublimate within 20s. Downdrafts frequently accompany vertical updrafts in dynamically active regions inside MCSs (e.g. **Figs.5f** and **8f**). Therefore, sublimation of newly formed small secondary ice particles may play an important role in suppressing ongoing SIP and the reduction of $N_{SIP}$. **Figure 22** summarises the potential interactions of newly formed secondary ice with a cloud environment.

### 8. Feasibility of different SIP mechanisms

This section revisits the discussion of the SIP mechanisms, which might be responsible for the enhanced concentration of small ice particles.

#### 8.1 Droplet fragmentation/shattering during freezing

     Images of fragmented frozen drops in **Figs.6,14,15** collocated with secondary ice particles explicitly indicate that the SIP mechanism due to shattering of freezing drops is a contributing factor in ice multiplication. A collection of

fragments of frozen drops from other SIP regions is shown in **Fig.18**. Fragments of frozen drops were also documented through in-situ observations reported by Korolev et al. (2004) and Rangno (2008).

     It should be noted that small fragments of frozen droplets may not be identified from the CPI imagery due to limited pixel resolution and issues related to the segregation of irregularly shaped fragments from natural particles. Fragments of large frozen drops may also not be found in the SIP region, since they rapidly leave the region of their

origin due to the fast sedimentation. For these reasons the fragments of shattered frozen droplets may not always be seen by CPI in the SIP cloud regions associated with shattering of freezing drops (e.g. **Figs.7,9,10,17**).

     Drop freezing by impaction of ice splinters is supported by observations of single frozen drops with deformed shapes (**Fig.20**), and frozen drops with partially developed facets (**Fig.20**). Because of the absence of any visible large ice particles attached to them, these drops must have been nucleated by small ice particles.

As it is seen from **Figs.11** and **16**, secondary ice particles were observed at temperatures as warm as -0.5°C and colder than -8°C. These temperatures are outside of the HM and riming-splintering temperature range. However, shattering of freezing drops may explain the observation of SIP in a greater temperature range. Such an explanation is consistent with the laboratory observation of the frequency of droplet shattering by Takahashi and Yamashita (1970), Takahashi (1975) and Lauber et al. (2018).

#### 8.2 Splintering during riming and HM mechanism

     As discussed in section 4, some SIP cloud regions comprised both liquid droplets and graupel and therefore, they formally satisfy conditions for the HM process (i.e. **Figs.6** and **14**). However, in a number of SIP cases, graupel was not observed (i.e. **Figs.7,15** and **17**). Whereas in cases like in **Figs.9** and **10** graupel is present, but LWC is very low or absent. Hence, such cases did not meet the formal conditions for the HM process.

These inconsistencies of the environmental conditions imply the existence of another SIP mechanism, that does not involve graupel. One of such mechanisms could be splintering during riming (Ono, 1971; Choularton et al. 1978; Mossop, 1980). After sticking to an ice surface, some drops during freezing may form an ice shell around a liquid core and rupture, ejecting splinters. Such a scenario is supported by the observation in SIP regions of both liquid droplets and rimed ice.

However, Macklin and Payne (1969) and Dong and Hallett (1989) showed that droplets spread out after hitting an ice surface at temperatures warmer than -3°C. Therefore, an ice shell does not form, and it limits the riming-splintering mechanism at the high temperature end. On the other hand, Griggs and Choularton (1983) argued that the ice shell might be too strong to break from internal pressure at temperatures $T_a$ <-9°C. So, these laboratory studies suggest that the temperature range of the splintering during riming remains approximately the same as for the HM-

process.

Unfortunately, in the framework of this study, it is not possible to segregate droplet shattering, rime splintering, and HM mechanisms and assess their occurrences.

### 8.3 Fragmentation due to ice-ice collisions

Takahashi (1993) argued that a collision between large graupel grown by riming and small graupel grown by deposition (or a rimed snowflake) results in SIP. In laboratory experiments, Takahashi et al. (1995) found that collision between large and small graupel might be an efficient source of secondary ice particles.

Formally, the condition for presence of graupel and rimed ice particles is satisfied in the cases shown in **Figs.6,7,9,10,14,17**. Therefore, formation of the small faceted ice particles in theses cases can be attributed to the collision-fragmentation mechanism.

However, analysis of the CPI imagery in ice clouds lacking graupel and far away from any sources of liquid or updrafts did not reveal any noticeable presence of small faceted ice crystals. This observation suggests that the collision-fragmentation mechanism most likely has low significance for SIP for the cases of deposition grown ice crystals in pure ice clouds. Another possible explanation for the absence of evidence of the collision-fragmentation SIP is that the ice fragments formed due to ice-ice collision do not regrow into small faceted ice particles. In cases like that, the employed method cannot be used for the identification of secondary ice formed due to this mechanism.

So, in the frame of the obtained observations, the contribution of the collision-fragmentation mechanism to SIP remains uncertain.

### 8.4 Ice fragmentation during thermal shock

Laboratory studies by Dye and Hobbs (1968) and Hobbs and Farber (1972) yielded positive results on the fragmentation of ice particles due to thermal shock caused by a droplet freezing on the surface of an ice particle. This mechanism is expected to be active at $T_a$ <-5°C (King and Fletcher, 1976ab). Since a large fraction of our observations of SIP can be related to originating temperatures $T_a$ >-5°C, it is expected that the thermal shock mechanism has low importance for this study. However, for lower temperatures, the role of this mechanism in SIP remains uncertain.

### 8.5 Ice fragmentation during sublimation

A cloud environment subsaturated with respect to ice is a necessary condition for initiating the mechanism of ice fragmentation during sublimation. As it was discussed in section 4, most of the SIP events were observed in mixed-phase clouds. Such clouds are supersaturated with respect to ice, and therefore, the necessary condition is not satisfied. Hence, the fragmentation during sublimation mechanism can be ruled out.

### 8.6 INP activation in transient supersaturation around freezing drops

Maximum supersaturation formed around a freezing droplet with $D$ =200μm at $T_a$=-4°C is estimated as $S_w$ =1% (Nix and Fukuta, 1974). Such supersaturation can also be achieved in moderate vertical updrafts (e.g. $u_z$ =4m/s, $N_{dr}$ =50cm⁻³ and $D$ =30μm), which are typical for convective regions in MCS (e.g. **Fig.5**). Therefore, if activation of INPs around freezing drops has any significance at $T_a$ >-4C, it should be observed in the bulk of convective updrafts, since the total volume with $S_w$~1% is much higher there compared to that around a freezing drop. However, many MCS regions (not shown here) with vertical updrafts exceeding 4m/s lacked notable concentrations of small ice particles at temperatures close to -4°C. Therefore, the mechanism of INP nucleation in transient supersaturation

around freezing drops is unlikely to be responsible for the observed concentration of small ice observed in this study at $T_a$ >-4°C. However, this mechanism may be active at lower temperatures.


### 9.    Effect of the melting layer

One of the most striking findings of this study is the persistent observation of SIP immediately above the melting layer. This phenomenon was observed in clouds in different geographical regions and clouds with different dynamics. So, the question arises: what are the conditions that make the cloud environment above the melting layer favorable for

SIP?

One possible explanation is the formation of large drops ($D$~60-300μm) due to the recirculation of ice and liquid through the melting layer. Thus, ice particles turn into drops after falling through the melting layer. Then these drops are brought back above the melting layer by convective or turbulent updrafts.

The recirculation hypothesis is supported by the observation of distortion of the bright band altitude in the

convective cloud regions. An example of such distortion is presented in **Fig.23**. **Figure 23** shows a zoomed segment of the time series in **Fig.5**, which includes reflectivity (c) and Doppler velocity (d) measured by onboard X-band radar when traversing a convective cell in the tropical MCS (09:40-09:45). Comparison of panels (b) and (c) in **Fig.23** shows a peak-to-peak correlation between the vertical wind velocity and elevation of the bright band in the convective cell. In a few points the bright band moves up to ~600-700m above the level of the bright band in

undisturbed cloud regions (indicated by dashed line **Fig.23c,d**). Such distortion of the bright band is explained by moving melted drops by vertical updrafts to higher levels. A spatial coincidence of the SIP area (**Fig.23a**), convective updraft (**Fig.23b**) and the region with the elevated bright band (**Fig.23c**) is supportive of the droplet recirculation hypothesis.

In order for a drop to ascend through the melting layer, the velocity of the updraft ($u_z$) should exceed the drop fall

velocity ($u_{fall}$). **Figs.5f** and **13f** show examples, when the vertical velocity above the melting layer in the tropical MCS reached $u_z$ ≈8m/s and in frontal clouds $u_z$ ≈3m/s, respectively. Such updraft velocity is sufficient to move drops with $D$=100-200μm ($u_{fall}$ =0.3-1m/s at $P$=500mb) through the melting layer (Δ$Z$=500m) during a reasonable time of a few tens of seconds to a few minutes.

The vertical travel distance of the liquid drops formed in the melting layer depends on the sustainability and

endurance of the convective updraft, its vertical velocity, and droplet size. Smaller droplets have higher chances to travel deeper in the cloud compared to large ones. This is consistent with the observation of occurrence of droplets with $D$=80μm and 100μm as shown in **Figs.5b** and **8b**, which were measured in the same MCS at two different altitudes 5600m and 7000m, respectively. Rapid decrease of the concentration of large drops with temperature (and therefore altitude) in tropical MCSs is also seen from **Fig.11**.

Another explanation of the formation of drizzle size drops is related to the collision-coalescence process. However, the observed LWC and number concentration of cloud droplets with $D$<40μm in a mature tropical MCS during HIWC typically varied in the ranges 0.01<$LWC$<0.1g/m³ and 5<$N_{dr}$<40cm⁻³, respectively, and were always associated with a mixed phase dominated by ice 0.5<$IWC$<3g/m³ (e.g. **Figs.5dg** and **8dg**). High $IWC$ and low $N_{dr}$ and $LWC$ will hinder the collision-coalescence process due to riming and WBF processes, which result in depletion of

droplets. However, the collision-coalescence process cannot be ruled out in midlattitude frontal clouds as in **Fig.13**.

After arriving in the supercooled environment above the melting layer, drops collide with aged ice particles and some of these drops may form ice shells during freezing and shatter. This may result in initiation of SIP. Images of large drops frozen on the surface of aged ice particles observed above the melting layer are shown in **Fig.24**. Most of the drops have deformed shapes with bulges. Formation of bulges may be accompanied by production of ice splinters by jetting or bubble bursting (Lauber et al. 2018).

In laboratory studies, Takahashi (1975) and Lauber et al., (2018) concluded that large drops have higher occurrence of shattering compared to small ones. Therefore, despite of their lower concentration, shattering of fewer large drops may play the role of a trigger in initiating SIP. As follows from Tables 1 and 2, the concentration of small ice particles has the highest correlation with the droplets from the size range 40-60μm. Therefore, it is expected that the droplets from this size range have the highest contribution to SIP through maintenance of a chain reaction as shown in **Fig.25a**. The conceptual model summarizing the effect of the melting layer of SIP in presented in **Fig.25b**.

## 10. Conclusions

In the frame of this study, we explored the microphysics of SIP cloud regions in tropical MCSs at the mature stage of their development and mid-latitude frontal cloud systems within the temperature range -15°C< $T_a$ <0°C. SIP cloud regions were identified based on the presence of numerous small faceted ice crystals with $L_{max}$ <100μm. The concentration of such small crystals peaked at 500-1000 L$^{-1}$. Such particles cannot be a result of the recirculation of pre-existing aged ice. Based on the estimate that the age of such small crystals is limited by $\tau_{cr}$~60-120s, it was deduced that such ice crystals are still associated with the environment of their origin. This approximation was employed to assess the environmental conditions associated with SIP. As discussed below, our method has a number of limitations. However, it allowed obtaining the following conclusions:

1) Most SIP cases were associated with:
   (a) presence of liquid droplets in the SIP region or somewhere in the vicinity
   (b) convective updrafts or regions of enhanced turbulence
   (c) aged rimed ice particles
2) The highest correlation between the concentration of small faceted ice crystals and liquid droplets was found for droplets in the range 40μm< $D$ <60μm (Table 1 and 2).
3) In several cases, no liquid was observed in SIP cloud regions.
4) Graupel was not always present in the SIP cloud regions.
5) The shape of small faceted ice particles suggests that they were grown in conditions supersaturated with respect to ice, but subsaturated with respect to water.
6) The smallest size of the splinters generated during SIP were estimated at 10μm or less.
7) The aspect ratio of small hexagonal ice particles observed in the same volume may vary up to ten times.
8) Both in tropical MCSs and mid-latitude frontal clouds, secondary ice particles were observed immediately above the melting layer starting at $T_a$ <-0.5°C. In MCSs SIP was observed at temperatures down to -15°C. No data points were available below this temperature.
9) In MCSs, SIP regions vertically correlate with the locations of the coldest tops. No such dependence was found for the frontal cloud systems we analyzed.

We hypothesize that the initiation of SIP above the melting layer is related to the circulation of liquid drops through the melting layer. Liquid drops formed via melting ice particles are advected by the convective updrafts above the melting layer, where they collide with aged ice, freeze and shatter. The ice splinters generated by shattering initialize the chain reaction of SIP.

In many cases, concentrations of frozen drops and their fragments exceeding expected concentrations of INPs by orders of magnitude were observed in SIP regions. This discrepancy implies that something other than heterogenous drop freezing must be contributing to SIP. The roles of mechanisms such as HM rime-splintering, ice-ice collisional breakup, thermal shock fragmentation, and INP activation around freezing drops cannot be confidently linked to SIP based on the collected data, for reasons explained at length. Thus, we conclude by process of elimination that the mechanism of droplet shattering during freezing is very likely a critical contributing factor to SIP in these cases.

The conclusions obtained in this study are based on the interpretation of observations which were obtained along needle-like penetrations of large cloud systems at some time of their evolution. The fact that initial and boundary conditions of the studied cloud systems are poorly known, and the trajectories of cloud volumes and cloud particles are not identifiable, brings a certain ambiguity into the interpretation of the obtained observations. So, in many ways, the conclusions in this work bear a qualitative character, and the emphasis of this study is on the observational part. The obtained results are expected to contribute in our understanding of SIP, and they may be used by cloud modeling studies for evaluation of secondary ice production in the numerical simulations of clouds (e.g. Qu et al. 2018), for instance by evaluating where such small particles appear in high concentrations in simulations.

In microphysics schemes that predict the number concentration of ice crystals, i.e. spectral (bin) and multi-moment bulk schemes (e.g. Khain et al. 2004; Milbrandt and Yau 2005), SIP is most commonly modeled exclusively with a simple parameterization of the HM-process. If riming of graupel is occurring in the temperature range between -3 and -8C, an ice splinter production rate is computed for this process, with a maximum at -5C, decreasing linearly to zero at the ends of the temperature range. Assumptions of the crystal number concentration tendency and the size of the new crystals are made, based broadly on the published results of Hallet and Mossop (1974). Parameterizations that exist for other mechanisms of secondary ice production have been less widely included in modeling efforts to explain apparent SIP in observed cloud systems, but when INP are treated rigorously in a prognostic manner, such mechanisms are generally found to be too weak to explain observed ice even when considered additively, including drop shattering and ice-ice collisions (e.g., Fridlind et al. 2007; Fu et al. 2019). It is perhaps unsurprising that such additional mechanisms are not more widely adopted if they provide only weak ice generation and still unsatisfactory results compared with observations, in addition to being highly uncertain owing to a paucity of robust laboratory data. Ultimately, it may be important in atmospheric models for some purposes to improve the representation of both primary and secondary ice production in microphysics parameterization schemes based on more recent observations and the hypothesized processes. It will be a topic of future research to apply the observations presented to develop new parameterizations of SIP. However, parameterizations based on field observations will necessarily remain to some degree speculative without a strong foundation of laboratory measurements that can provide clear and repeatable evidence of specific mechanism strengths.

The obtained results bring up a more general question about the limitations of airborne techniques in the identification of major mechanisms and their efficiencies in SIP. Airborne observations deal mostly with the results of

SIP in the form of different stages of aged secondary ice. However, attempts to quantify or parameterize the secondary ice production from in-situ observations are limited because the initial and boundary conditions are mostly unknown. One of the fundamental limitations of airborne techniques is that they do not allow for monitoring and identifying the process of secondary ice directly. In this regard, the pursuit of SIP research lends itself well to laboratory experiments and should be emphasized in this area.

*Data availability.* The data used in this study are available upon request from the first author.

*Authors' contribution.* A. Korolev led the collection of the cloud microphysical data and data analysis. I. Heckman performed analysis of the CPI and cloud microphysical data. M. Wolde carried out the airborne data collection, and analysis of in-situ atmospheric state parameters and NAWX radar data. L. Ladino supported the analysis of cloud microphysical data. E. Williams supported the airborne data collection during BAIRS2. A. Korolev prepared the manuscript with the co-authors contribution from A. Ackerman, A. Fridlind, P. Lawson, J. Milbrandt.

*Competing interests.* The authors declare that they have no conflict of interests.

*Acknowledgements.* The HIWC program was supported by Environment and Climate Change Canada (ECCC), National Research Council (NRC), Transport Canada (TC) and the Federal Aviation Administration (FAA). BAIRS2 program was supported by FAA. WERVEX program was supported by ECCC and TC. The NRC RAIR program contributed funding for the BAIRS2/WERVEX campaign.Participation of the Massachusetts Institute of Technology in BAIRS2 was supported by FAA under Air Force Contract FA8702-15-D-0001. Any opinions, findings, conclusions, or recommendations expressed in this material are those of the authors and do not necessarily reflect the views of the FAA and TC. Special thanks to the NRC FRL pilots Anthony Brown, Paul Kissman, and Rob Erdos for their outstanding cooperation during in-situ data collection during the HIWC and BAIRS2/WERVEX projects. The authors are grateful for the technical support provided during the data collection period by ECCC and NRC technical and engineering teams. Authors acknowledge Ed Emery, Chris Lynch, and Quentin Schwinn NASA Glenn Research Center for supporting wind tunnel tests of CPI probe. CFD analysis of the flow around CPI was performed by Kirk Creelman Auriga Design Inc. We appreciate the help of University of Waterloo student Keegan Cove who supported CPI data analysis. The authors grateful to Charlie Knight and anonymous reviewers for their valuable comments.

**Appendix A**

**Effect of ice particle shattering on CPI measurements**

A set of tests in the Cox and Co. wind tunnel facility (Plainview, NY) was conducted to identify the performance of different airborne instruments in ice sprays. The primary objective of these tests was to identify and document the effect of shattering and bouncing on the measurements of airborne particle probes with different types of tips and inlets. More detail about the nature of this study can be found in Korolev et al. (2013b).

**Figure A1** shows two snapshots from a high-speed video of the CPI inlet in an ice spray at an air speed of 80m/s. The CPI sampling tube has diameter 2.5mm with a rounded edge having a radius of curvature of approximately 0.5mm. The purpose of such sharpened edge is to mitigate the effect of shattering. However, as it is seen from **Figure**

**A1**, despite of its relative sharpness, ice particles still shatter and rebound from the edge of the CPI inlet. **Figure A1** also shows that the rebound particles are deflected both outside and inside the CPI sampling tube. This observation led to the conclusion that the CPI measurements can be affected by mechanical shattering of ice particles on impact with the CPI inlet.

**Figure A2** presents results of the CFD simulations of the airflow around the CPI housing. The simulation was conducted for the airspeed 150m/s, $P$ =450mb, $T_a$ =-40C. As it is seen from **Figure A2c,d** the velocity of the air changes by approximately 30m/s at the distance of ~2cm when passing through the front part of the inlet tube. This will result in large aerodynamic stresses, which ice particles may experience when entering the CPI inlet. Another area where ice particles may experience strong aerodynamic stresses is located near the walls of the inlet tube (**Figure**

**A2b**). Such aerodynamic stresses may result in deformation of the shape of liquid drops and fragmentation of large fragile ice particles and aggregates with weak bonding.

    It is worth noting that the CPI used in this study had a modified shortened inlet tube. The original CPI front inlet tube is longer, and due to the inner step at the front edge, it has a higher velocity jump at the entrance compared to that in **Figure A2d**.

**Figure A3** shows examples of CPI images of fragmented ice particles sampled in clouds. The image frame in **Figure A3a** includes 55 fragments, which corresponds to a local concentration of approximately $6\times10^3$cm$^{-3}$ to $7\times10^3$cm$^{-3}$. Such concentrations of ice particles do not seem to be possible in natural clouds. The only reasonable explanation is that these fragments result from ice particle shattering due to mechanical impact with the CPI inlet, and on immediately after shattering the fragments form a spatially dense cluster of particles with high local concentration.

The cluster of multiple images shown in **Figure A3b** is unlikely to occur in clouds due to significantly different fall velocities, which range from approximately 1cm/s (for the smallest particle in the image frame) to 1m/s (for the largest particle). Most likely the images in **Figure A3b** are debris from a shattered ice particle originated from impact with the CPI inlet.

    The origin of fragmentation of the particle in **Figure A3c,d** is most likely related to fragmentation due to

910 aerodynamic stresses. If such fragmentation occurs due to some natural causes, the fragments due to their different sizes are unlikely to stay together due to different fall velocities.

    In the present study, CPI images similar to those in **Figure A4** were identified as shattered artifacts. The shapes of most of these particles conflict with the concept of growth of crystal lattice. However, their shapes can be explained by the fragmentation of ice crystals.

Images as in **Figure A4** usually form spatial clusters with close spacing, and they appear in CPI image frames (2.3 mm ×2.3mm) as multiple images as in **Figure A3**. In this regard, the number of images in CPI image frames was used as an indicator of shattering. In this work, CPI image frames with more than one image were identified as shattering artifacts, and such frames were excluded from the analysis. The SPEC CPIview processing software was modified to recognize such image frames and discard them. Shattered fragments, which appear in the CPI imagery as

single particle images (i.e. the rest fragments did not pass through the sample volume), could not be identified by this technique. However, since the entire analysis of the CPI data was built on identification and calculation of concentrations of small hexagonal prisms with $L < L_{max}$ and droplets with $D$<300μm, the unidentified shattered ice fragments in the CPI imagery did not affect outcomes of this study.

It should be noted that some of the images as in **Figure A4** may have a natural origin. However, their exclusion from the analysis does not affect the conclusions obtained in this study.

The analysis of the CPI data showed that the number of shattering artifacts increases with the increase of particle size. Misalignment between the direction of local airflow and the axis of the CPI sampling tube also results in an increase of the shattering artifacts and a decrease of the counting rate of intact particles. Thus, for a 4° angle between the airflow and axes of the sampling tube, the CPI sampling volume will be in the geometrical shadow. This will result in a reduction of the counting rate of primarily large particles. Smaller particles will follow the airflow, and their counting rate will be less affected.

The orientation of the CPI sampling tube was aligned with the local flow at $H$ =3km and $TAS$ =100m/s at the mounting location on the Convair-580. For other flight conditions, the misalignment between the local airflow and the axis CPI inlet tube will persist.

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

**Figure Captions**

**Figure 1.** Conceptual diagram of the transport of secondary ice production particles in a cloud after its formation.

**Figure 2.** Calculated ice column growth at vapor saturation over water at -3C, -5C, -8C. Triangles, circles, and squares are laboratory observations by Fukuta and Takahashi (1999).

**Figure3.** (a) Examples of CPI images used for neural net training to identify small faceted ice crystals. These ice crystals were collected in the mesoscale convective clouds at altitudes 6200<$H$<7000 and temperature range -10C<T<-3C. (b) Examples of images misidentified by the image recognition software as pristine faceted ice. The numbers below each image frame indicate maximum size of the images in μm.

**Figure 4.** GOES-13 infrared image of the MCS with the Convair-580 track (courtesy Pat Minnis) corresponding to time segments shown in **Figs.5** and **8**. Circles indicate the cloud regions along the flight track where SIP was identified (see **Figs.5**). The marked regions also coincide with convective cloud regions (see text).

**Figure 5.** Time series of microphysical parameters collected in oceanic MCS offshore French Guiana on 15 May 2015. (a) CPI count rate of small pristine ice with $D_{max}$<60μm, 100μm; (b) CPI count rate of cloud droplets with D>40μm, 60μm, 80μm,100μm; (c) concentration of cloud particles D>40mm measured by 2DS; (d) concentration of cloud droplets measured by FSSP and CDP; (e) Rosemount Icing Detector frequency; (f) vertical velocity measured by AIMMS20 and Doppler velocity calculated from W-band radar; (g) IWC calculated from 2DS+PIP; (h) air temperature. Grey strips indicate cloud regions with enhanced concentration of small faceted ice particles; red and yellow strips indicate regions where ice and liquid were present, but no SIP was observed (see text). The altitude of measurements varied between 5600m and 5700m.

**Figure 6.** Spatial sequence of CPI images of (a) droplets and faceted ice crystals and (b) aged large ice particles. Blue frames indicate frozen droplets with modified shapes, and red frames fragments of shattered frozen drops. Numbers under each image indicate their maximum sizes $L_{max}$. Cloud particles in (a) and (b) are spatially mixed, and they were split between two panels because of their difference in size. The images were sampled at $T_a$ =-5C and $H$ =5650m during UTC 09:40:42 – 09:40:47 on 15 May 2015 during measurements shown in **Figs.5.**

**Figure 7.** (a) Spatial sequence of CPI images; (b) Subset of droplets and faceted ice crystals from panel (a). Numbers under each image indicate their maximum sizes $L_{max}$. The images were sampled at $T_a$ =-5C and $H$ =5620m during UTC 09:46:36 – 09:46:39 on 15 May 2015 during measurements shown in **Figs.5**. (a) Purple frames indicate images of ice particles with evidence for their vertical circulation in the storm.

**Figure 8.** Same as in **Fig.5**. The altitude of measurements varied between 7000m and 7300m.

**Figure 9.** (a) Spatial sequence of CPI images; (b) Subset of droplets and faceted ice crystals from panel (a). (b) Blue frames indicate frozen droplets with modified shapes, and green frames frozen drops with developed facets. Numbers under each image indicate their maximum sizes $L_{max}$. The images were sampled at $T_a$ =-14C and $H$ =7200m during UTC 12:05:27 – 12:05:38 on 15 May 2015 during measurements shown in **Figs.8**.

**Figure 10.** (a) Spatial sequence of CPI images; (b) Subset of droplets and faceted ice crystals from panel (a). Numbers under each image indicate their maximum sizes $L_{max}$. No liquid droplets were present in this cloud region. The images were sampled $T_a$ =-14C and $H$ =7200m during UTC 12:05:47 – 12:05:53 on 15 May 2015 during measurements shown in **Fig.8.**

**Figure 11.** Average concentration of small faceted ice crystals (a) and drops (b) estimated from CPI measurements. The concentration was averaged over the entire flight length sampled during 13 flights in 10 tropical MCSs. The concentration was normalized on the sampling distance in each 1C temperature interval. Total number of 1s average samples $8.4\times10^4$, total in-cloud length 9580km.

**Figure 12.** Flight track of the Convair-580 in the frontal cloud system on 24 March 2017 overplayed over (a) GOES-16 infrared image (download from Univ. Wisconsin); (b) KBUF (Buffalo, NY) NEXRAD reflectivity at elevation 0.46°. Dashed line circles indicate SIP cloud regions.

**Figure 13.** Time series of cloud microphysical parameters collected in a frontal cloud system over upstate NY on 24 March 2017. (a) CPI count rate of small pristine ice with $D_{max}$<60µm, 100µm; (b) CPI count rate of cloud droplets with $D$>40µm, 60µm, 80µm,100µm; (c) concentration of cloud particles D>40mm measured by 2DS; (d) concentration of cloud droplets measured by FSSP and CDP; (e) Rosemount Icing Detector frequency; (f) vertical velocity measured by AIMMS20 and Rosemount 858 probes; (g) IWC calculated from composite 2DS and PIP PSDs; (h) air temperature. Grey strips indicate cloud regions with enhanced concentration of small faceted ice particles; red and yellow strips indicate regions where ice and liquid were present, but no SIP was observed (see text).

**Figure 14.** Spatial sequence of CPI images of (a) droplets and faceted ice crystals and (b) background large ice particles. (a) Blue frames indicate frozen droplets with modified shapes, green frames - frozen drops with developed facets, red frames - fragments of shattered drops. Numbers under each image indicate their maximum sizes $L_{max}$. Cloud particles in (a) and (b) are spatially mixed, and they were split between two panels because of their difference in sizes. The images were sampled at $T_a$ =-2C and $H$ =3500m during UTC 12:29:20 – 12:30:00 on 24 March 2017 during measurements shown in **Figs.13**.

**Figure 15.** Spatial sequence of CPI images of droplets and faceted ice crystals. Blue frames indicate frozen droplets with modified shapes, green frames - frozen drops with developed facets, red frames - fragments of shattered drops. Numbers under each image indicate their maximum size $D_{max}$. The images were sampled during UTC 14:06:30-14:07:30 on 24 March 2017 (not shown in **Figs.13**), $T_a$ =-3C, $H$ =2100m.

**Figure 16.** Average concentration of ice crystals (a) and drops (b) estimated from CPI measurements and normalized on the sampling distance in each temperature interval. The data were collected during two flights in mid-latitude frontal cloud systems with temperatures -10°C<$T_a$ <-0°C. Total number of 1s average samples $1.4\times10^4$, total in-cloud aircraft path length 1380 km

**Figure 17.** Spatial sequence of CPI images of (a) droplets and faceted ice crystals and (b) background large ice particles. (a) Blue frames indicate frozen droplets with modified shapes, green frames - frozen drops with developed facets, red frames - secondary ice particles developed into thin hexagonal plates. Numbers under each image indicate

their maximum size $D_{max}$. Cloud particles in (a) and (b) are spatially mixed, and they were split between two panels because of their difference in sizes. The images were sampled during UTC 04:59:50-05:00:18, on 24 January 2017. $T$ = -1.5C, $H$ =2400m.

**Figure 18.** Images of fragmented frozen droplets collected in the SIP cloud regions indicated by grey areas in **Figs.5** and **13** at -5C< $T$ <-1C

**Figure 19.** Images of small faceted ice particles, which were sampled in SIP cloud regions at -5.5C< $T_a$ <-5C, $H$ =5600m indicated by grey color in **Fig.5**. The aspect ratio of the small hexagonal prisms varies in the range 0.3<R<6

**Figure 20.** CPI images of single frozen droplets, where shape was modified during freezing collected in SIP cloud regions in the temperature range -5 C< $T$ <-1C.

**Figure 21.** Images of frozen droplets partially regrown into faceted ice crystals collected at -5C< $T$ <-1C.

**Figure 22.** Different scenarios of evolution of SIP particles after their production.

**Figure 23.** Zoomed time segments of the time series in **Fig.5** with the counting rate of small pristine ice particles (a),
vertical velocity (b), X-band radar reflectivity (c) and Doppler velocity (d), measured during a traverse of the convective region inside tropical MCS. Horizontal dashed lines in (c,d) show the level of the bright band in undisturbed by convective updrafts cloud regions. Two vertical solid lines indicate SIP cloud region, which spatially coincides with the convective cell (b) and elevated bright band (c).

**Figure 24.** Images of frozen droplets attached to ice crystals that initiated their freezing. The shape of the frozen
droplets was modified during freezing. Images were collected in the temperature range in the temperature range -15C< $T$ <-1C.

**Figure 25.** (a) Conceptual model of secondary ice production due to shattering of freezing drops. (b) Conceptual model of the effect of melting layer on the secondary ice particle formation in MCSs and frontal clouds.

**Figure A1.** Snapshots from a high-speed video of trajectories of shattered and rebound ice particle fragments formed
on impact with the CPI inlet. The measurements were conducted in the Cox and Co. wind tunnel facility (Long Island, NY, USA) in ice spray at TAS=80m/s.

**Figure A2.** Results of the CFD analysis of flow around and through the CPI sampling tube. (a) airspeed around the CPI sensor head; (b) cross-section of speed inside the CPI inlet tube at the location of the sample volume; (c) zoomed CPI inlet area as in (a); (d) changes of the air velocity along the CPI inlet tube centerline. The simulation was
performed for P =450mb, $T_a$ =-40C, TAS =150m/s

**Figure A3.** Multiple images registered in 2.3 mm ×2.3mm CPI image frames (a,b,c). Images in (a,b) are identified as a result of shattering due to mechanical impact with the CPI inlet. Images in (c,d) are likely result from fragmentation due to aerodynamic stresses in the CPI inlet tube.

**Figure A4**. Examples of CPI images identified as shattering artifacts. Such images were excluded from analysis.





**Table 1**. Correlation coefficient between droplet concentration in different size ranges and concentration of small faceted ice crystals with $D_{\mathrm{max}}$ <100μm for the cloud segment in **Fig.5** for 30 s and 60 s averaging.

| Dropl.Conc. | $D$>20μm | $D$>40μm | $D$>60μm | $D$>80μm | $D$>100μm |
|---|---|---|---|---|---|
| Corr.Coeff. (30s) | 0.48 | 0.66 | 0.85 | 0.77 | 0.69 |
| Corr.Coeff. (60s) | 0.56 | 0.71 | 0.9 | 0.85 | 0.8 |





**Table 2**. Correlation coefficient in different size ranges between droplet concentration and concentration of small faceted ice crystals with $L_{max} < 100\mu m$ for the cloud segment in **Fig.13** with 30s and 60s averaging.

| Dropl.Conc. | $D{>}20\mu m$ | $D{>}40\mu m$ | $D{>}60\mu m$ | $D{>}80\mu m$ | $D{>}100\mu m$ |
|---|---|---|---|---|---|
| Corr.Coeff. (30s) | 0.44 | 0.51 | 0.48 | 0.26 | 0.11 |
| Corr.Coeff. (60s) | 0.65 | 0.71 | 0.59 | 0.29 | 0.18 |


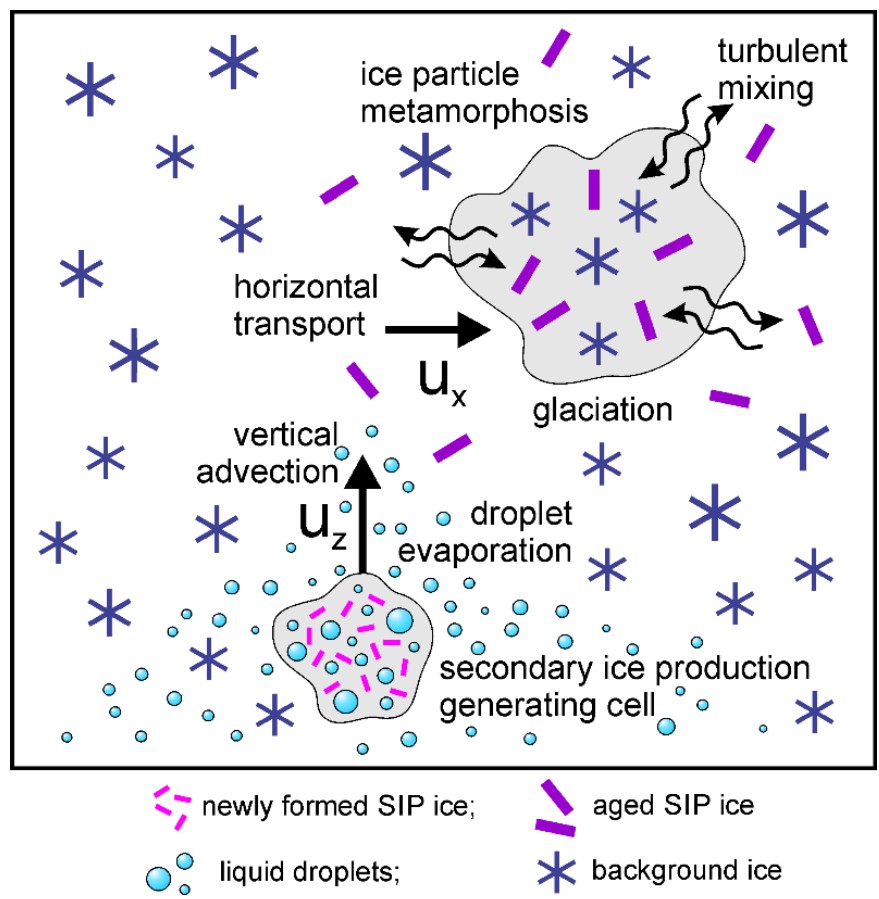

**Figure 1**. Conceptual diagram of the transport of secondary ice production particles in a cloud after its formation.

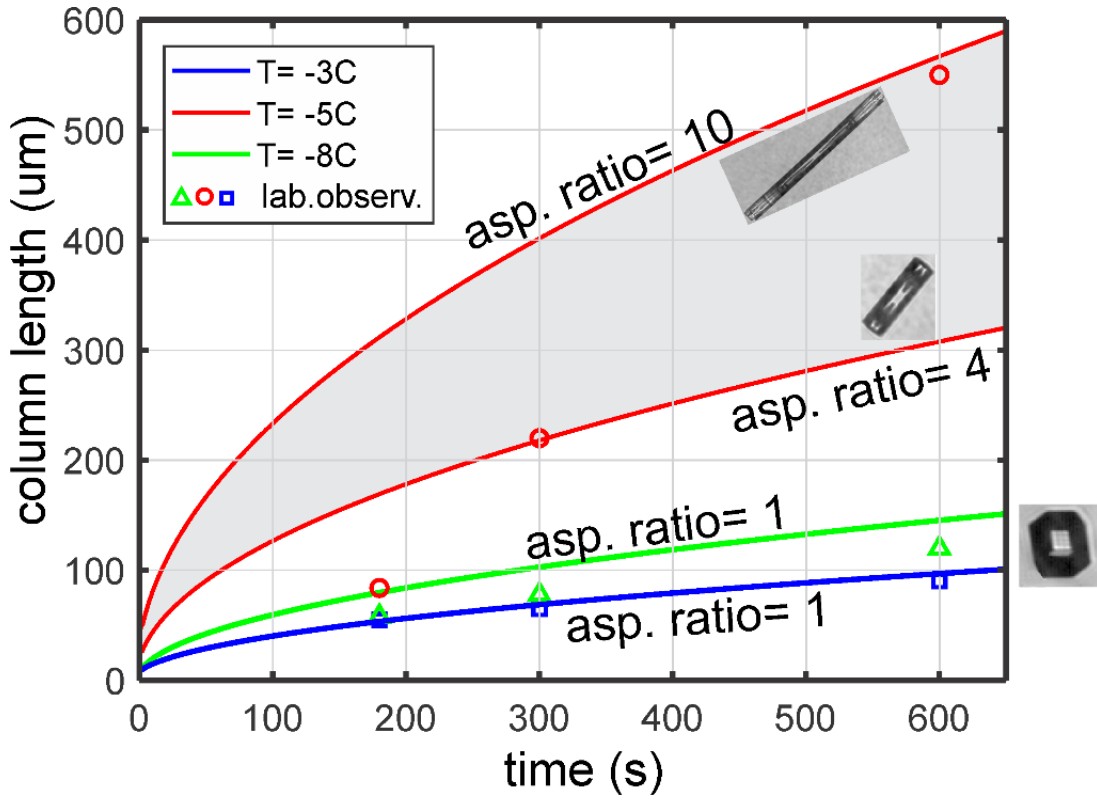

**Figure 2**. Calculated ice column growth at vapor saturation over water at -3C, -5C, -8C. Triangles, circles and squares are laboratory observations by Fukuta and Takahashi (1999).


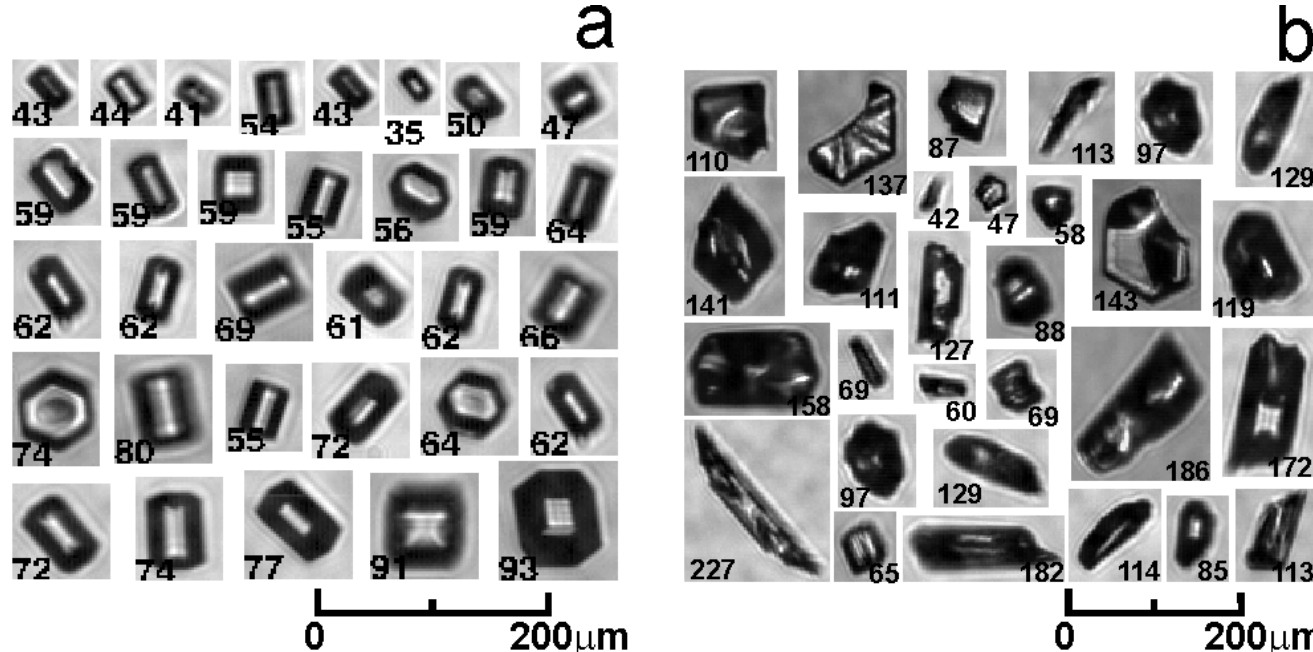

**Figure 3.** (a) Examples of CPI images used for neural net training to identify small faceted ice crystals. These ice crystals were collected in the mesoscale convective clouds at altitudes 6200<*H*<7000 and temperature range -
10C<T<-3C. (b) Examples of images misidentified by the image recognition software as pristine faceted ice. The numbers below each image frame indicate maximum size of the images in μm.

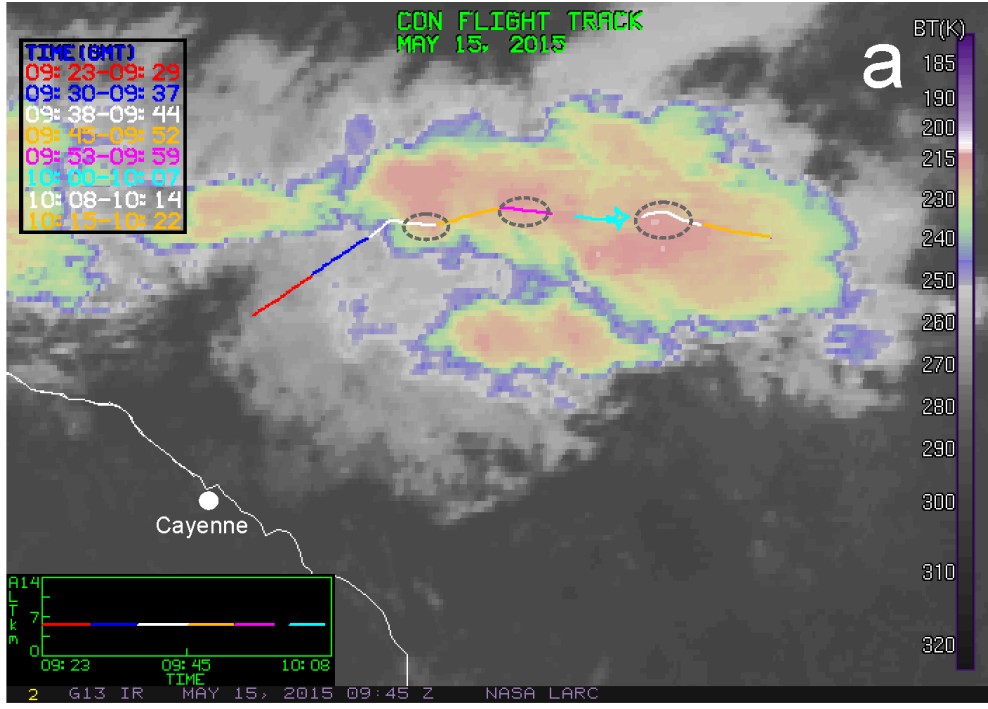

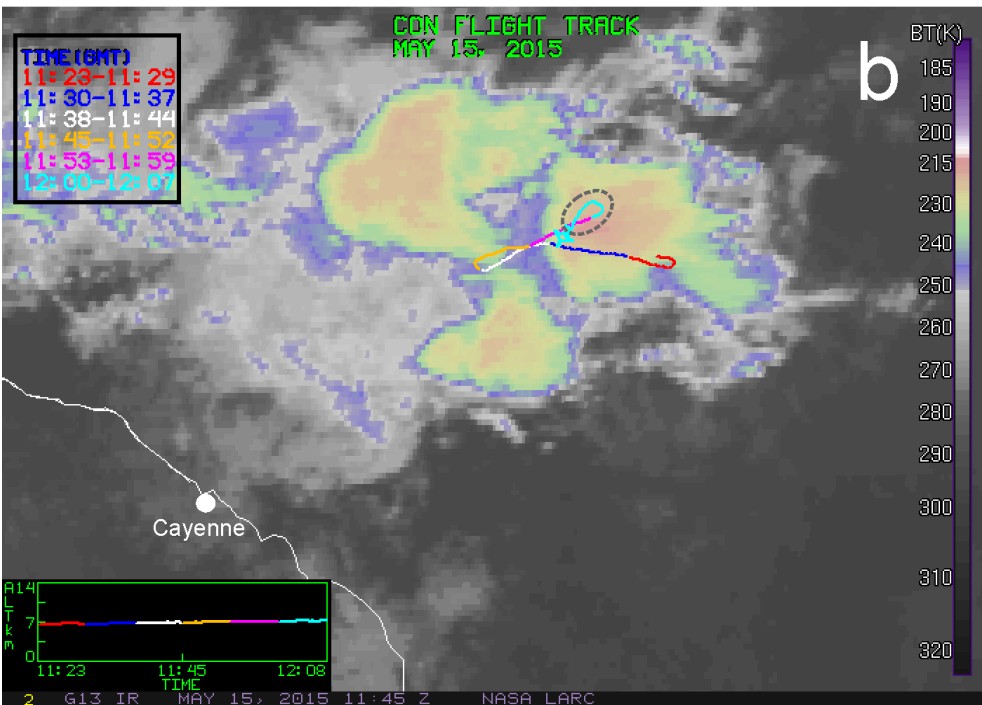


**Figure 4.** GOES-13 infrared image of the MCS with the Convair-580 track (courtesy Pat Minnis) corresponding to time segments shown in **Figs.5 and 8**. Circles indicate the cloud regions along the flight track where SIP was identified (see **Fig.5**). The marked regions also coincide with convective cloud regions (see text).

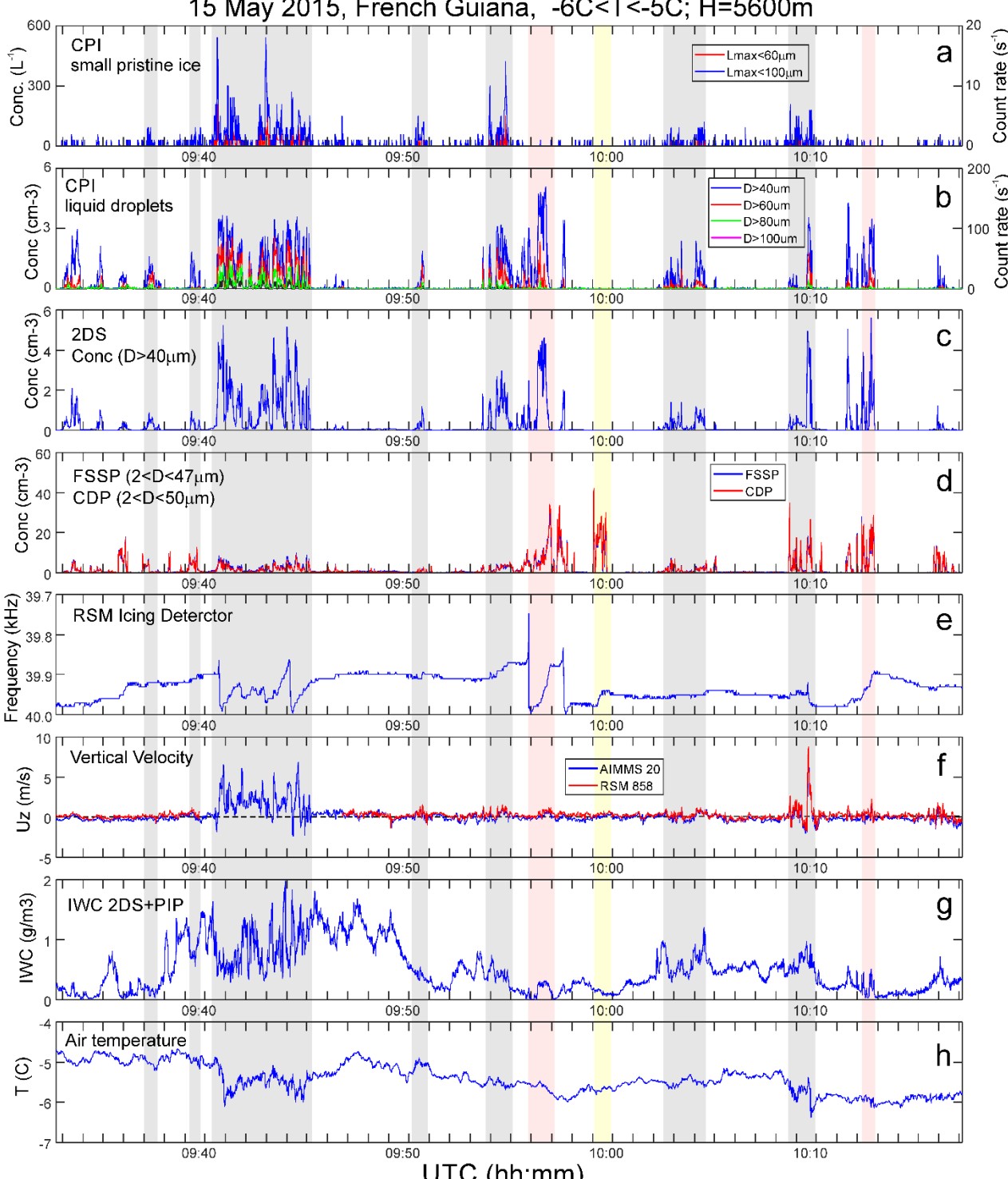

**Figure 5**. Time series of microphysical parameters collected in oceanic MCS offshore French Guiana on 15 May 2015. (a) CPI count rate of small pristine ice with $D_{max}$<60μm, 100μm; (b) CPI count rate of cloud droplets with D>40μm, 60μm, 80μm,100μm; (c) concentration of cloud particles D>40mm measured by 2DS; (d) concentration of cloud droplets measured by FSSP and CDP; (e) Rosemount Icing Detector frequency; (f) vertical velocity measured by AIMMS20 and Doppler velocity calculated from W-band radar; (g) IWC calculated from 2DS+PIP; (h) air temperature. Grey strips indicate cloud regions with enhanced concentration of small faceted ice particles; red and yellow strips indicate regions where ice and liquid were present, but no SIP was observed (see text). The altitude of measurements varied between 5600m and 5700m.

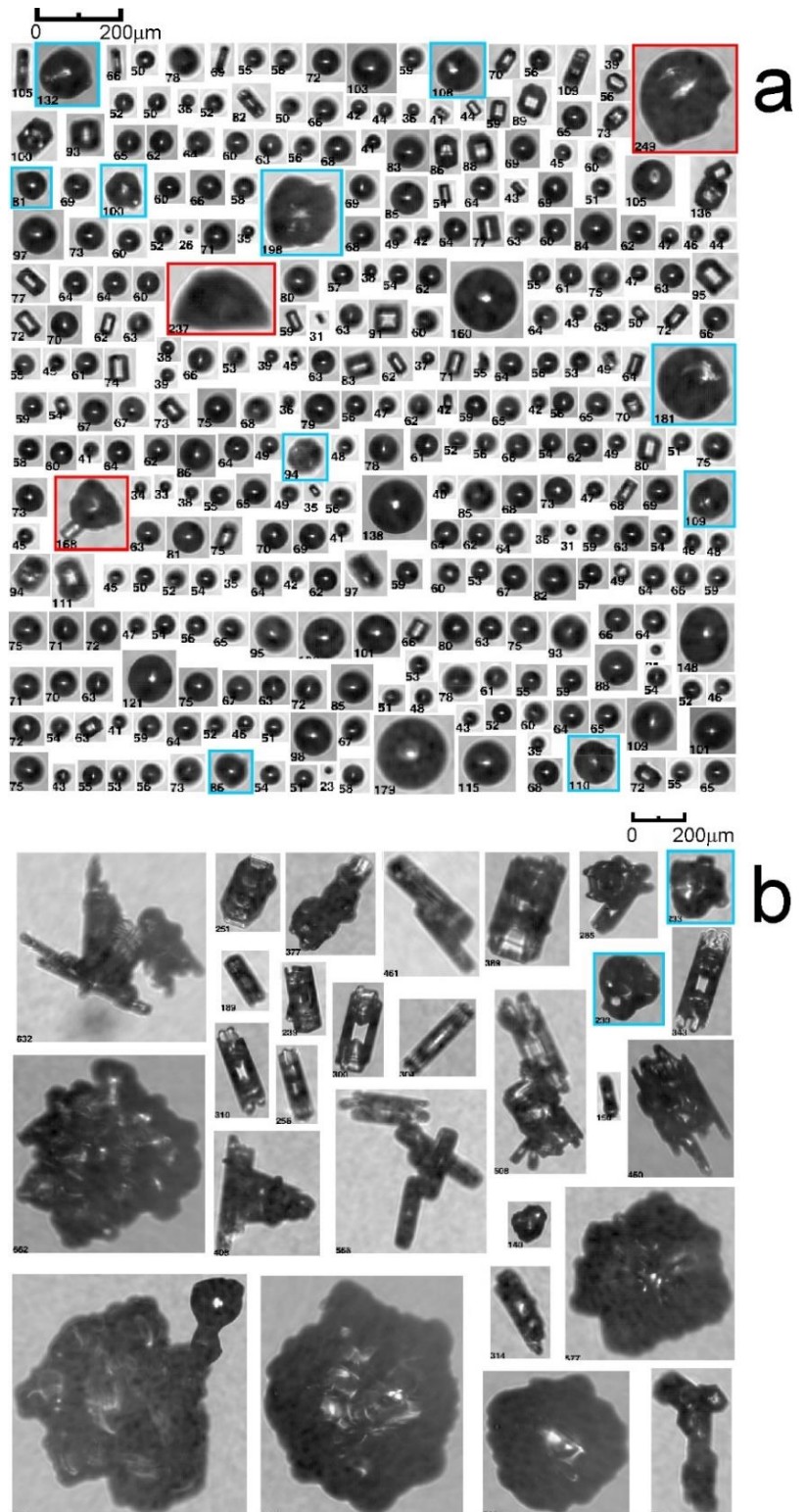

**Figure 6**. Spatial sequence of CPI images of (a) droplets and faceted ice crystals and (b) aged large ice particles. (a) Blue frames indicate frozen droplets with modified shapes, and red frames fragments of shattered frozen drops. Numbers under each image indicate their maximum sizes $L_{max}$. Cloud particles in (a) and (b) are spatially mixed, and they were split between two panels because of their difference in size. The images were sampled at $T_a$ =-5C and $H$ =5650m during UTC 09:40:42 – 09:40:47 on 15 May 2015 during measurements shown in **Fig.5**.

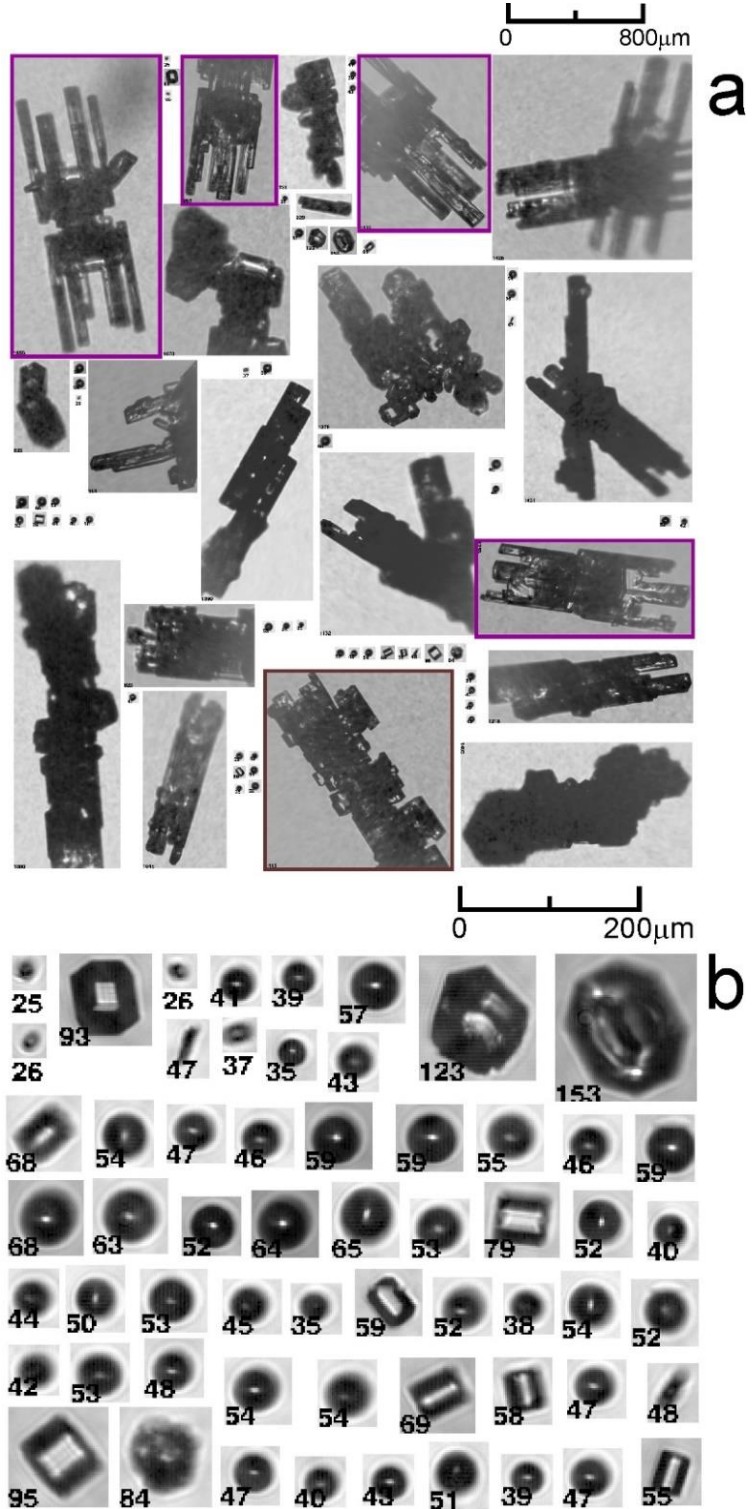

**Figure 7**. (a) Spatial sequence of CPI images; (b) Subset of droplets and faceted ice crystals from panel (a). Numbers under each image indicate their maximum sizes $L_{max}$. The images were sampled at $T_a$ =-5C and $H$ =5620m during UTC 09:46:36 – 09:46:39 on 15 May 2015 during measurements shown in **Fig.5**. (a) Purple frames indicate images of ice particles with evidence for their vertical circulation in the storm.

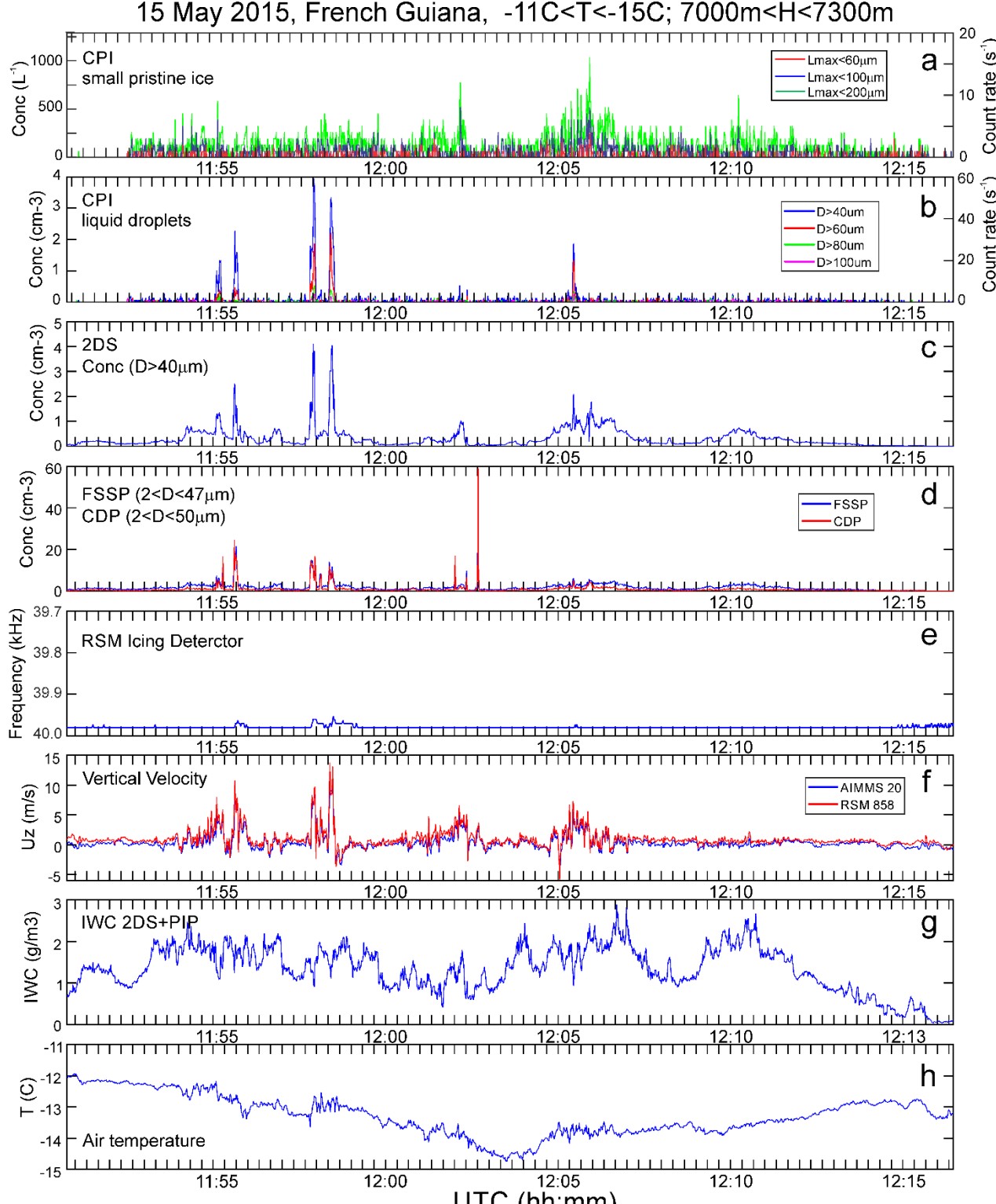

**Figure 8**. Same as in **Fig.5**. The altitude of measurements varied between 7000m and 7300m.

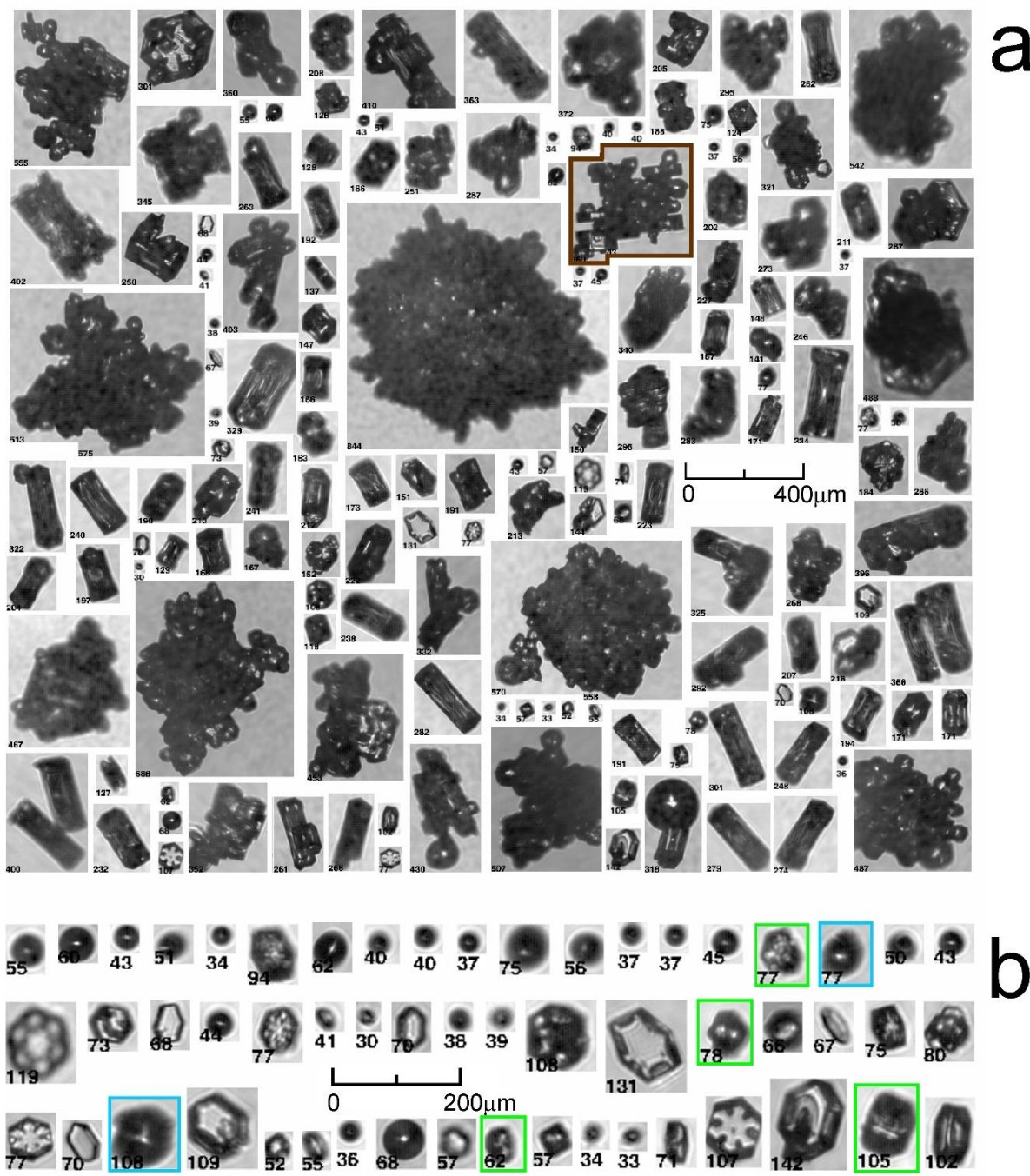

**Figure 9.** (a) Spatial sequence of CPI images; (b) Subset of droplets and faceted ice crystals from panel (a). (b) Blue frames indicate frozen droplets with modified shapes, and green frames frozen drops with developed facets. Numbers under each image indicate their maximum sizes $L_{max}$. The images were sampled at $T_a$ =-14C and $H$ =7200m during UTC 12:05:27 – 12:05:38 on 15 May 2015 during measurements shown in **Fig.8.**

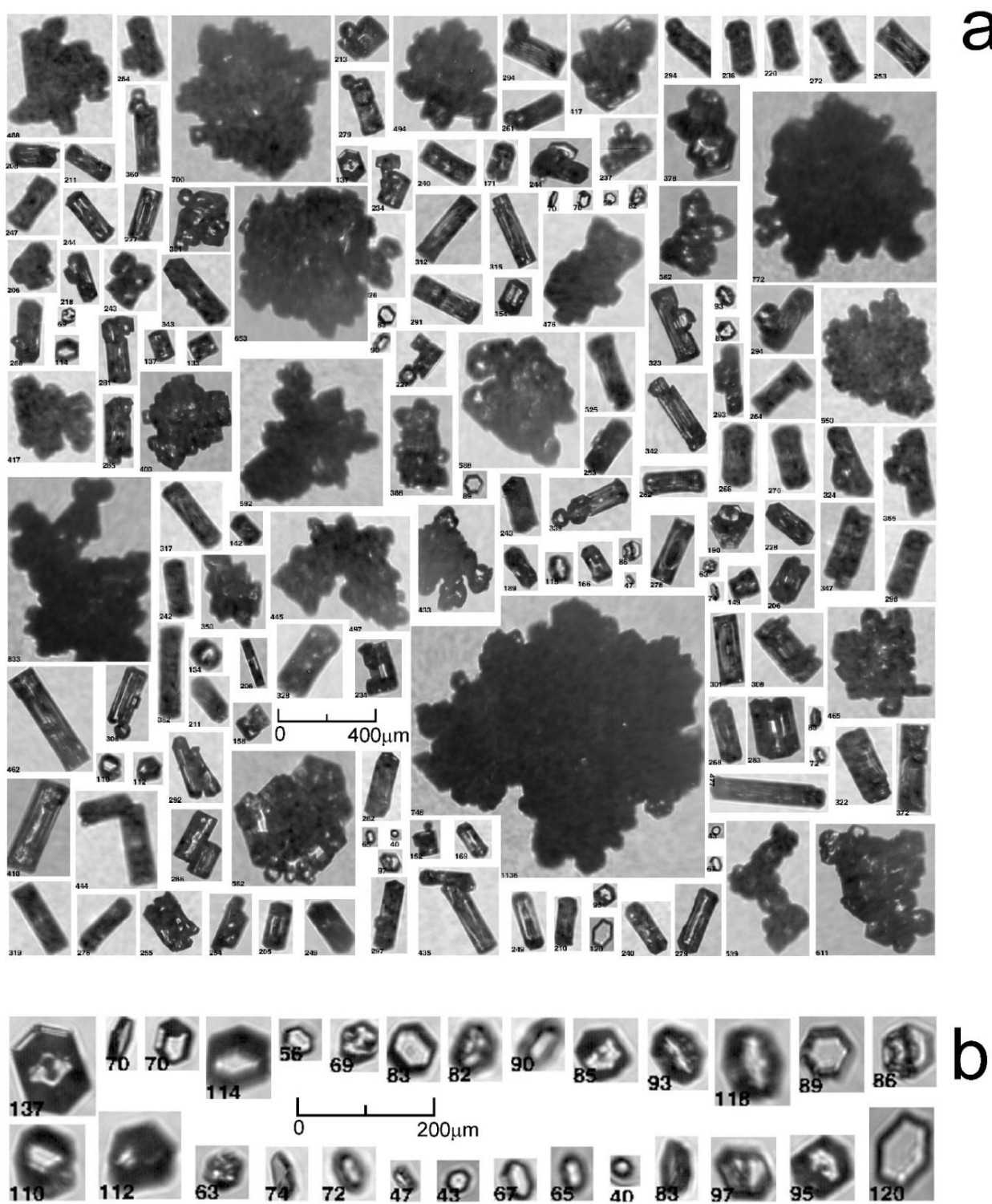

**Figure 10**. (a) Spatial sequence of CPI images; (b) Subset of droplets and faceted ice crystals from panel (a). Numbers under each image indicate their maximum sizes $L_{max}$. No liquid droplets were present in this cloud region. The images were sampled $T_a =$ -14C and $H =$ 7200m during UTC 12:05:47 – 12:05:53 on 15 May 2015 during measurements shown in **Fig.8.**

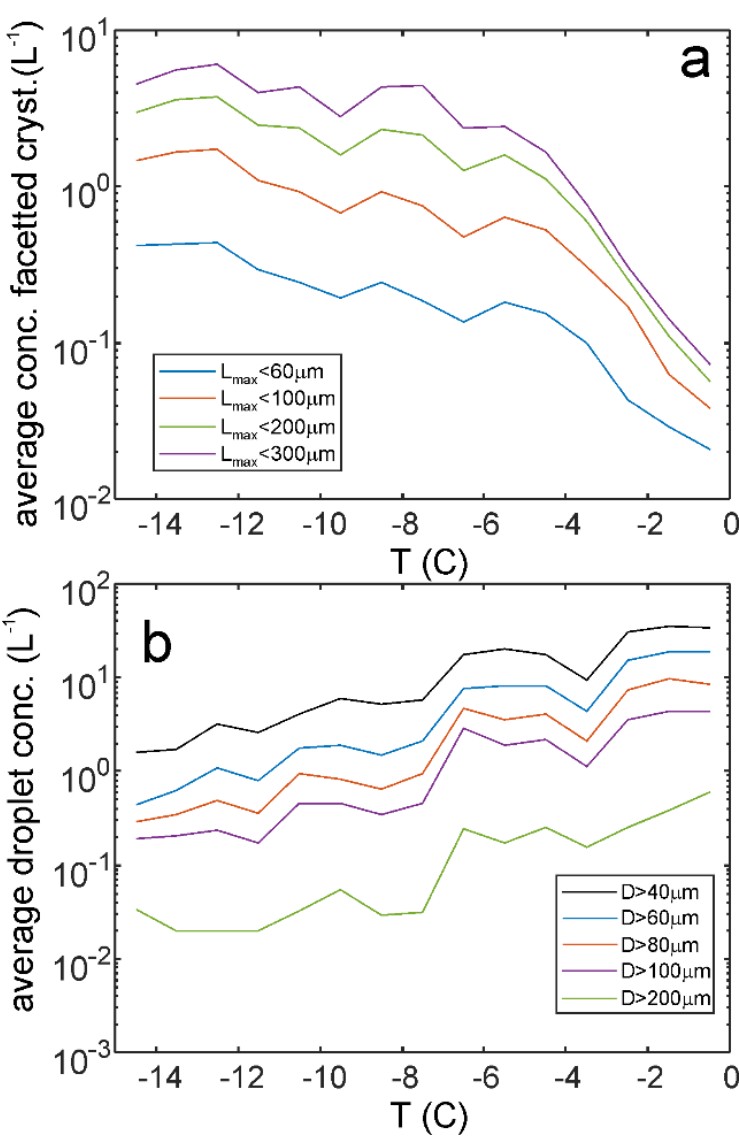

**Figure 11**. Average concentration of small faceted ice crystals (a) and drops (b) estimated from CPI measurements. The concentration was averaged over the entire flight length sampled during 13 flights in 10 tropical MCSs. The concentration was normalized on the sampling distance in each 1C temperature interval. Total number of 1s average samples $8.4 \times 10^4$, total in-cloud length 9580km.



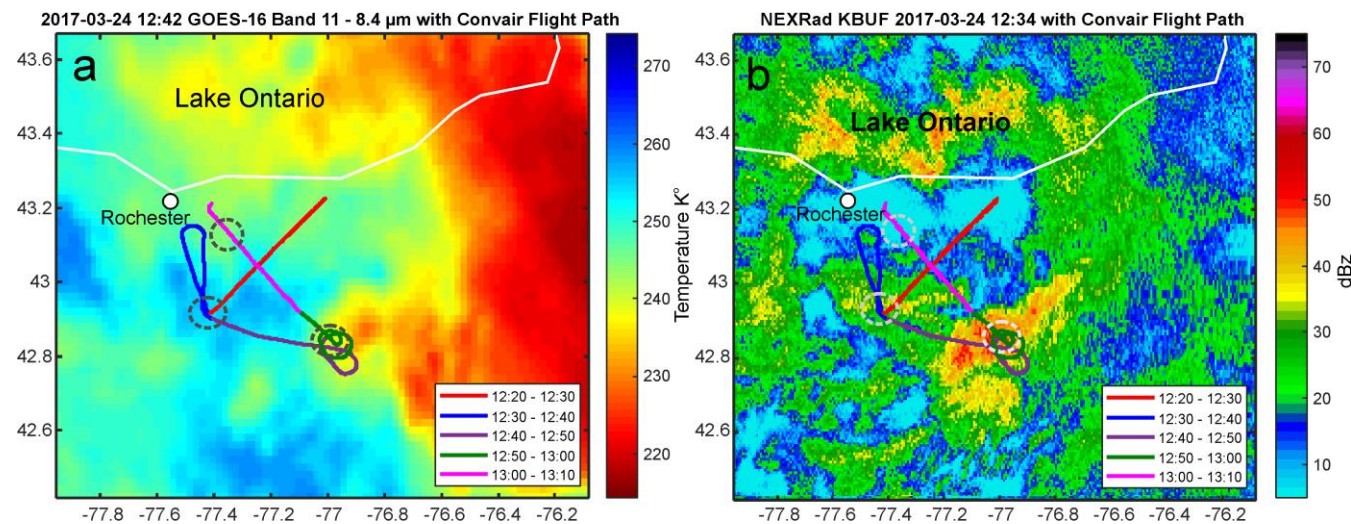

**Figure 12**. Flight track of the Convair-580 in the frontal cloud system on 24 March 2017 overplayed over (a) GOES-16 infrared image (download from Univ. Wisconsin); (b) KBUF (Buffalo, NY) NEXRAD reflectivity at elevation
0.46°. Dashed line circles indicate SIP cloud regions.

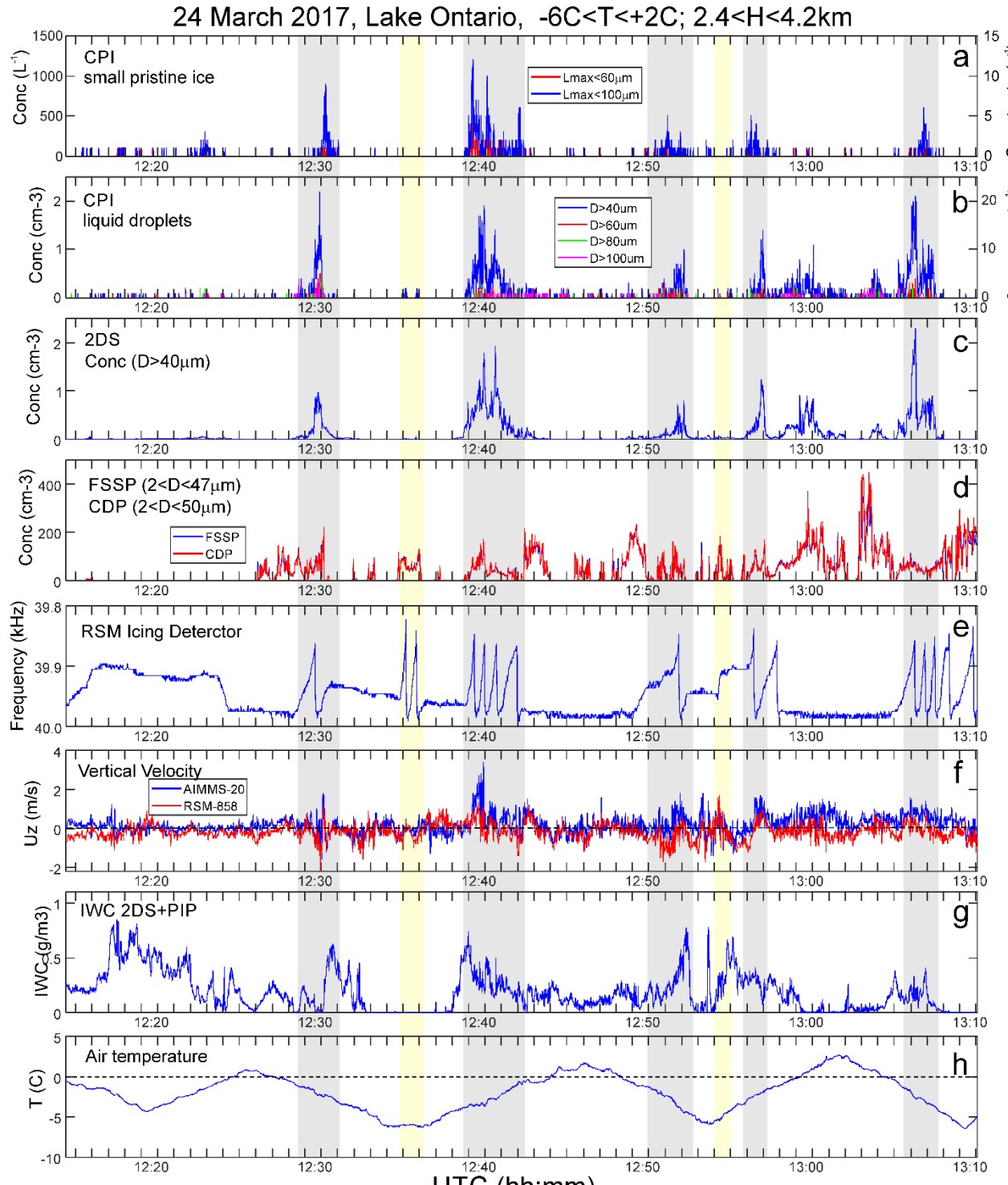

**Figure 13**. Time series of cloud microphysical parameters collected in a frontal cloud system over upstate NY on 24 March 2017. (a) CPI count rate of small pristine ice with $D_{max}$<60μm, 100μm; (b) CPI count rate of cloud droplets with $D$>40μm, 60μm, 80μm,100μm; (c) concentration of cloud particles D>40mm measured by 2DS; (d) concentration of cloud droplets measured by FSSP and CDP; (e) Rosemount Icing Detector frequency; (f) vertical velocity measured by AIMMS20 and Rosemount 858 probes; (g) IWC calculated from composite 2DS and PIP PSDs; (h) air temperature. Grey strips indicate cloud regions with enhanced concentration of small faceted ice particles; red and yellow strips indicate regions where ice and liquid were present, but no SIP was observed (see text).

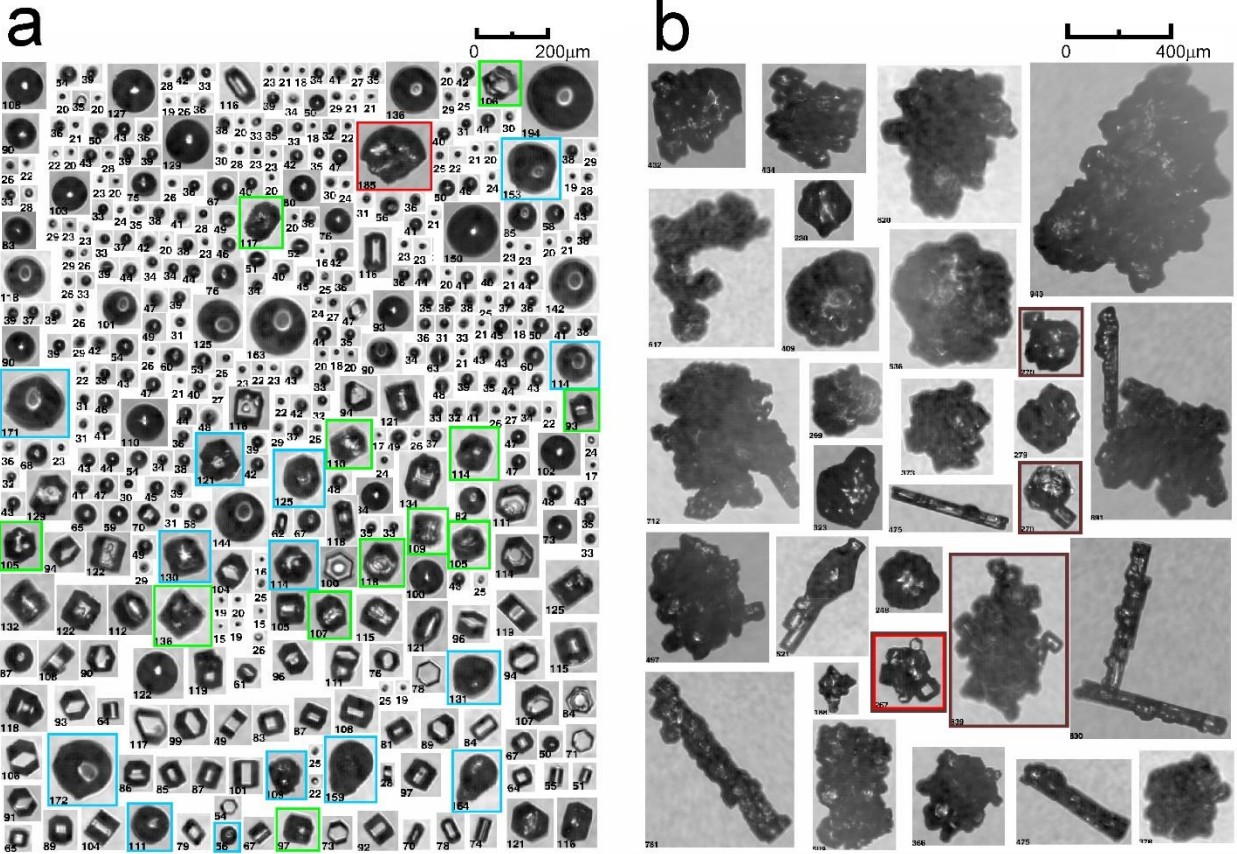

**Figure 14**. Spatial sequence of CPI images of (a) droplets and faceted ice crystals and (b) background large ice particles. (a) Blue frames indicate frozen droplets with modified shapes, green frames - frozen drops with developed facets, red frames - fragments of shattered drops. Numbers under each image indicate their maximum sizes $L_{max}$. Cloud particles in (a) and (b) are spatially mixed, and they were split between two panels because of their difference in sizes. The images were sampled at $T_a$ =-2C and $H$ =3500m during UTC 12:29:20 – 12:30:00 on 24 March 2017 during measurements shown in **Fig.13**.

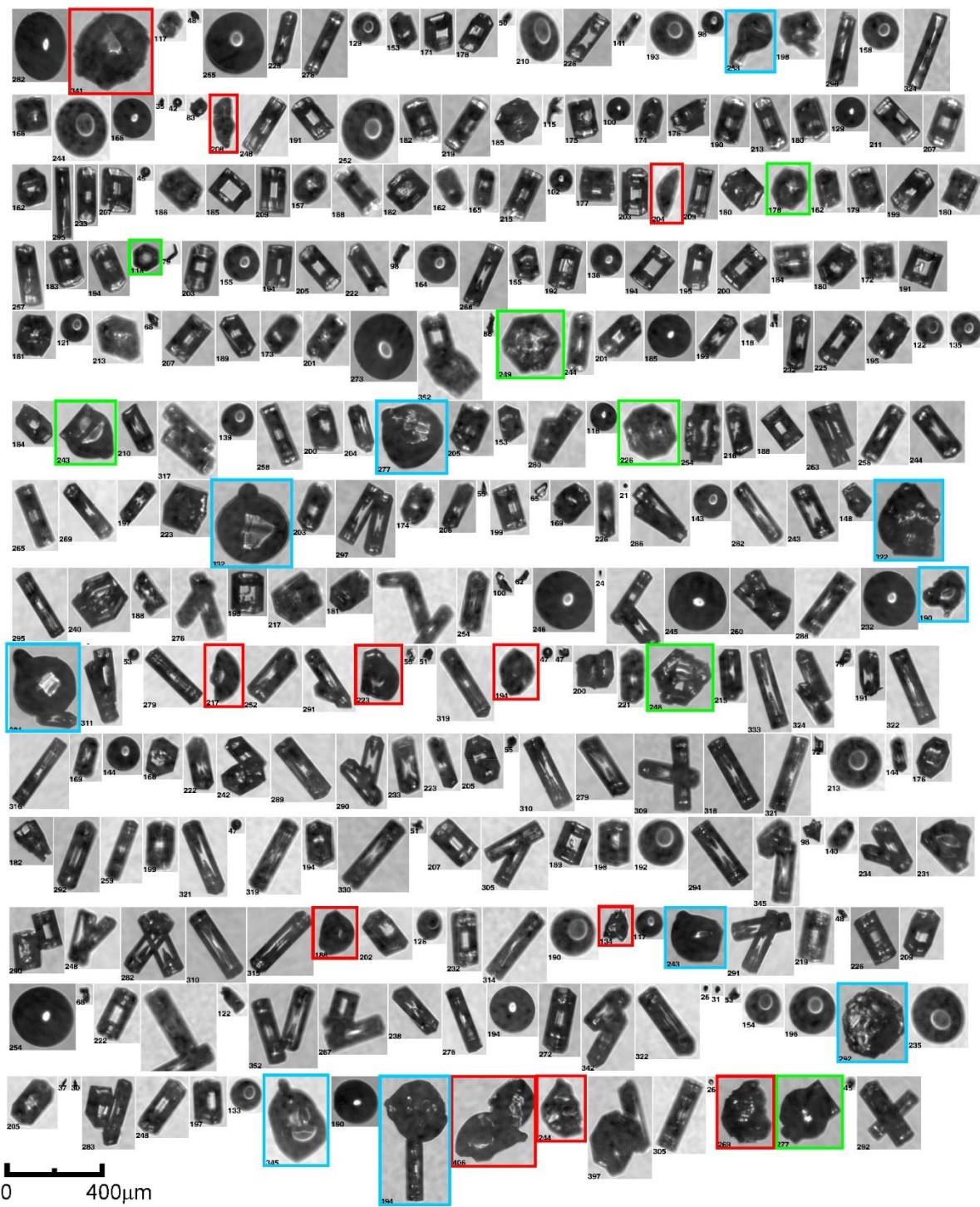


**Figure 15**. Spatial sequence of CPI images of droplets and faceted ice crystals. Blue frames indicate frozen droplets with modified shapes, green frames - frozen drops with developed facets, red frames - fragments of shattered drops. Numbers under each image indicate their maximum size $D_{max}$. The images were sampled during UTC 14:06:30-
14:07:30 on 24 March 2017 (not shown in **Fig.13**), $T_a$ =-3C, $H$ =2100m.


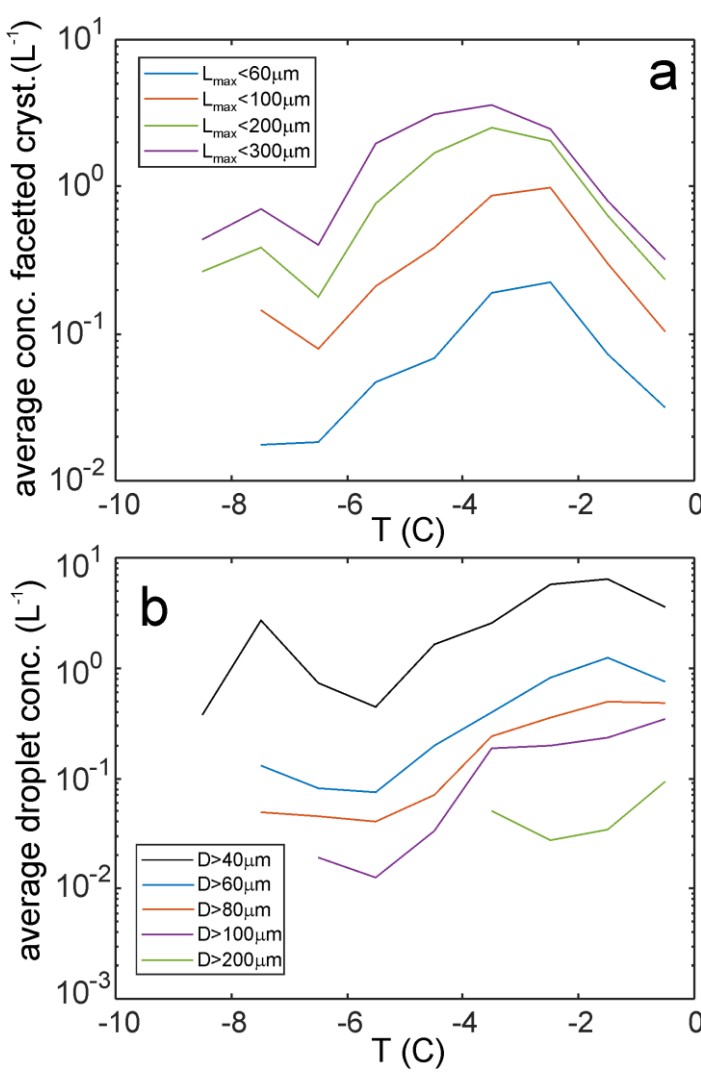

**Figure 16.** Average concentration of ice crystals (a) and drops (b) estimated from CPI measurements and normalized on the sampling distance in each temperature interval. The data were collected during two flights in mid-latitude frontal cloud systems with temperatures -10°C<$T_a$ <-0°C. Total number of 1s average samples 1.4×10⁴, total in-cloud aircraft path length 1380 km.


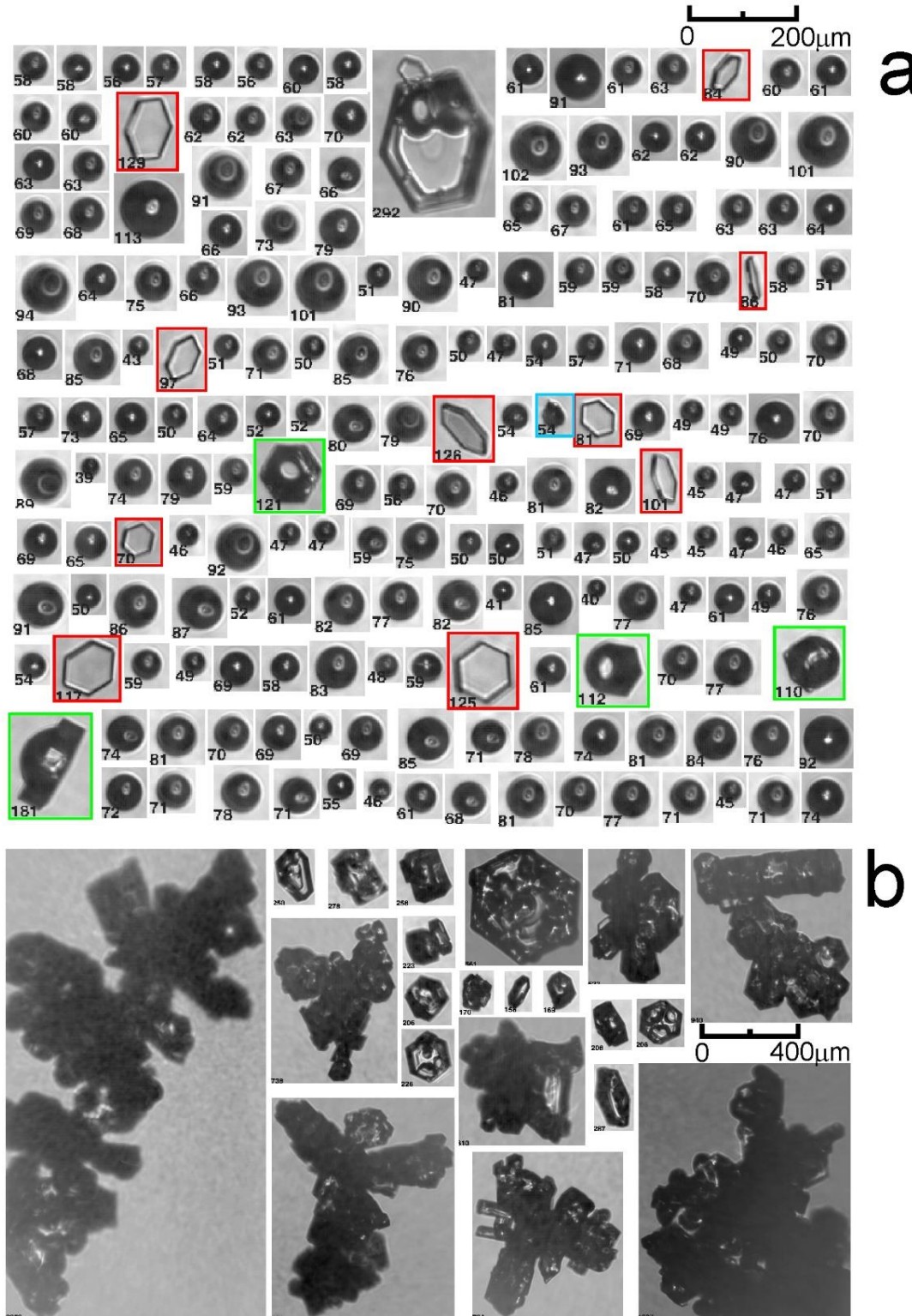

**Figure 17**. Spatial sequence of CPI images of (a) droplets and faceted ice crystals and (b) background large ice particles. (a) Blue frames indicate frozen droplets with modified shapes, green frames - frozen drops with developed facets, red frames - secondary ice particles developed into thin hexagonal plates. Numbers under each image indicate their maximum size $D_{max}$. Cloud particles in (a) and (b) are spatially mixed, and they were split between two panels because of their difference in sizes. The images were sampled during UTC 04:59:50-05:00:18, on 24 January 2017. $T = $ -1.5C, $H =$ 2400m


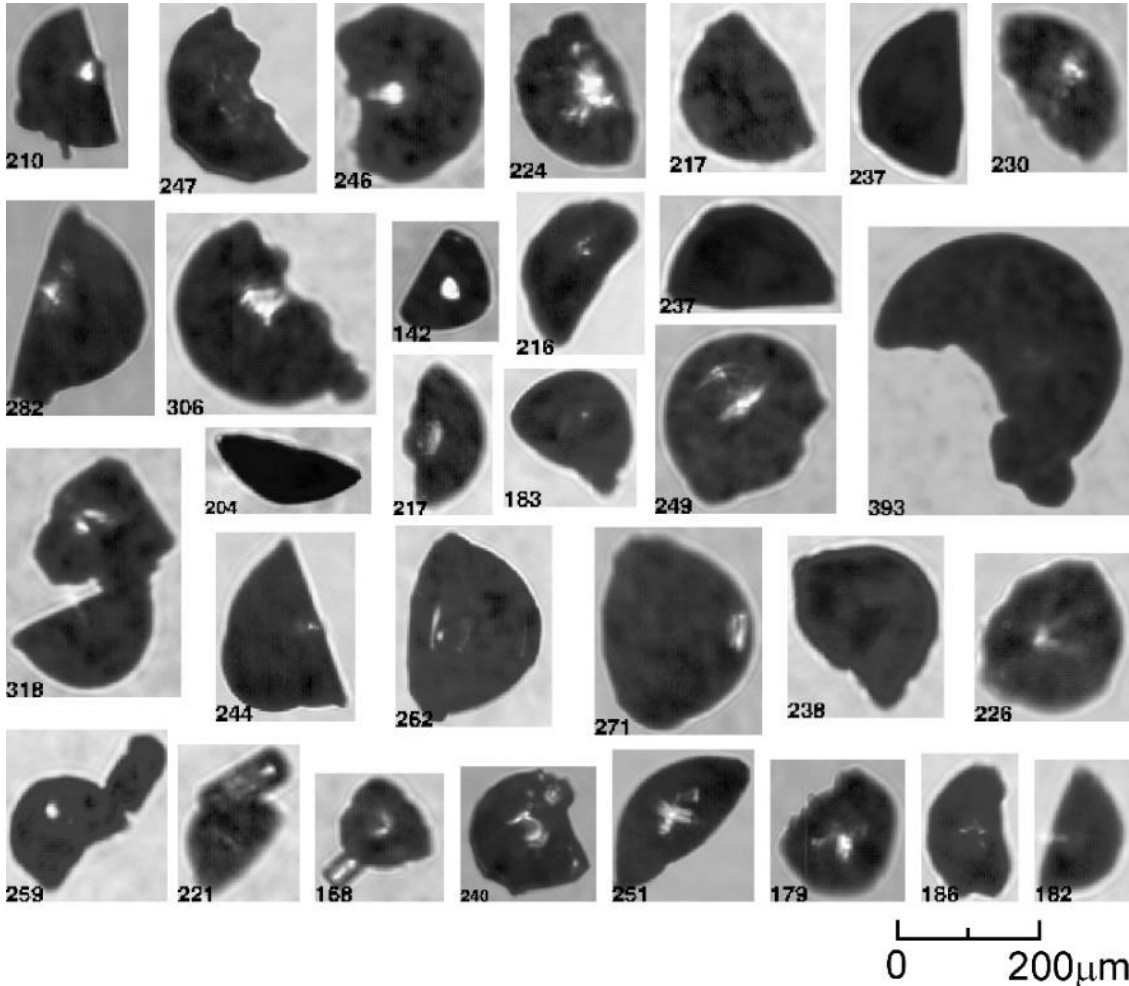

**Figure 18**. Images of fragmented frozen droplets collected in the SIP cloud regions indicated by grey areas in **Fig.5** and **13** at -5C< $T$ <-1C.


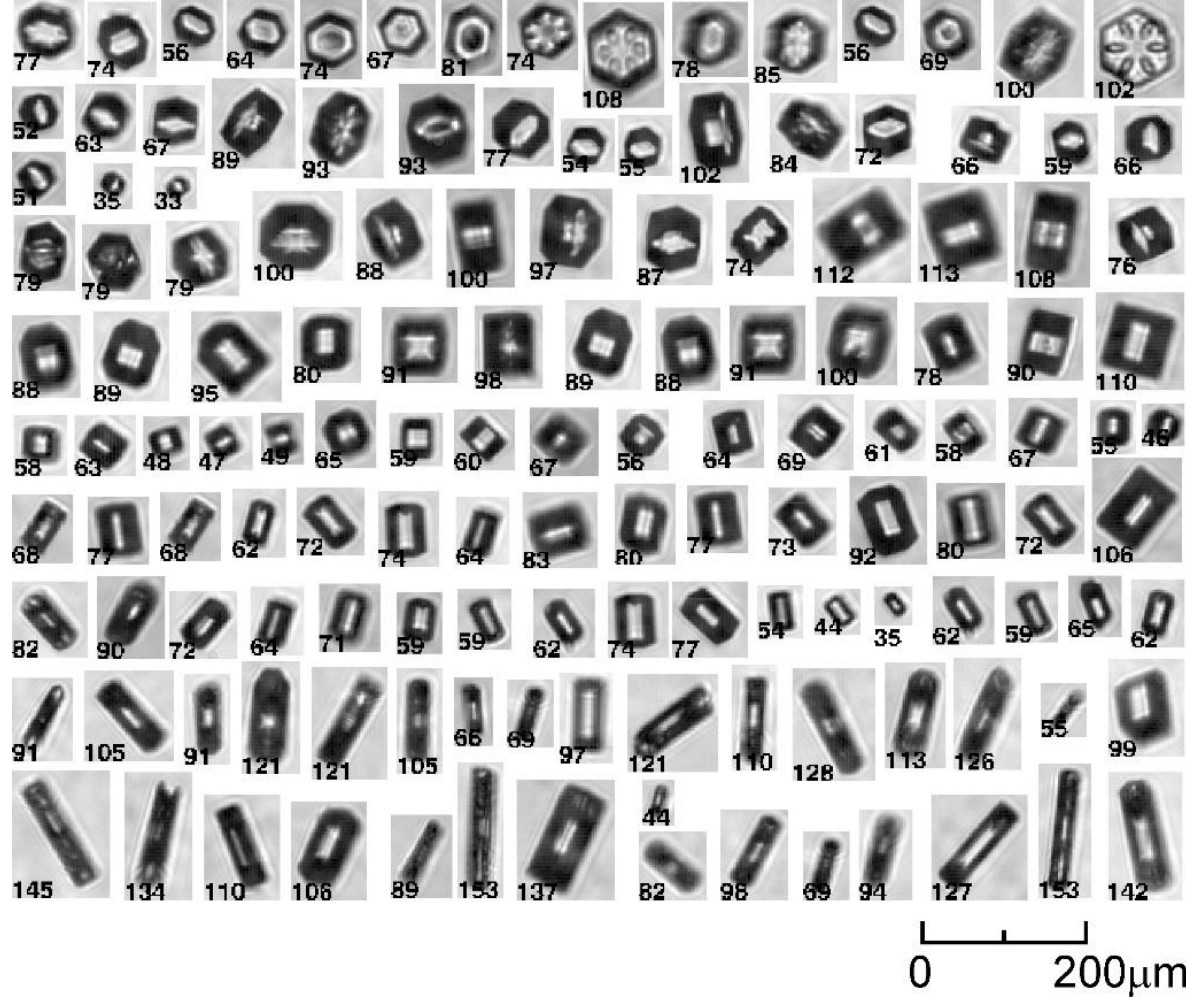

**Figure 19.** Images of small faceted ice particles, which were sampled in SIP cloud regions at -5.5C< $T_a$ <-5C, $H$ =5600m indicated by grey color in **Fig.5**. The aspect ratio of the small hexagonal prisms varies in the range 1545 0.3<R<6

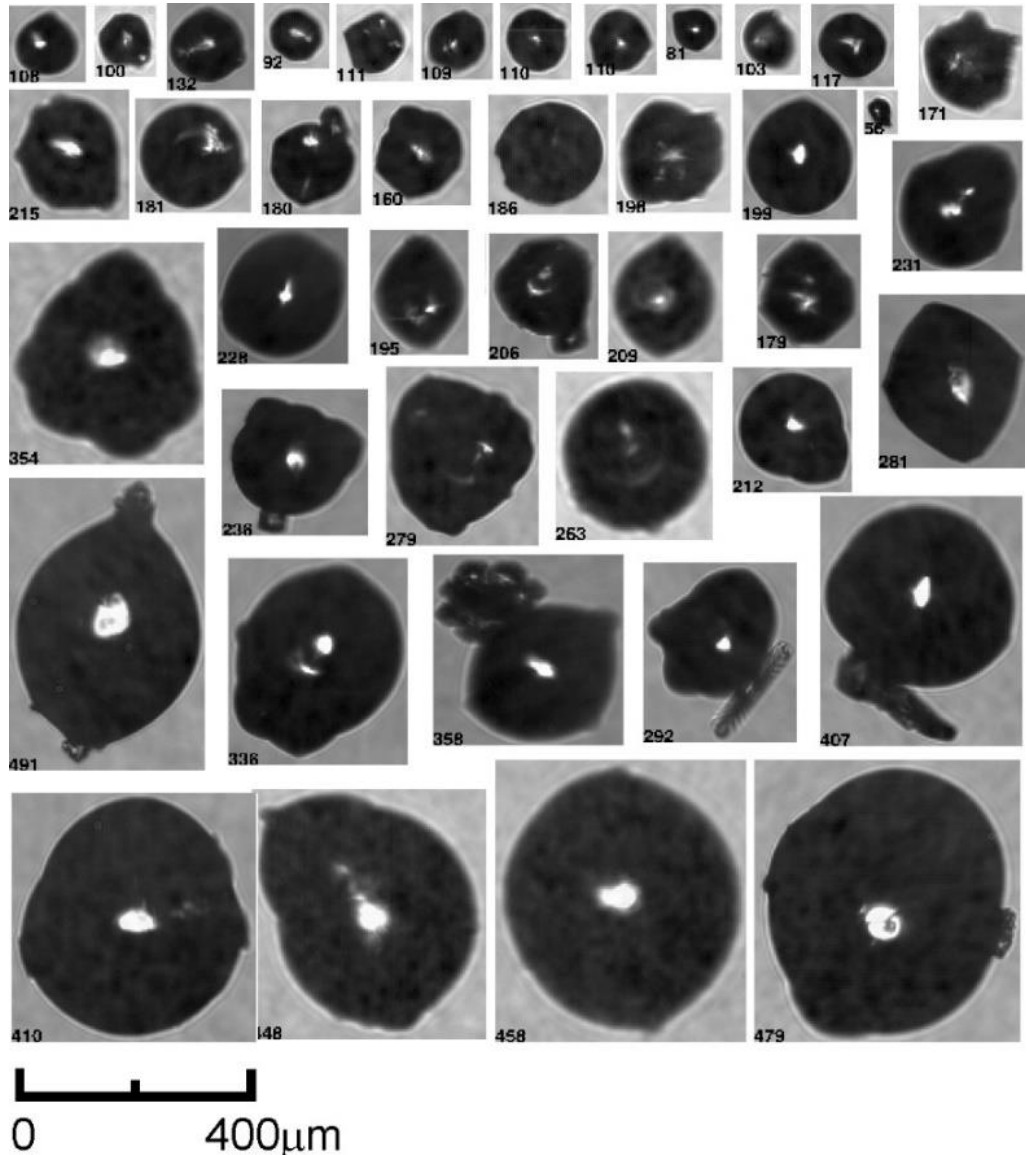

**Figure 20**. CPI images of single frozen droplets, which shape was modified during freezing collected in SIP cloud regions in the temperature range -5C< $T$ <-1C.

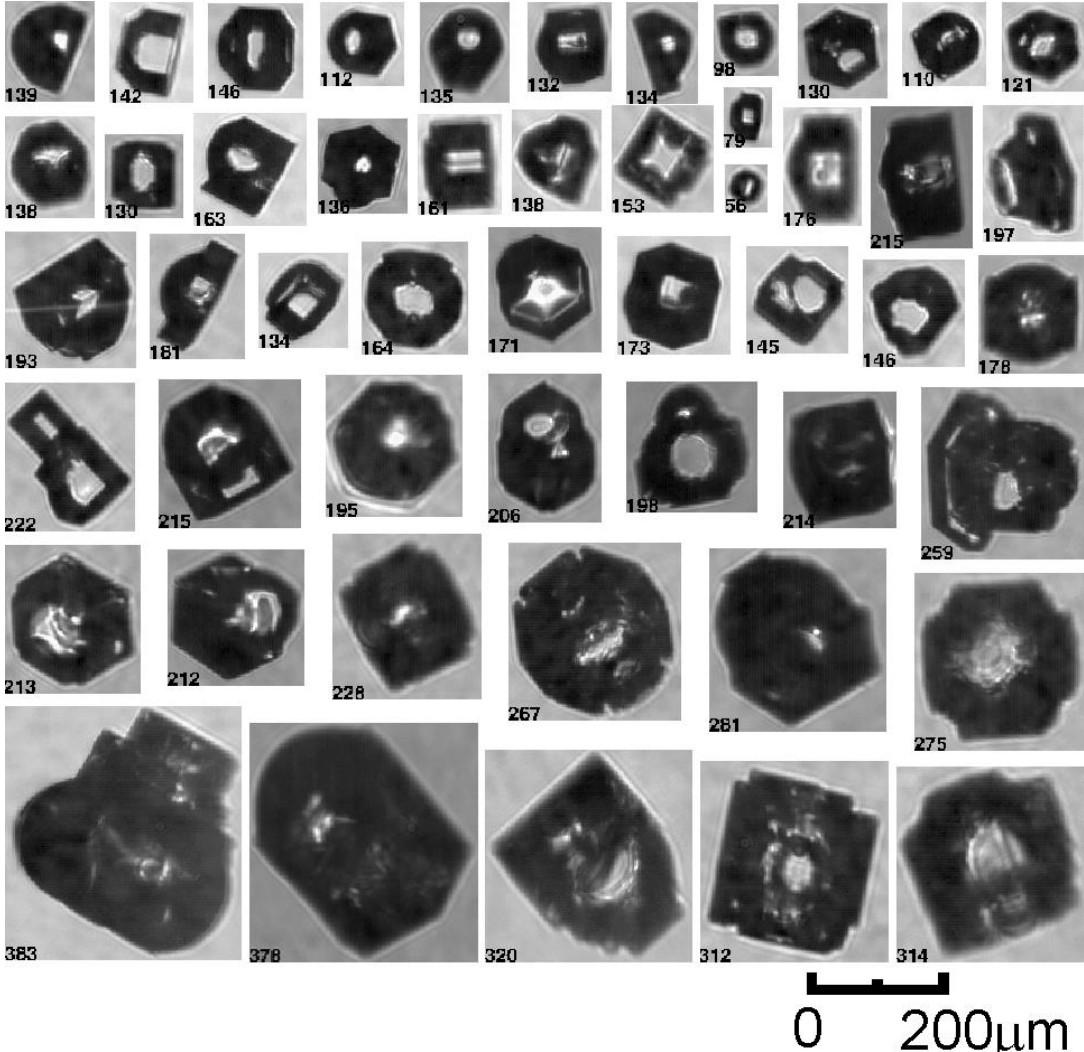

**Figure 21.** Images of frozen droplets partially regrown into faceted ice crystals -5C< *T* <-1C.



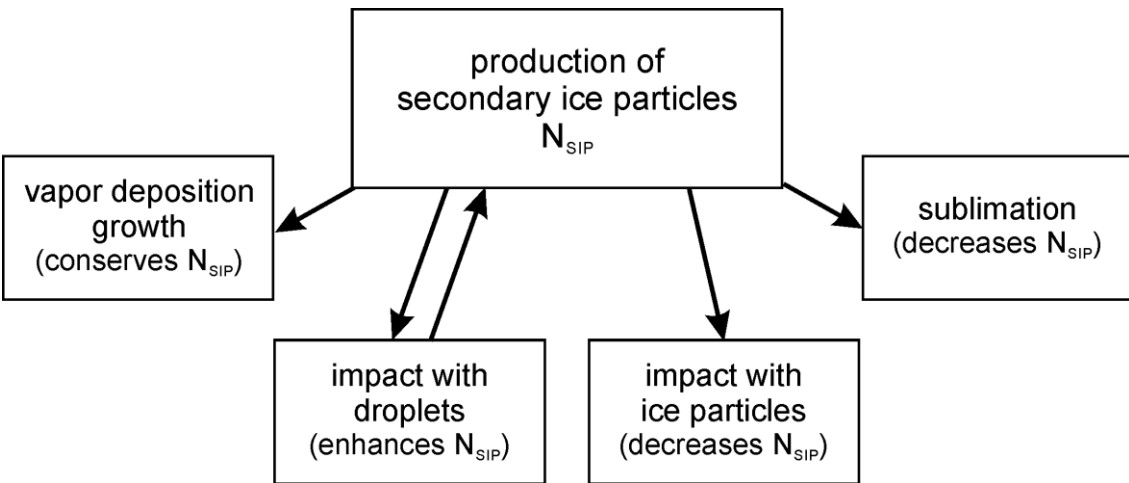

**Figure 22.** Different scenarios of evolution of SIP particles after their production.

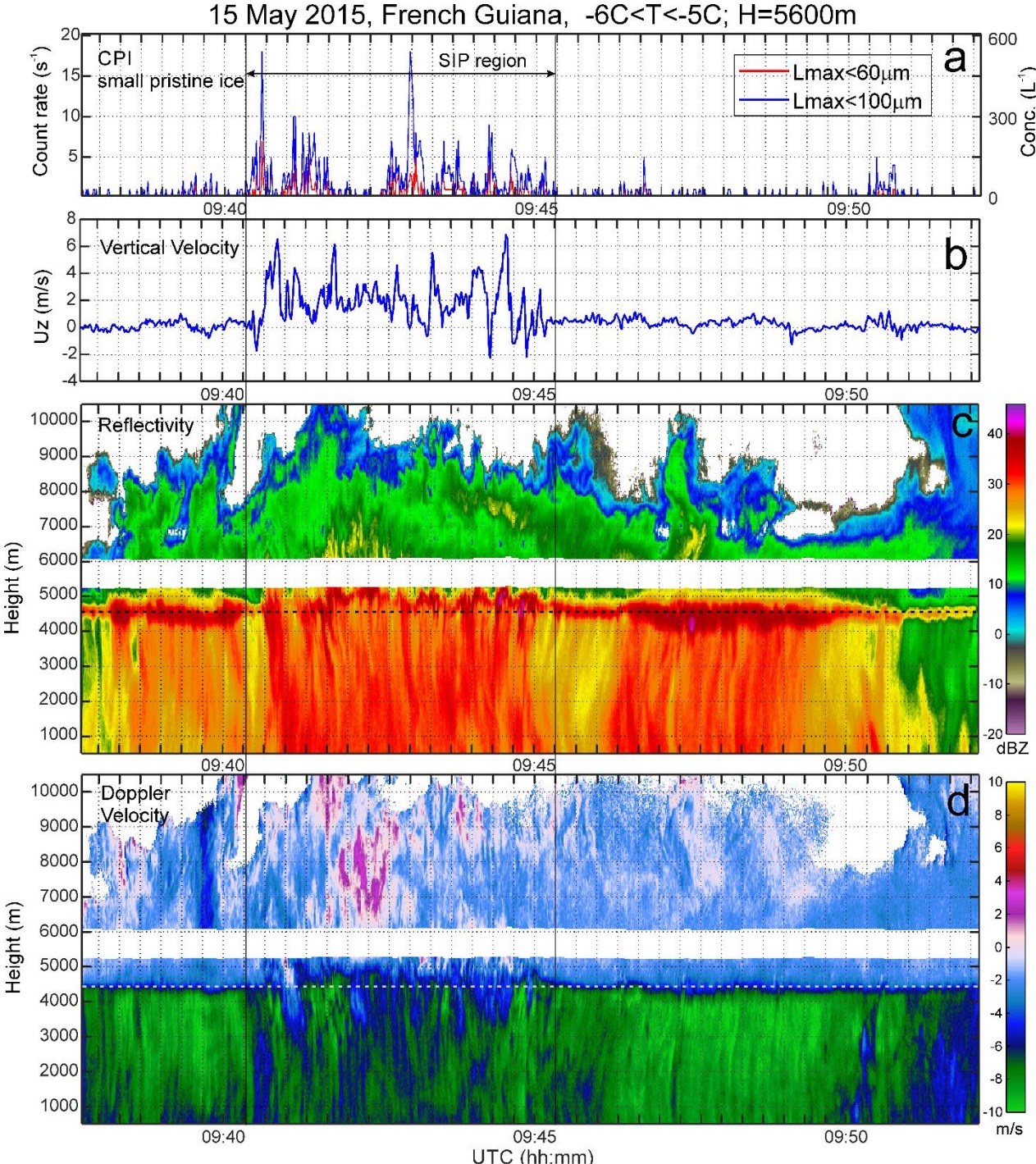

**Figure 23**. Zoomed time segments of the time series in **Fig.5** with the counting rate of small pristine ice particles (a), vertical velocity (b), X-band radar reflectivity (c) and Doppler velocity (d), measured during a traverse of the convective region inside tropical MCS. Horizontal dashed lines in (c,d) show the level of the bright band in undisturbed by convective updrafts cloud regions. Two vertical solid lines indicate SIP cloud region, which spatially coincides with the convective cell (b) and elevated bright band (c).


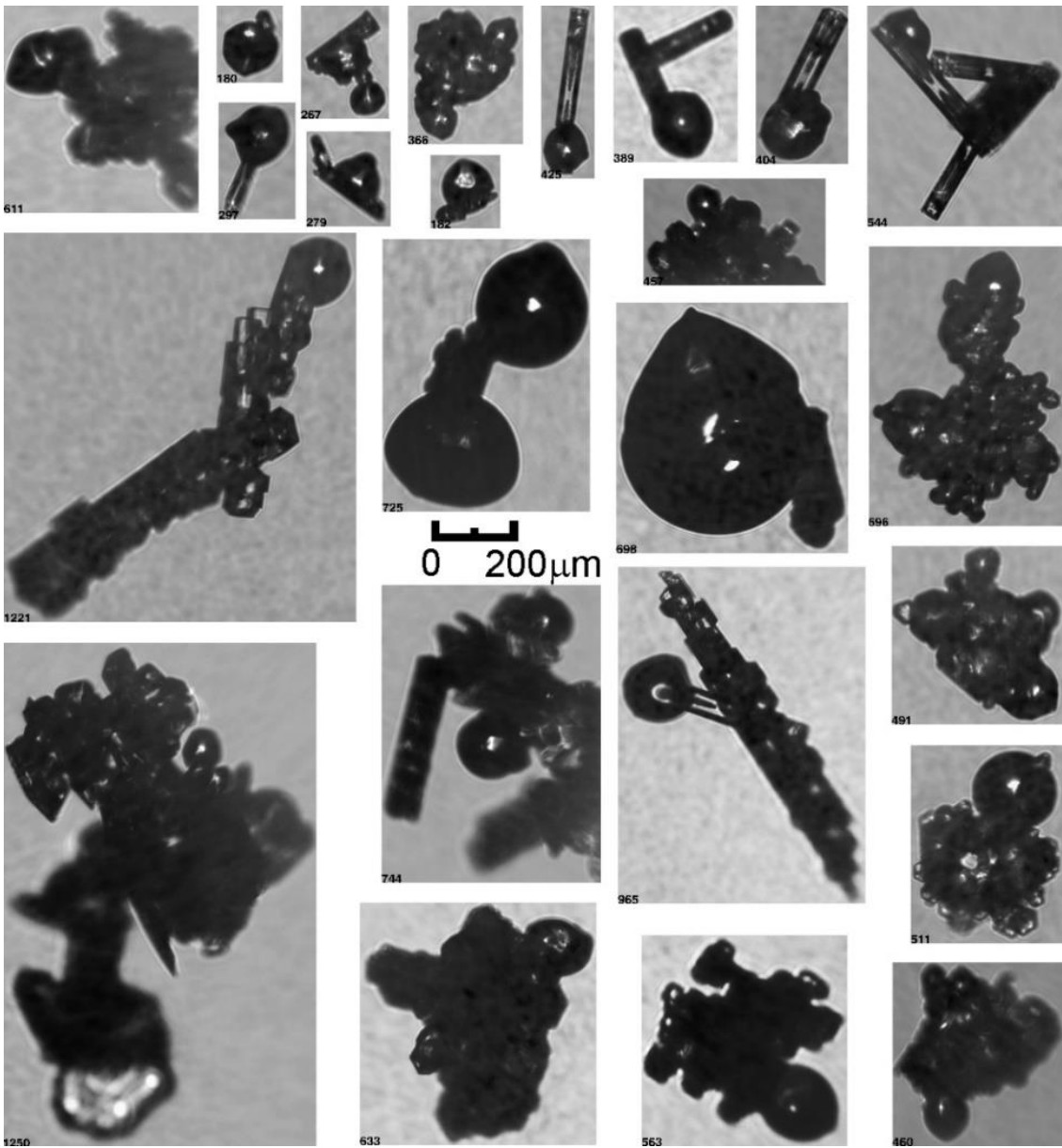

**Figure 24**. Images of frozen droplets attached to ice crystals that initiated their freezing. The shape of the frozen droplets was modified during freezing. Images were collected in the temperature range in the temperature range -
15C< *T* <-1C.

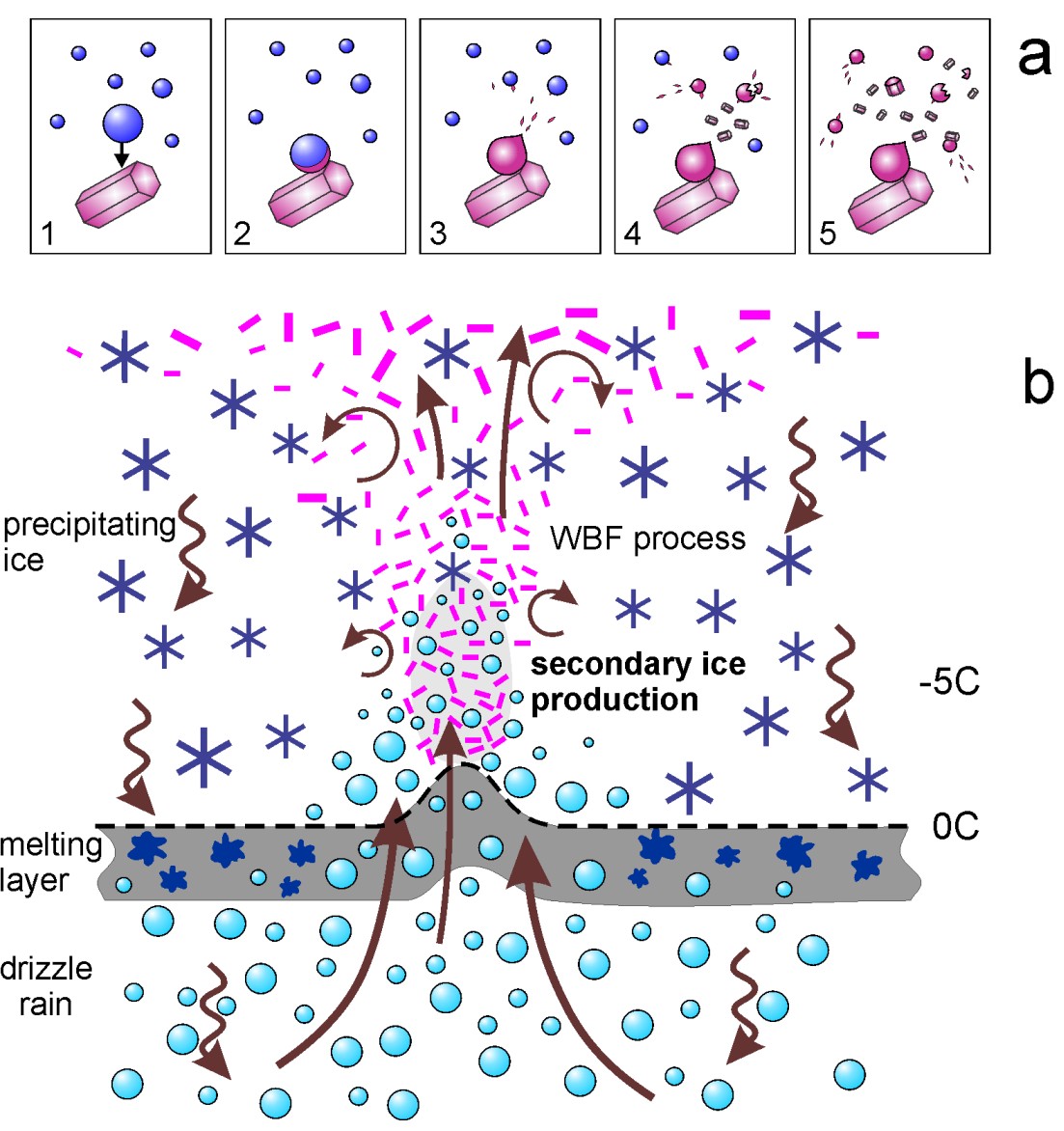

**Figure 25.** (a) Conceptual model of secondary ice production due to shattering of freezing drops. (b) Conceptual model of the effect of melting layer on the secondary ice particle formation in MCSs and frontal clouds.


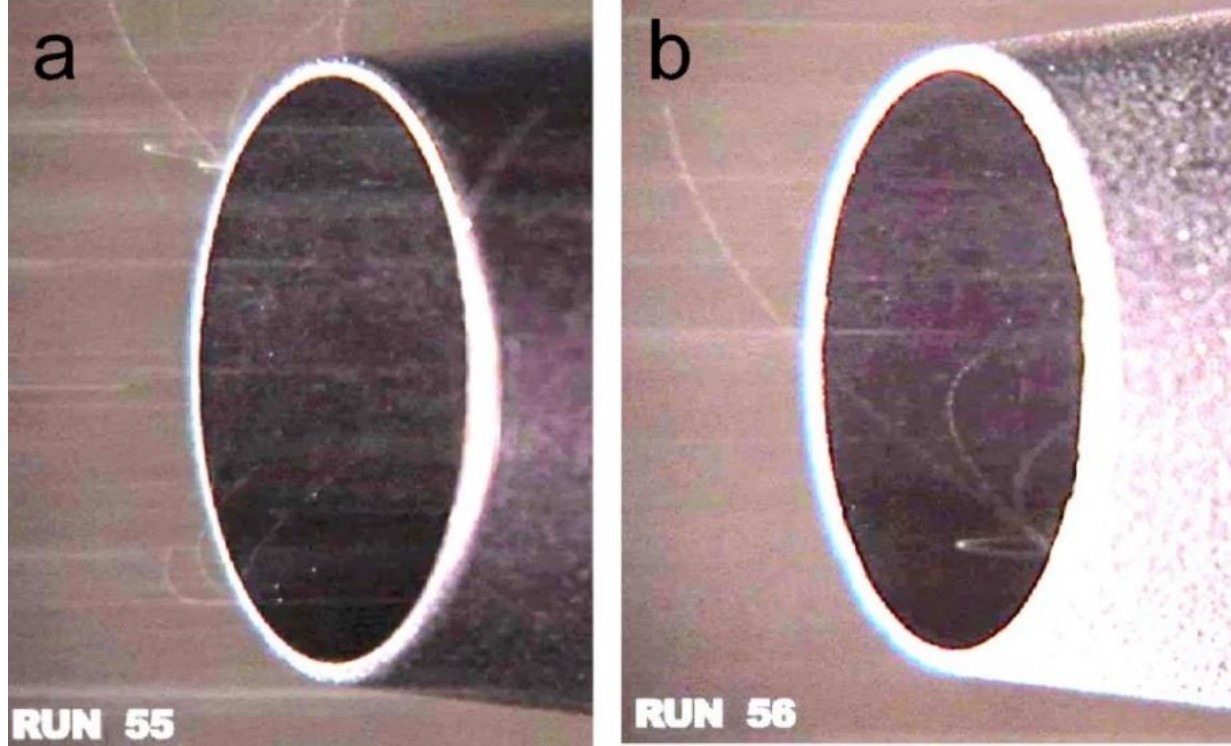

**Figure A1.** Snapshots from a high-speed video of trajectories of shattered and rebound ice particle fragments formed on impact with the CPI inlet. The measurements were conducted in the Cox and Co. wind tunnel facility (Long
Island, NY, USA) in ice spray at TAS=80m/s.


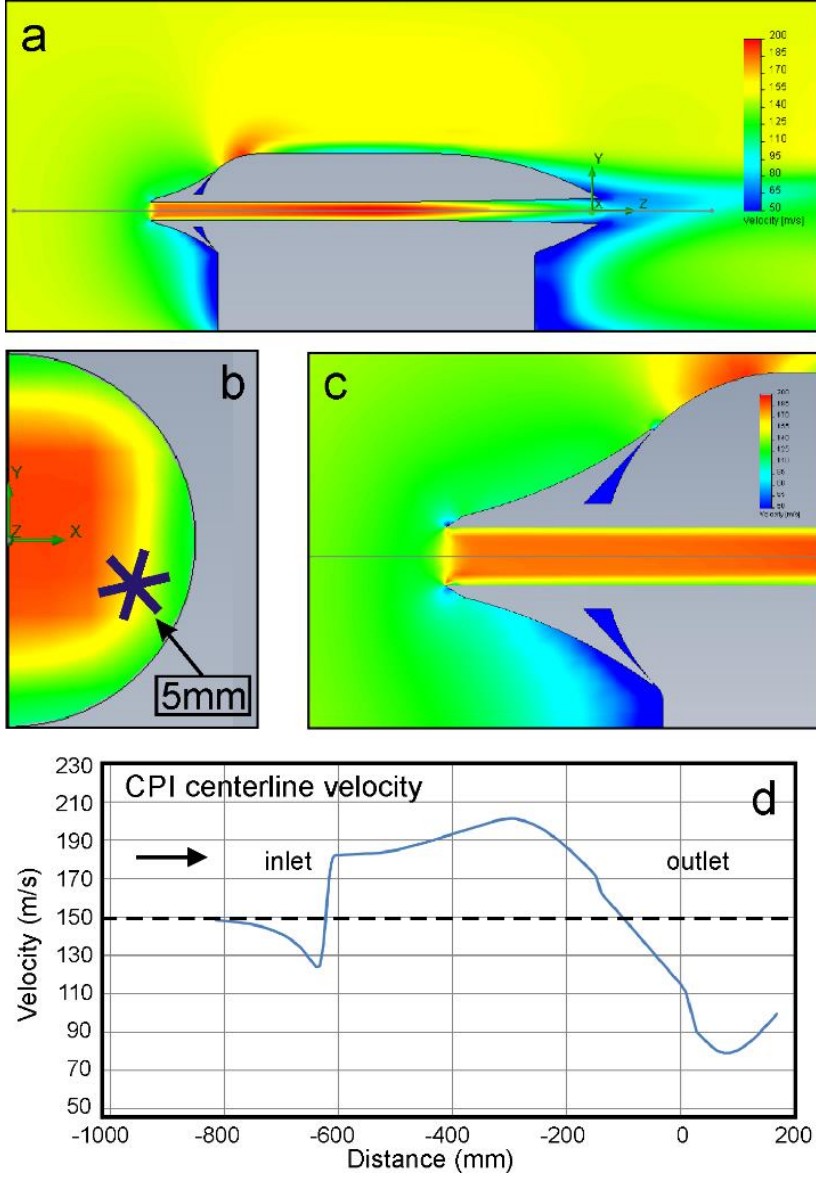


**Figure A2.** Results of the CFD analysis of flow around and through the CPI sampling tube. (a) airspeed around the CPI sensor head; (b) cross-section of speed inside the CPI inlet tube at the location of the sample volume; (c) zoomed CPI inlet area as in (a); (d) changes of the air velocity along the CPI inlet tube centerline. The simulation was performed for P =450mb, $T_a$ =-40C, TAS =150m/s



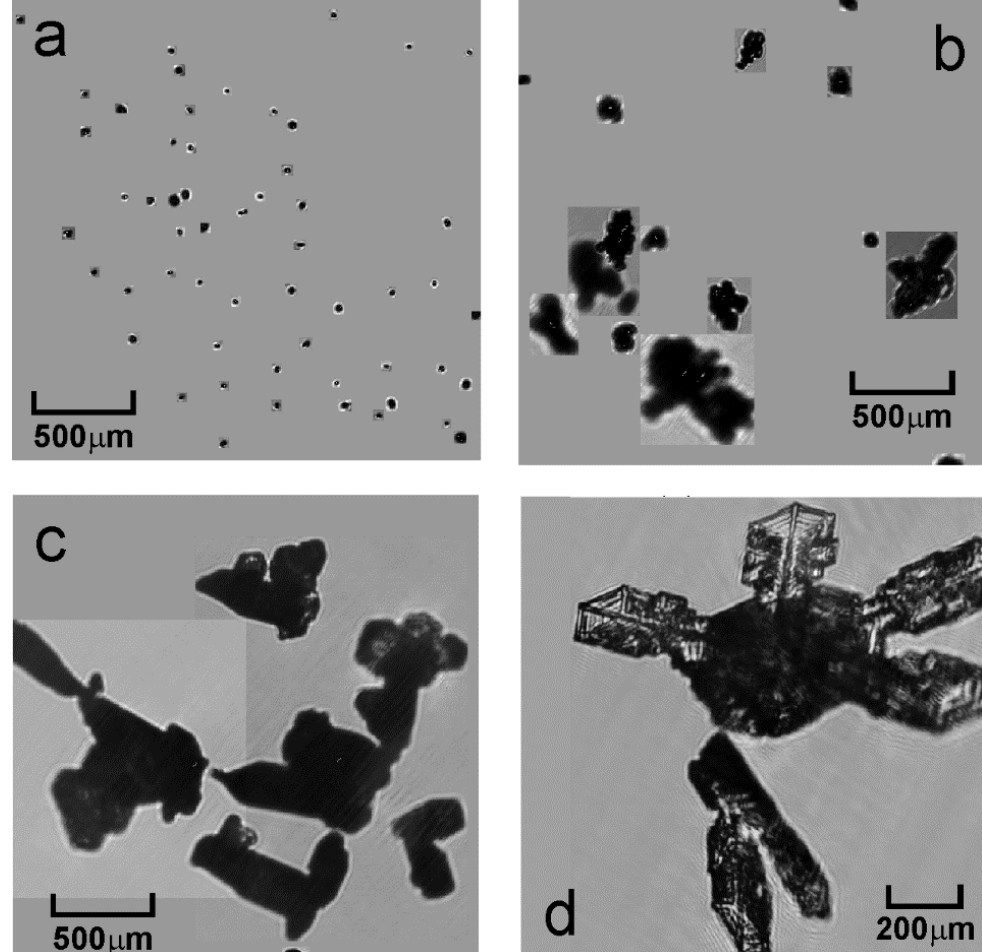

**Figure A3.** Multiple images registered in 2.3 mm ×2.3mm CPI image frames (a,b,c). Images in (a,b) are identified as
a result of shattering due to mechanical impact with the CPI inlet. Images in (c,d) are likely result from fragmentation
due to aerodynamic stresses in the CPI inlet tube.

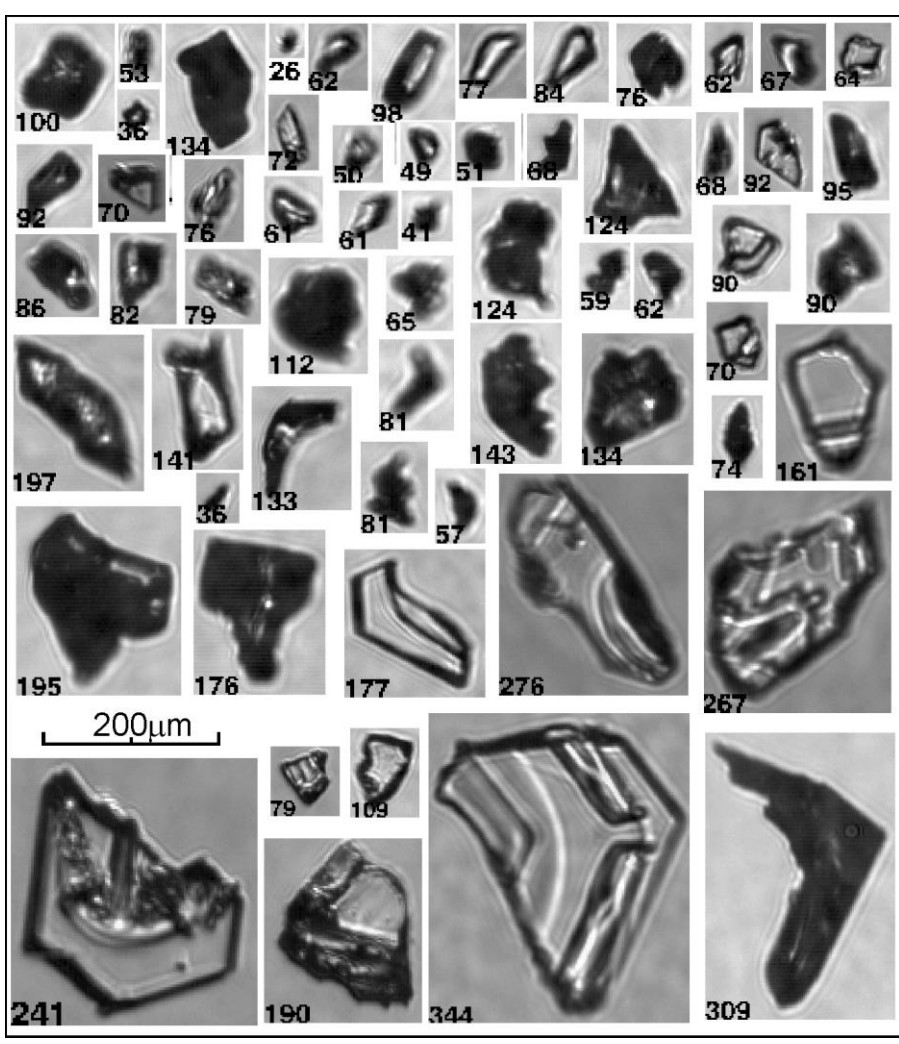

**Figure A4**. Examples of CPI images identified as shattering artifacts. Such images were excluded from analysis.