# Peer review of "A new look at the environmental conditions favorable to secondary ice production."

_Atmospheric Chemistry and Physics, 2019_

## Referee Comment (RC1) · Charles Knight (Referee) · 21 Aug 2019

I have a question and a comment to communicate. The question is, were the flight plans made to take APIPs into account? (aircraft-produced ice particles) This could be a serious issue especially, of course, when repeated penetrations in the same cloud have been made. The comment is (after saying that this seems to me a very important paper, and should be accepted) that the small ice particles that look like either fat ice columns or very thick, small plates. look to me like overgrowths on frozen, large, cloud droplets, not a result of overgrowths on the sort of shards that would result from the little explosions of freezing droplets. They are not at all what I would have expected from that. That does not help in identifying the actual cause of the freezing, of course, which seems to me the main mystery. Charles Knight

---

## Referee Comment (RC2) · Anonymous Referee #2 · 30 Aug 2019

This paper summarizes observations from two field projects where the Convair-580 sampled mixed phase clouds from a tropical MCS and from a frontal cloud. This paper is an excellent contribution to the scientific literature. I find that the most important conclusion of this study is to emphasize how we sorely need laboratory studies to have any sort of robust quantification of the relative frequency of SIP mechanisms, and the authors have done a good job in placing the context the two studies they did with the past work that has been done.

---

## Referee Comment (RC3) · Anonymous Referee #3 · 8 Sep 2019

Summary: This manuscript addresses possible mechanisms for generating secondary ice in two different case studies: a tropical, maritime MCS collected during the High Ice Water Content (HIWC) field campaign, and a midlatitude, continental frontal cloud system collected during the Buffalo Area Icing and Radar Study 2/Weather Radar Validation Experiment (BAIRS 2/WERVEX). Both cases used observations collected from the National Research Council (NRC) Convar580, with nearly the same instrumentation, but the main focus is upon the CPI image analysis. A theoretical section in the paper describes the approach, to derive a maximum ice particle size that would be indicative of those ice particles occupying similar thermodynamic characteristics as where they originated. Using that maximum size, the authors examine the characteristics surrounding large number concentrations of particles less than or equal to that

maximum size in order to identify possible secondary ice production mechanisms responsible for those large numbers. The authors conclude that at times rime-splintering may be active, at others freezing-raindrop shattering may be active, and explain why other mechanisms can be ruled out. They also hypothesize that in some instances raindrop recycling from below the melting level upward might be important to the shattering mechanism.

I found the manuscript difficult to follow, for several reasons. First, it suffers from an "identity crisis", in that the introductory section leaves the reader confused about what the main goal of the paper is. (I expand on this point below.) Then, the repetitive nature of explaining each SIP mechanism, and then addressing them separately for each case, adds up to a lot of text, and back-and-forth jumping by the reader to understand why a conclusion is different in one section versus another. And finally, the conclusions section is rather lack-luster, essentially making points that we have known for quite a while, and can be found in numerous other manuscripts.

I believe that this manuscript presents a unique approach that is not adequately emphasized in the text, in the novel use of the CPI data to try and address multiple SIP mechanisms. No other study to my knowledge has attempted to do this with the theoretical basis constructed here. I do have some concerns about this approach that I'll discuss below, but I believe for this paper to have the impact that it could, the manuscript should be rewritten **to emphasize its unique contribution** (including the title), rather than **to try and make hard conclusions about which mechanism was likely most important** (I agree with the concluding section that the observations are too limited in this regard). As written, the manuscript attempts to draw hard conclusions (in places), but then undermines its own message by discussing how limited the observations are in determining the active mechanisms (lines 838-845, and 897-903).

Major comments: 1. Section 1 is very long. Because of the recent AMS Monograph review article by Field et al. (2017), this much explanation does not seem necessary regarding the different mechanisms. I'm not opposed to having this section in there,

but it reads like a review article, leaving the reader to wonder what this study is actually addressing. 2. Unclear focus. Following comment #1, at the end of the Introduction, a question is posed: "The question that arises is, could these observations reflect an actual occurrence of different types of SIP?" Is this then the focus of the paper? The 2nd section states the objectives as "Based on the results obtained the authors attempt to revisit the role of different SIP mechanisms and identify conditions favorable for SIP." Later in the Conclusions section however, it is stated that "The obtained results are expected to contribute in our understanding of SIP, and they may be used by cloud modeling studies for evaluation of secondary ice production in the numerical simulations of clouds." So was this the real objective, to essentially create some cases for future numerical modeling? The focus is extremely important here, because it would allow the authors to address one of these completely. In the current state, the manuscript falls short of meeting any of these objectives, in that the results are not novel, nor is the data set sufficiently described to provide a good target case for future numerical modeling. 3. Without more information, the observations as presented are insufficient to make a good case to test with numerical modeling. It is clearly stated that both of these cases were seeded with ice from overlaying cloud, but we are not told what temperatures the overlying clouds are, nor are the values of measured INP active at these temperatures given. How can a model get this case correct if such "initial conditions" are not provided? 4. Abstract. The final lines 27-31 are inconsistent with the discussion in the conclusions section. Please revise. 5. Line 345: In specifying the tau values, a uz of 1- 4 m/s was specified. But earlier in the manuscript, strong updrafts were noted, up to 15-20 m/s in the MCS (Line 254), and Fig. 8 shows the analysis was applied in 10-15 m/s updrafts. So how can these values of tau be applicable in that case? My concern is that in stronger updrafts, the small ice particles would be leaving their place of origin much more quickly than assumed in the paper. That would make the Lmax derived in the next section MUCH smaller. (Again, having some radar pictures of the vertical velocity structures would be helpful here and might have helped me understand why you used these values in deriving the taus.) 6. The arguments presented in sections

3.3 and 3.4 appear logical, but because the entire study hinges on this derived Lmax, a more thorough discussion of uncertainty in its estimate, and the implications of its uncertainty, is required. 7. I have two large concerns with the CPI analysis. First, I was happy to find the analysis of shattering on the CPI in the Appendix—I've always wondered about that. But that information does not seem to be discussed much in the main manuscript. I think it should be, or at least should be discussed a little more in the main text, how those fragments were eliminated compared to the small secondary ice. Second, I am not confident in the CPI-derived estimates of ice particle number concentration based on a scaling with water droplets from the 2DS—they would not have the same shattering effect as the ice crystals, and it is not clear that the CPI cameras would necessarily trigger at the same rate for water droplets versus ice. Why not use some times when the cloud is completely glaciated, and compare the CPI image rate with the CIP numbers at those times? While I don't feel that the authors have completely misidentified SIP episodes, their magnitude is highly questionable with the current approach. 8. Sections 7 and 8 could be condensed into more focused text, which would greatly help the reader comprehend the main points. 9. Lines 769-770: here is where a discussion of shattering on the CPI, and the removal of those particles and its uncertainty, is extremely important (and in the following sections as well). 10. Melting Layer Hypothesis: Especially in the maritime MCS, why can't the large supercooled drops just be growing in updrafts—why would they have to be recycled melted particles to be important? And if they were heavy enough to fall and melt, what mechanism would bring them back into an updraft? This section seems to really "reach" past what one can identify with your observations. Why not just let the data speak for themselves? Dual-polarization data would really be needed to pin this down.

Minor Comments: 1. Line 133: I think you mean "overestimated" rather than "underestimated". 2. Lines 170-175: This assumes that air at the cloud edge containing sublimating crystals in incapable of being reintroduced into the core of the cloud; numerous studies have shown that entrainment in cumuliform clouds can bring outside air into the interior of the cloud (but no studies have looked at the effects on ice, to my

knowledge.) 3. Lines 237-243: "One of the important finding of this study is that melting layers in many cases work as a source of large liquid drops, which then ascended to a supercooled environment via convective or turbulent updrafts. After impaction freezing by preexisting ice, the drops may shatter and initiate a chain reaction of secondary ice particle production." Why is this stated here in the "objectives and data sets" section? It's written like an abstract that lists the results. It's very confusing here- please move this to the abstract or the conclusions section. 4. Lines 252-254: some evidence of this glaciation would be good. Can't this be shown with observations and radar data? This might help the reader understand the nature of the "seeding" of these clouds, as referenced in Line 237. 5. Line 271: I'm pretty sure you didn't have an X-band radar mounted on the Convair. Please fix text. 6. Equation 6 needs a reference, or some explanation. 7. Lines 455-461: I am not convinced that one can tell from the CPI images if droplets spread out sufficiently to prevent rime-splintering from occurring. Comparing it to the roughness of the rods used in the laboratory (in the next paragraph) seems unreasonable.

---

## Author Comment (AC2) · 9 Nov 2019

**Replies to Reviewer 2 "A new look at the environmental conditions favorable to secondary ice production" by Alexei Korolev et al.**

*This paper summarizes observations from two field projects where the Convair-580 sampled mixed phase clouds from a tropical MCS and from a frontal cloud. This paper is an excellent contribution to the scientific literature. I find that the most important conclusion of this study is to emphasize how we sorely need laboratory studies to have any sort of robust quantification of the relative frequency of SIP mechanisms, and the authors have done a good job in placing the context the two studies they did with the past work that has been done.*

**Replies**
Authors appreciate the Reviewer's comments and positive assessment of the result of our study.

---

## Author Comment (AC3) · 9 Nov 2019

**Replies to Reviewer 3 comment on "A new look at the environmental conditions favorable to secondary ice production" by Alexei Korolev et al.**

**Summary:**
*This manuscript addresses possible mechanisms for generating secondary ice in two different case studies: a tropical, maritime MCS collected during the High Ice Water Content (HIWC) field campaign, and a midlatitude, continental frontal cloud system collected during the Buffalo Area Icing and Radar Study 2/Weather Radar Validation Experiment (BAIRS 2/WERVEX). Both cases used observations collected from the National Research Council (NRC) Convar580, with nearly the same instrumentation, but the main focus is upon the CPI image analysis. A theoretical section in the paper describes the approach, to derive a maximum ice particle size that would be indicative of those ice particles occupying similar thermodynamic characteristics as where they originated. Using that maximum size, the authors examine the characteristics surrounding large number concentrations of particles less than or equal to that maximum size in order to identify possible secondary ice production mechanisms responsible for those large numbers. The authors conclude that at times rime-splintering may be active, at others freezing-raindrop shattering may be active, and explain why other mechanisms can be ruled out. They also hypothesize that in some instances raindrop recycling from below the melting level upward might be important to the shattering mechanism.*
*I found the manuscript difficult to follow, for several reasons. First, it suffers from an "identity crisis", in that the introductory section leaves the reader confused about what the main goal of the paper is. (I expand on this point below.) Then, the repetitive nature of explaining each SIP mechanism, and then addressing them separately for each case, adds up to a lot of text, and back-and-forth jumping by the reader to understand why a conclusion is different in one section versus another. And finally, the conclusions section is rather lack-luster, essentially making points that we have known for quite awhile, and can be found in numerous other manuscripts.*
*I believe that this manuscript presents a unique approach that is not adequately emphasized in the text, in the novel use of the CPI data to try and address multiple SIP mechanisms. No other study to my knowledge has attempted to do this with the theoretical basis constructed here. I do have some concerns about this approach that I'll discuss below, but I believe for this paper to have the impact that it could, the manuscript should be rewritten "to emphasize its unique contribution" (including the title), rather than "to try and make hard conclusions about which mechanism was likely most important" (I agree with the concluding section that the observations are too limited in this regard). As written, the manuscript attempts to draw hard conclusions (in places), but then undermines its own message by discussing how limited the observations are in determining the active mechanisms (lines 838-845, and 897-903).*

**Replies**:
Authors appreciate the Reviewer's comment and stimulating comments that help improving this paper.
Below are the point-by-point replies to the Reviewer's comments (in italic).

**Q:** *I found the manuscript difficult to follow, for several reasons. First, it suffers from an "identity crisis", in that the introductory section leaves the reader confused about what the main goal of the paper is. (I expand on this point below.) Then, the repetitive nature of explaining each SIP mechanism, and then addressing them separately for each case, adds up to a lot of text, and back-and-forth jumping by the reader to understand why a conclusion is different in one section versus another. And finally, the conclusions section is rather lack-luster, essentially making points that we have known for quite awhile, and can be found in numerous other manuscripts.*
**Reply**: The manuscript was a subject of significant changes to address the Reviewer's comment, make it more focused and minimize repetitive parts. The introductory part was reduced approximately two times. Other changes and modifications of the text are described in the following replies.

**Q:** *I believe that this manuscript presents a unique approach that is not adequately emphasized in the text, in the novel use of the CPI data to try and address multiple SIP mechanisms. No other study to my knowledge has attempted to do this with the theoretical basis constructed here. I do have some concerns about this approach that I'll discuss below, but I believe for this paper to have the impact that it could, the manuscript should be rewritten "to emphasize its unique contribution" (including the title), rather than "to try and make hard conclusions about which mechanism was likely most important" (I agree with the concluding section that the observations are too limited in this regard). As written, the manuscript attempts to draw hard conclusions (in places), but then*

*undermines its own message by discussing how limited the observations are in determining the active mechanisms (lines 838-845, and 897-903).*

**Reply**: Thanks for the comment. The novel use of the CPI to identify SIP was aimed to (a) identify the environmental conditions associated with SIP, and (b) attempt to assess feasibility of different SIP mechanisms based on the rich data set collected during HIWC and BAIRS2/WERVEX field campaigns.

The objective of last paragraph of the paper (*lines 897-903*) is to draw attention of researches outside of the "in-situ" community to fundamental limitations of in-situ measurements to study SIP processes. This urges to move the focus of exploration of SIP mechanisms from in-situ observations to laboratory.

**Major comments:**

*1. Section 1 is very long. Because of the recent AMS Monograph review article by Field et al. (2017), this much explanation does not seem necessary regarding the different mechanisms. I'm not opposed to having this section in there, but it reads like a review article, leaving the reader to wonder what this study is actually addressing.*

**Reply**: The authors understood that the introductory part of the manuscript was atypically long, and its length was debated among the co-authors. The motivation was related to the absence of a detailed review of lab experiments on SIP. Note, that two co-authors on this paper are co-authors on the Field et al. (2017) article. It was decided to make a decision on the length of the introduction section based on recommendations of Reviewers' comments (if any). Following the Reviewer's comment the length of introduction was reduced approximately two times.

*2. Unclear focus. Following comment #1, at the end of the Introduction a question is posed: "The question that arises is, could these observations reflect an actual occurrence of different types of SIP?" Is this then the focus of the paper? The2nd section states the objectives as "Based on the results obtained the authors attempt to revisit the role of different SIP mechanisms and identify conditions favorable for SIP." Later in the Conclusions section however, it is stated that "The obtained results are expected to contribute in our understanding of SIP, and they may be used by cloud modeling studies for evaluation of secondary ice production in the numerical simulations of clouds." So was this the real objective, to essentially create some cases for future numerical modeling? The focus is extremely important here, because it would allow the authors to address one of these completely. In the current state, the manuscript falls short of meeting any of these objectives, in that the results are not novel, nor is the dataset sufficiently described to provide a good target case for future numerical modeling.*

**Reply**: To address the Reviewer's comment the paragraph formulating objectives of this paper was rewritten to make, it coherent with the mentioned question about the occurrence of the HM process:

> "The present study is focused on revisiting the role of different SIP mechanisms and identifying conditions favorable for SIP. Cloud regions with ongoing ice multiplication were identified with the help of a new technique based on identification of small faceted ice crystals smaller than 60-100μm measured by Cloud Particle Imager (CPI). The newly developed technique was applied to the data set collected in mature tropical mesoscale convective systems (MCS) and in midlattitude frontal clouds. The roles of six possible mechanisms to generate the SIP particles are assessed using additional observations: fragmentation of freezing drops, splintering during the HM-process, ice-ice collisional breakup, ice fragmentation during thermal shock, fragmentation during ice sublimation, and INP nucleation in transient supersaturation. The variety of environmental conditions associated with SIP will be considered based on six specific cases sampled tropical MSC (4 cases) and midlattitude frontal clouds (2 cases)."

The question regarding "numerical simulation" is addressed in the the reply to Comment #3.

*3. Without more information, the observations as presented are insufficient to make a good case to test with numerical modeling. It is clearly stated that both of these cases were seeded with ice from overlaying cloud, but we are not told what temperatures the overlying clouds are, nor are the values of measured INP active at these temperatures given. How can a model get this case correct if such "initial conditions" are not provided?*

**Reply**: The objectives of this paper does not include provision of information required for initialization of numerical simulation. However, such information is readily available from the project archives, like for most recent field campaigns (e.g. weather balloon soundings, ground based measurements, satellite and weather radar

observations). The temperature of the above cloud layers is available from Convair580 vertical soundings (BAIRS3/WERVEX) or second research aircraft operated during the HIWC project. An example of numerical simulations of a tropical MCS observed during the HIWC project can be found in Qu et al. (QJRMS, 144:1681–1694, 2018). In order to address the Reviewer's comment, this publication was referenced in the text.

*4. Abstract. The final lines 27-31 are inconsistent with the discussion in the conclusions section. Please revise.*
**Reply**: TO address the Reviewer's comment the abstract was modified as:

> "This study attempts a new identification of mechanisms of secondary ice production (SIP) based on the observation of small faceted ice crystals (hexagonal plates or columns) with characteristic sizes smaller than 100 μm. Due to their young age, such small ice crystals can be used as tracers for identifying the conditions for SIP. Observations reported here were conducted in oceanic tropical mesoscale convective systems (MCS) and mid-latitude frontal clouds in the temperature range from 0°C to -15°C and heavily seeded by aged ice particles. It was found that in both MCSs and frontal clouds, SIP was observed right above the melting layer and extended to the higher altitudes with colder temperatures. The roles of six possible mechanisms to generate the SIP particles are assessed using additional observations. In most observed SIP cases, small secondary ice particles spatially correlated with liquid phase, vertical updrafts and aged rimed ice particles. However, in many cases neither graupel nor liquid drops were observed in the SIP regions, and therefore, the conditions for an active Hallett-Mossop process were not met. In many cases large concentrations of small pristine ice particles were observed right above the melting layer starting at temperatures as warm as -0.5°C. It is proposed that the initiation of SIP above the melting layer is stimulated by the recirculation of large liquid drops through the melting layer with convective turbulent updrafts. After re-entering a supercooled environment above the melting layer they impact with aged ice, freeze and shatter. The size of the splinters generated during SIP was estimated as 10 μm or less. A principal conclusion of this work is that only the freezing drop shattering mechanism could be clearly supported by the airborne in-situ observations."

This modification makes it consistent with the relevant section in conclusions (page23), which was modified as:

> "In many cases, concentrations of frozen drops and their fragments exceeding expected concentrations of INPs by orders of magnitude were observed in SIP regions. This discrepancy implies that something other than heterogenous drop freezing must be contributing to SIP. The roles of mechanisms such as HM rime-splintering, ice-ice collisional breakup, thermal shock fragmentation, and INP activation around freezing drops cannot be confidently linked to SIP based on the collected data, for reasons explained at length. Thus, we conclude by process of elimination that the mechanism of droplet shattering during freezing is very likely a critical contributing factor to SIP in these cases."

*5. Line 345: In specifying the tau values, a uz of 1- 4 m/s was specified. But earlier in the manuscript, strong updrafts were noted, up to15-20 m/s in the MCS (Line 254), and Fig. 8 shows the analysis was applied in 10-15m/s updrafts. So how can these values of tau be applicable in that case? My concern is that in stronger updrafts, the small ice particles would be leaving their place of origin much more quickly than assumed in the paper. That would make the Lmax derived in the next section MUCH smaller. (Again, having some radar pictures of the vertical velocity structures would be helpful here and might have helped me understand why you used these values in deriving the taus.)*
**Reply**: The objective of section 3 is to assess the characteristic time within which a newly born ice particle is still associated with the environmental conditions corresponding to its origin. In the frame of this paper this technique was adjusted for velocities corresponding to moderate regular updrafts and turbulent motions (1-4m/s). For higher $u_z$ the value of $L_{max}$ has to be reassessed. It is worth noting, that changing environmental conditions (e.g. air temperature from $T_a$=-5°C range to -30°C range) will result in changing of $\tau_{corr}$ and $L_{max}$. Similarly changing the range of the updraft velocities will also affect the assessment of $\tau_{corr}$ and $L_{max}$. The sensitivity of $\tau_{corr}$ and $L_{max}$ to the background parameters was addresses in the statement: "It should be noted that $\tau_p$, $\tau_{gl}$, , $\tau_t$, , $\tau_v$ are sensitive to the above parameters and may be different from the obtained estimates. However, the above assessment provides the magnitude of the characteristic times for SIP cloud regions."

To avoid confusion with Fig.8f, it should be noted that this diagram contains a convective cell with a vertical velocity peaking to $u_z \sim$15m/s as indicated by the Reviewer. However, the consideration of SIP shown in Figs.9 and 10 was performed for the convective region with 2<$u_z$<5m/s (at ~12:05:30). So, the consideration remains within the envelope of the conditions corresponding to the obtained assessments of $\tau_{corr}$ and $L_{max}$

In addition to the vertical velocities shown in Figs. 5f, 8f, 13f, the range of changing of $u_z$ can also be assessed from the X-band radar Doppler velocity shown in a newly added Fig.23d.

*6. The arguments presented in sections 3.3 and 3.4 appear logical, but because the entire study hinges on this derived Lmax, a more thorough discussion of uncertainty in its estimate, and the implications of its uncertainty, is required.*

**Reply**: $L_{max}$ does not present a precise value, but rather an estimate. The text already includes the ranges of uncertainties of $\tau_{corr}$ as 60-120s (section 3.3) and $L_{max}$ as 50μm to 150μm (section 3.4). As indicated in section3.3 $L_{max}$ depends on parameters such as $T$, $P$, $L$, $\varepsilon$, $N_d$, $N_i$, $\bar{r}_d$, $\bar{r}_i$, $\Delta T$, $u_{ice}$. Inclusion a sensitivity test of $\tau_{corr}$ and $L_{max}$ to these ten parameters does not seem to be feasible in the frame of this paper. The authors consider that further refinement of the assessment of accuracy of estimation of $\tau_{corr}$ and $L_{max}$ may unreasonably expand the paper and unnecessary here.

*7a. I have two large concerns with the CPI analysis. First, I was happy to find the analysis of shattering on the CPI in the Appendix A I've always wondered about that. But that information does not seem to be discussed much in the main manuscript. I think it should be, or at least should be discussed a little more in the main text, how those fragments were eliminated compared to the small secondary ice.*

**Reply**: The analysis of the CPI data was briefly discussed at the end of section 3.5 and Appendix A. In order to address the Reviewer's comment, the section related to identification of the shattered fragment in Appendix A was expanded. In its present form it appears as (page25):

> "Images as in **Figure A4** usually form spatial clusters with close spacing, and they appear in CPI image frames (2.3 mm ×2.3mm) as multiple images as in **Figure A3**. In this regard, the number of images in CPI image frames was used as an indicator of shattering. In this work, CPI image frames with more than one image where identified as shattering artifacts, and such frames were excluded from the analysis. The SPEC CPIview processing software was modified to recognize such image frames and discard them. Shattered fragments, which appear in the CPI imagery as single particle images (i.e. the rest fragments did not pass through the sample volume), could not be identified by this technique. However, since the entire analysis of the CPI data was built on identification and calculation of concentrations of small hexagonal prisms with $L < L_{max}$ and droplets with $D$<300μm, the unidentified shattered ice fragments in the CPI imagery did not affect outcomes of this study."

*7b. Second, I am not confident in the CPI-derived estimates of ice particle number concentration based on a scaling with water droplets from the 2DS they would not have the same shattering effect as the ice crystals, and it is not clear that the CPI cameras would necessarily trigger at the same rate for water droplets versus ice. Why not use some times when the cloud is completely glaciated, and compare the CPI image rate with the CIP numbers at those times? While I don't feel that the authors have completely misidentified SIP episodes, their magnitude is highly questionable with the current approach.*

**Reply**: The authors of the paper are sceptical regarding using a standing alone CPI for quantification of microphysical measurements. As it was discussed in few microphysical instrumentation workshops its measurements can be quantified after "anchoring" to the measurements of other probes. The "anchoring" technique was adopted in this study. For that matter the counting rate of droplets $D$>40μm measured by CPI was compared to the concentration of droplets with $D$>40μm measured by 2DS. Both identification of droplet images $D$>40μm from CPI data and calculation of concentration of droplets $D$>40μm from 2DS measurements are well established procedures. After identification of the scaling coefficient for the conversion of the CPI droplet rate into concentration, this coefficient was applied to small hexagonal crystals. This procedure is based on the assumption that the droplets and ice crystals with $L < L_{max}$ are in the same size range and they have approximately the same sample volume. In order to address the Reviewer's comment, the last paragraph in Section 3.5 was modified as:

"Due to uncertainty of the CPI sample area definition affected by the settings of acceptance out-of-focus images during sampling and post processing, we will be using counting rate ($s^{-1}$) of small faceted ice particles to characterize their concentration. The assessment of concentration of faceted ice provided in the foregoing discussion was done based on the comparisons of the CPI counting rate of droplets with $D$ >40μm and that measured by 2DS. After identification of the scaling coefficient for the conversion of the CPI droplet rate into concentration, this coefficient was applied to the counting rate of small hexagonal crystals. This procedure is based on the assumption that the droplets and ice crystals $< L_{max}$ are in the same size range and their CPI sample volumes are approximately the same. The accuracy of such estimation of concentration of small ice particles is estimated as ±50%."

*8. Sections 7 and 8 could be condensed into more focused text, which would greatly help the reader comprehend the main points.*
**Reply**: After re-reading Sections 7 and 8 the authors consider that these sections are concise enough and further reduction of its volume may affect clarity of its presentation.

*9. Lines 769-770:here is where a discussion of shattering on the CPI, and the removal of those particles and its uncertainty, is extremely important (and in the following sections as well).*
**Reply**: The uncertainty of removal of shattered particles was discussed in comment #7a of the Replies.

*10. Melting Layer Hypothesis: Especially in the maritime MCS, why can't the large super-cooled drops just be growing in updraft? why would they have to be recycled melted particles to be important? And if they were heavy enough to fall and melt, what mechanism would bring them back into an updraft? This section seems to really "reach" past what one can identify with your observations. Why not just let the data speak for themselves? Dual-polarization data would really be needed to pin this down.*
**Reply**: The concentration of droplets and LWC in MCS appears to be too low to grow drizzle size drops through collision-coalescence. However, the collision-coalescence process should not be ruled out in frontal clouds during BAIRS2/WERVEX project. We agree with the Reviewer that if ice particles managed to fall through the updraft they won't recirculate back above the melting layer. However, if an updraft occurred after formation of a melting layer, then the mechanism of recirculation becomes feasible. In order to support the recirculation hypothesis a new Fig.23 was added. This figure shows deformation of the bright band in convective updrafts and structure of Doppler velocity in the vicinity of the melting layer measured by X-band radar. In order of address the set of questions in Comment 10 the following text was added to section 9:

"The recirculation hypothesis is supported by observation of distortion of the bright band altitude in the convective cloud regions. An example of such distortion is presented in **Fig.23**. **Figure 23** shows a zoomed segment of the time series in **Fig.5**, which includes reflectivity (c) and Doppler velocity (d) measured by onboard X-band radar, when traversing a convective cell in the tropical MCS (09:40-09:45). Comparison of panels (b) and (c) in **Fig.23** shows a peak-to-peak correlation between the vertical wind velocity and elevation of the bright band in the convective cell. In few points the bright band moves up to ~600-700m above the level of the bright band in undisturbed cloud regions (indicated by dashed line **Fig.23c,d**). Such distortion of the bright band is explained by moving melted drops by vertical updrafts to higher levels. A spatial coincidence of the SIP area (**Fig.23a**), convective updraft (**Fig.23b**) and the region with the elevated bright band (**Fig.23c**) is supportive of the droplet re-circulation hypothesis.

In order for a drop to ascend through the melting layer, the velocity of the updraft ($u_z$) should exceed the drop fall velocity ($u_{fall}$). **Figs.5f** and **13f** show examples, when the vertical velocity above the melting layer in the tropical MCS reached $u_z$ ≈8m/s and in frontal clouds $u_z$ ≈3m/s, respectively. Such updraft velocity is sufficient to move drops with $D$=100-200μm ($u_{fall}$ =0.3-1m/s at $P$=500mb) through the melting layer ($\Delta Z$=500m) during a reasonable time of few tens of seconds to a few minutes.

The vertical travel distance of the liquid drops formed in the melting layer depends on the sustainability and endurance of the convective updraft, its vertical velocity and droplet size. Smaller droplets have higher chances to travel deeper in the cloud compared to large ones. This is consistent with the observation of occurrence of droplets with $D$=80μm and 100μm as shown in **Figs.5b** and **8b**,

which were measured in the same MCS at two different altitudes 5600m and 7000m, respectively. Rapid decrease of the concentration of large drops with temperature (and therefore altitude) in tropical MCSs is also seen from **Fig.11**.

Another explanation of formation of drizzle size drops is related to the collision-coalescence process. However, the observed LWC and number concentration of cloud droplets with $D<40\mu m$ in a mature tropical MCS during HIWC typically varied in the ranges $0.01<LWC<0.1g/m^3$ and $5<N_{dr}<40cm^{-3}$, respectively, and were always associated with mixed phase dominated by ice $0.5<IWC<3g/m^3$ (e.g. **Figs.5dg** and **8dg**). High $IWC$ and low $N_{dr}$ and $LWC$ will hinder the collision-coalescence process due to riming and WBF processes, which result in depletion of droplets. However, the collision-coalescence process cannot be ruled out in midlattitude frontal clouds as in **Fig.13** "

**Minor Comments:**

*1. Line 133: I think you mean "overestimated" rather than "under-estimated".*
**Reply**: Thanks for noticing. However, this text was deleted after reduction of Introduction.

*2. Lines 170-175: This assumes that air at the cloud edge containing sublimating crystals is incapable of being reintroduced into the core of the cloud; numerous studies have shown that entrainment in cumuliform clouds can bring outside air into the interior of the cloud (but no studies have looked at the effects on ice, to my knowledge.)*
**Reply**: The authors agree with this comment. After all, the question is whether small ice fragments can survive the dry environment and re-enter the cloud.

*3. Lines 237-243: "One of the important finding of this study is that melting layers in many cases work as a source of large liquid drops, which then ascended to a supercooled environment via convective or turbulent updrafts. After impaction freezing by pre-existing ice, the drops may shatter and initiate a chain reaction of secondary ice particle production." Why is this stated here in the "objectives and data sets" section? It's written like an abstract that lists the results. It's very confusing here- please move this to the abstract or the conclusions section.*
**Reply**: This section was rewritten and moved to Introduction.

*4. Lines 252-254: some evidence of this glaciation would be good. Can't this be shown with observations and radar data? This might help the reader understand the nature of the "seeding" of these clouds, as referenced in Line 237.*
**Reply**: This study was presented at the poster session at the AMS Cloud Physics Conference in 2018. Unfortunately, no peer reviewed publication is available yet. The conference reference was added.

*5. Line 271: I'm pretty sure you didn't have an X-band radar mounted on the Convair. Please fix text.*
**Reply**: Convair580 has two radars installed onboard: X-band and W-band. Some results of the measurements of the X-band radar are presented in added Fig.23cd.

*6. Equation 6 needs a reference, or some ex-planation.*
**Reply**: Reference was added as per Reviewer's comment

*7. Lines 455-461: I am not convinced that one can tell from the CPI images if droplets spread out sufficiently to prevent rime-splintering from occurring. Comparing it to the roughness of the rods used in the laboratory (in the next paragraph) seems unreasonable*
**Reply**: Below are two sets of images of graupel from Fig.6 (top) and Fig.9 (bottom). The particles in the top row appears to have smooth surfaces with lack of features which could splinter. However, images graupel in the bottom row have rough surface. Such particles may generate splinters on mechanical impact or result in droplet

shattering, which freeze on their surfaces. The roughness of the surface of riming ice particles is determined by temperature and size of droplets, LWC and fall velocity.

---

## Author Comment (AC1)

**Replies to Reviewer 1 comment on "A new look at the environmental conditions favorable to secondary ice production" by Alexei Korolev et al.**

*I have a question and a comment to communicate. The question is, were the flight plans made to take APIPs into account? (aircraft-produced ice particles) This could be a serious issue especially, of course, when repeated penetrations in the same cloud have been made. The comment is (after saying that this seems to me a very important paper, and should be accepted) that the small ice particles that look like either fat ice columns or very thick, small plates. look to me like overgrowths on frozen, large, cloud droplets, not a result of overgrowths on the sort of shards that would result from the little explosions of freezing droplets. They are not at all what I would have expected from that. That does not help in identifying the actual cause of the freezing, of course, which seems to me the main mystery.*
*Charles Knight*

**Replies**
Authors appreciate the Reviewer's comments and time spent to evaluate this work.
Below are the point-by-point replies to the Reviewer's comments (in Italic).

**Q**: *The question is, were the flight plans made to take APIPs into account? (aircraft-produced ice particles) This could be a serious issue especially, of course, when repeated penetrations in the same cloud have been made.*
**Reply**: The effect of APIP on the measurements is excluded for the cases 1,2,5,6. But, APIP may potentially affect cases 3 and 4. However, the authors consider that the APIP will be advected away from the area of measurements by the vertical updraft Uz~2-5m/s. In order to address the Reviewer's comments a section 4.3 was added in the text.

**4.3 Effect of aircraft produced ice particles on the measurements**
Aircraft-produced ice particles (APIP) (e.g. Rangno and Hobbs, 1983; Woodley et al., 1991) may be confused with SIP ice crystals, and therefore, result in biases in interpretation of measurements. Contamination by APIP may occur if the aircraft re-enters the cloud region where the APIP were transported by vertical or horizontal advection. Typically, this may happen if the aircraft traverses through the region of its previous operation.

The contamination by APIP is excluded for the cases 1 and 2 (**Figs.6,7**) (sections 4.1.1, 4.1.2) since the Convair580 flew along a nearly straight line and never re-entered regions of earlier operations (**Fig.4a**). The cases 3 and 4 (**Figs.9,10**) (sections 4.1.3, 4.1.4) might be contaminated by APIP since the clouds were sampled in an area close to which the Convair580 flew 8 minutes earlier. However, since cases 3 and 4 were sampled in a convective region with an updraft velocity $u_z$=2-5m/s (**Fig.8f**), the potential APIP were expected to be removed from the area of the measurements by vertical wind.

Case 5 (**Fig.14**) (section 4.2.5) was sampled during ascent through the cloud (Fig.13h) at approximately 12:30 (see also **Fig.12a**). This cloud region was not affected by the previous operation of the Convair580, and therefore, contamination by APIP of this area is dismissed. Similarly, case 6 (Fig.15) (section 4.2.6) was sampled during descent through a mixed phase layer, which was not affected by previous Convair580 flight operations.

**Q**: *The comment is (after saying that this seems to me a very important paper and should be accepted) that the small ice particles that look like either fat ice columns or very thick, small plates. look to me like overgrowths on frozen, large, cloud droplets, not a result of overgrowths on the sort of shards that would result from the little explosions of freezing droplets. They are not at all what I would have expected from that. That does not help in identifying the actual cause of the freezing, of course, which seems to me the main mystery.*
**Reply**: This is one of the findings of this study, that the aspect ratio (R) of the small hexagonal ice particles observed in the same cloud region may vary in wide range 0.3<R<6. Large fraction of the small pristine ice particles has a nearly isometric shape with R~1. As discussed in the paper, most likely such shape is related to freezing of small droplets due to their impact with ice splinters. Further development of understanding of this question requires laboratory studies and it goes beyond the frame of this work.

---

## Referee Report (RR1)

Review of MS acp-2019-611, revised version, Nov. 22, 2019

This is an important paper, taking advantage of advances in imaging cloud ice particles from airplanes to deduce  whatever may be deducible about secondary ice production, an important and long-standing problem in cloud physics.  It is acceptable for publication, I think, in its present form, but I will make several suggestions for minor revisions in the presentation.

Line 121  extent instead of extend; Lines 131-132 -- the last two sentences of this paragraph do not communicate much to a reader at this point, without a physical explanation of how the HM process is expected to operate and what the alternatives are.

Section 3.1  I don't agree with the use of the word "assumptions" here.  The two on lines 191-193 are more properly called approximations.  I think the argument that most of the small, hexagonal ice xtals derive from "secondary ice" is very strong, but we don't need to "assume" it.  And number 2 likewise is not an "assumption" in any sense of the word -- it's an approach used in the analysis.  Then on p. 6, the use of the word "characteristic" is, to me, quite inappropriate.  I would favor "typical" or "approximate" sizes or residence times.  I do understand what the author means here, but upon encountering the word characteristic so many times, at first I couldn't follow the meaning.

Line 291 -- using counting rate instead of concentration takes the meaning out of the measurement.  It is explained that the sample volume is quite uncertain, but the concentration is of course what is important, so it should be identified here in the text and it should be noted that the concentration scale is on the figure.  Personally, I would have put the concentration scale on the left and the counting scale on the right in the figures.  The explanation of the sample volume problem is good.

Lines 534-538  The argument about small particles having shorter residence times than larger ones doesn't make any sense at all to me, nor is it particularly important to the argument here, I think.  "Residence time" must depend critically upon updraft, downdraft, turbulence, fall speed, and the various possible "sinks."  Small ice in mixed or supercooled cloud has limited "residence time" mainly because it grows past being small, obviously.  Line 543 "characteristic" again.  Not a good word, for me.  But the small hexagonal plates are a wonderful observation.  It's too bad that there aren't any comparable small needles around -5C!, but maybe they would be too thin

for the instrument to detect.  The thin plates and their interpretation are for me a rather wonderful observation, and surely they grow around -1 or -2C, but the original secondary ice "must" have descended  from above??

Line 798 --Now it's 10? I thought it was more like 20, before.

I think that temperatures should be included in the figure captions for all of the multi-image figures.

This paper generates suggestions in me, for field and laboratory approaches testing some of the rather speculative (but not unreasonable) interpretations of the ice data.

The many other suggested secondary ice mechanisms are mentioned in the introduction and then each is again discussed at some length near the end.  This is a very long paper, and for my taste, I would have left out the re-cap of every secondary ice thought, at the end.  Perhaps in favor of more details about what the difference might be between the recycling-water-drops hypothesis and the HM process ("rime shattering," though the actual mechanism behind the lab results seems to me not demonstrated).

I recommend acceptance.  This is an important work.

---

## Author Response (AR2)

**Replies to the Reviewer 1 comments**
**Second Review of MS acp-2019-611, revised version, Nov. 22, 2019**

This is an important paper, taking advantage of advances in imaging cloud ice particles from airplanes to deduce whatever may be deducible about secondary ice production, an important and long-standing problem in cloud physics. It is acceptable for publication, I think, in its present form, but I will make several suggestions for minor revisions in the presentation.
Reply: Authors appreciate Reviewer's careful reading, valuable comments and time spent to evaluate this manuscript.

Line 121 extent instead of extend;
Reply: Corrected as per referee comment

Lines 131-132 -- the last two sentences of this paragraph do not communicate much to a reader at this point, without a physical explanation of how the HM process is expected to operate and what the alternatives are.
Reply: Unfortunately, in the cloud physics community there is no consensus regarding the physical process responsible for the HM process. As mentioned in the introduction (lines 67-75) there were several works aiming to understand the physical mechanism of the HM process (Macklin, 1960; Choularton et al., 1978, 1980; Emersic and Connolly, 2017). However, the explanation of this phenomenon is still under debate. In order to address the Reviewer's comment the last two sentences were modified as: **"As can be seen, the identification of SIP gravitates towards the HM-process, whereas mechanisms such as activation of INP in transient supersaturation around freezing drops, ice fragmentation due to thermal shock or sublimation were not even considered. In this regard the question that arises is, could these observations reflect an actual occurrence of different types of SIP?"**

Section 3.1 I don't agree with the use of the word "assumptions" here. The two on lines 191-193 are more properly called approximations. I think the argument that most of the small, hexagonal ice xtals derive from "secondary ice" is very strong, but we don't need to "assume" it. And number 2 likewise is not an "assumption" in any sense of the word -- it's an approach used in the analysis.
Reply: The term "assumption" was replaced by "approximation" as per Reviewer's comment.

Then on p. 6, the use of the word "characteristic" is, to me, quite inappropriate. I would favor "typical" or "approximate" sizes or residence times. I do understand what the author means here, but upon encountering the word characteristic so many times, at first I couldn't follow the meaning.
Reply: The term "characteristic" was replaced by "typical" as per Reviewer's comment.

Line 291 -- using counting rate instead of concentration takes the meaning out of the measurement. It is explained that the sample volume is quite uncertain, but the concentration is of course what is important, so it should be identified here in the text and it should be noted that the concentration scale is on the figure. Personally, I would have put the concentration scale on the left and the counting scale on the right in the figures. The explanation of the sample volume problem is good.

Reply: The concentration and counting rate scales in Figs. 5, 8 and 13 were reversed as suggested by the Reviewer.

Lines 534-538 The argument about small particles having shorter residence times than larger ones doesn't make any sense at all to me, nor is it particularly important to the argument here, I think. "Residence time" must depend critically upon updraft, downdraft, turbulence, fall speed, and the various possible "sinks." Small ice in mixed or supercooled cloud has limited "residence time" mainly because it grows past being small, obviously.

Reply: I believe there is some misinterpretation of the text. The original statement in Lines 534-538 said that **small** SIP particles will stay longer in the environment of their origin, whereas **large** SIP particles have shorter residence time in clouds. This is consistent with the Reviewer's comment. The statements about the residence times are important for cloud simulations.

Line 543 "characteristic" again. Not a good word, for me. But the small hexagonal plates are a wonderful observation. It's too bad that there aren't any comparable small needles around -5C!, but maybe they would be too thin for the instrument to detect.

Reply: The term "characteristic" was replaced by "typical" following the Reviewer's comment.

The thin plates and their interpretation are for me a rather wonderful observation, and surely they grow around -1 or -2C, but the original secondary ice "must" have descended from above??

Reply: Thank your for the comment. Yes, small SIP fragments descending from above (e.g. -3C to -4C) due to turbulent diffusion and re-growing into plates at -1C to -2C is a possible scenario.

Line 798 --Now it's 10? I thought it was more like 20, before.

Reply: In the text in Section 5 the thickness of plates was assessed as 10μm to 20μm. Since the initial size of the fragment should be smaller than the thickness of the plate, the *smallest* size of the SIP fragments is estimated as 10μm or less. In order to mitigate ambiguity in the interpretation of the text in Line 789, it was modified as: "**The smallest size of the splinters generated during SIP were estimated at 10m or less.**"

I think that temperatures should be included in the figure captions for all of the multi-image figures.

Reply: Following the Reviewer's comment the temperature was added to multi-image figures where it was missed (Figs.18,20,21,24)

This paper generates suggestions in me, for field and laboratory approaches testing some of the rather speculative (but not unreasonable) interpretations of the ice data.

Reply: The authors absolutely agree with this comment.

The many other suggested secondary ice mechanisms are mentioned in the introduction and then each is again discussed at some length near the end. This is a very long paper, and for my taste, I would have left out the re-cap of every secondary ice thought, at the end. Perhaps in favor of more details about what the difference might be between the recycling-water-drops hypothesis and the HM process ("rime shattering," though the actual mechanism behind the lab results seems to me not demonstrated).

Reply: The authors considers that the section at the end, which reiterates the considerations of different SIP mechanisms, is important here. It shows limitations of the in-situ observations in identifying different SIP mechanisms from in-situ observation and prepare a background for the statement about importance of lab SIP studies. At the same time, we cannot expand the discussion of the rime-shattering mechanism, without making it too speculative. More information is required to make the next step.

I recommend acceptance. This is an important work.
Reply: Authors appreciate the Reviewer's evaluation of this work.

[Figure]

[Figure]

**Atmospheric Science and Technology Branch**
**4905 Dufferin Street, ARMP**
**Downsview, Ontario**
**M3H 5T4    CANADA**

Atmospheric Chemistry and Physics

07 December 2019

**RE: 2nd revision of ACP-2019-611 paper**

Dear Martina,

Attached please find the 2nd revised version of the manuscript titled: "A new look at the environmental conditions favorable to secondary ice production" by A. Korolev, I. Heckman, M. Wolde, A.S. Ackerman, A.M. Fridlind, L. Ladino, P. Lawson, J. Milbrandt, E. Williams submitted for evaluation for publication in the Atmospheric Chemistry and Physics along with the point-by-point replies to the Reviewers' comments.

We are looking forward seeing your decision.

Sincerely,

Alexei Korolev,
Research Scientist
tel: (416) 739 5716
e-mail: alexei.korolev@canada.ca